# A Comprehensive Survey on Knowledge Distillation

**Amir M. Mansourian**                                          *amir.mansurian@sharif.edu*
*Sharif University of Technology*

**Rozhan Ahmadi**[*]                                           *roz.ahmadi@sharif.edu*
*Sharif University of Technology*

**Masoud Ghafouri**[*]                                         *masoud.ghafouri98@sharif.edu*
*Sharif University of Technology*

**Amir Mohammad Babaei**[*]                                    *amir.babaei79@sharif.edu*
*Sharif University of Technology*

**Elaheh Badali Golezani**[*]                                  *elahe.badali@sharif.edu*
*Sharif University of Technology*

**Zeynab Yasamani Ghamchi**[*]                                 *zeynab.yasamani77@sharif.edu*
*Sharif University of Technology*

**Vida Ramezanian**[*]                                         *vidaramezanian@sharif.edu*
*Sharif University of Technology*

**Alireza Taherian**[*]                                        *alireza.taherian49@sharif.edu*
*Sharif University of Technology*

**Kimia Dinashi**                                              *kimia.dinashi@sharif.edu*
*Sharif University of Technology*

**Amirali Miri,**                                              *amirali.miri79@sharif.edu*
*Sharif University of Technology*

**Shohreh Kasaei**                                             *kasaei@sharif.edu*
*Sharif University of Technology*

**Reviewed on OpenReview:** *https://openreview.net/forum?id=3cbJzdR78B*

## Abstract

Deep Neural Networks (DNNs) have achieved notable performance in the fields of computer vision and natural language processing with various applications in both academia and industry. However, with recent advancements in DNNs and transformer models with a tremendous number of parameters, deploying these large models on edge devices causes serious issues such as high runtime and memory consumption. This is especially concerning with the recent large-scale foundation models, Vision-Language Models (VLMs), and Large Language Models (LLMs). Knowledge Distillation (KD) is one of the prominent techniques proposed to address the aforementioned problems using a teacher-student architecture. More specifically, a lightweight student model is trained using additional knowledge from a cumbersome teacher model. In this work, a comprehensive survey of knowledge distillation methods is proposed. This includes reviewing KD from different aspects: distillation sources, distillation schemes, distillation algorithms, distillation by modalities, applications of distillation, and comparison among existing methods. In contrast to most existing sur-

---

[*]Equal contribution.

veys, which are either outdated or simply update former surveys, this work proposes a comprehensive survey with a new point of view and representation structure that categorizes and investigates the most recent methods in knowledge distillation. This survey considers various critically important subcategories, including KD for diffusion models, 3D inputs, foundational models, transformers, and LLMs. Furthermore, existing challenges in KD and possible future research directions are discussed. Github page of the project: `https://github.com/IPL-Sharif/KD_Survey`

## 1 Introduction

With the emergence of DNNs, the fields of Computer Vision (CV) and Natural Language Processing (NLP) have been revolutionized, and most tasks in these domains are now solved using DNNs. While a simple ResNet (He et al., 2016) model or BERT (Kenton & Toutanova, 2019) can be easily trained on most of existing GPUs, the advent of large models like LLMs and foundational vision models has made training and inference a significant challenge in terms of runtime and memory usage, especially for deployment on edge devices like mobile phones due to their widespread use. Despite their high performance, these large models often have complex architectures and are heavily over-parameterized (Kaplan et al., 2020; Fischer et al., 2024). For example, the weight matrices in LLMs have been shown to be low-rank matrices (Hu et al., 2021).

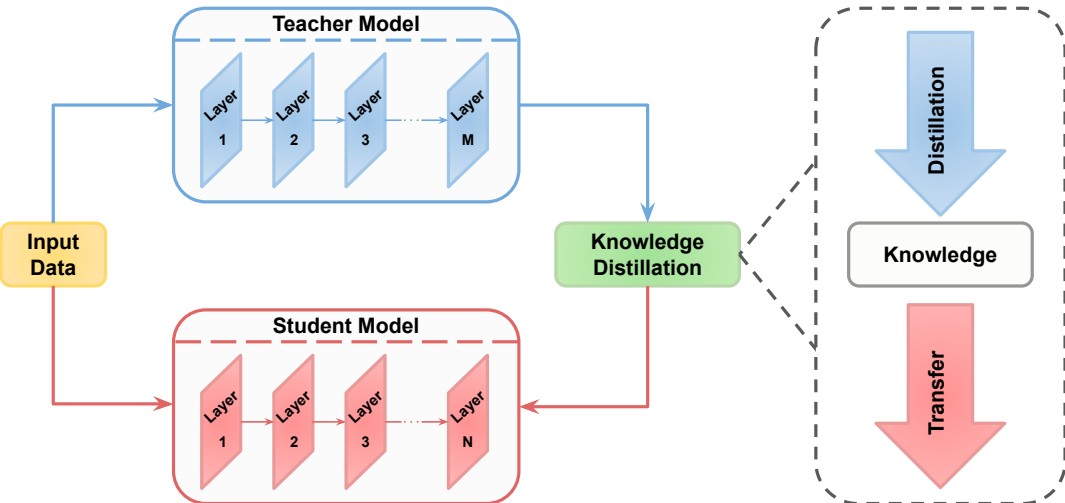

Figure 1: A general teacher-student framework for knowledge distillation.

To address the issue of large models, various solutions have been proposed, such as efficient network architectures, compression methods, pruning, quantization, low-rank factorization, and knowledge distillation. Efficient network blocks, including MobileNet (Howard, 2017), ShuffleNet (Zhang et al., 2018), and BiSeNet (Yu et al., 2018) have been introduced in recent years. Pruning methods, a type of compression technique, aim to remove unnecessary layers and parameters from models with minimal impact on performance. Low-rank factorization methods reduce parameters through matrix decomposition. Unlike other methods, KD does not modify the network's layers or parameters. In KD, a lightweight network (student) is trained under the supervision of a more complex model (teacher) with deeper architecture and greater number of parameters. This concept, first proposed by Buciluă et al. (2006) and later popularized by Hinton (2015) as KD, introduced the idea of training the student network by mimicking the teacher's output distribution as soft labels.

Aside from the large number of parameters, large models also require a significant amount of labeled data for training. Another purpose of KD is knowledge transfer, which can involve transferring knowledge from a

source task to a target task that lacks sufficient labeled data. Additionally, in cases where data privacy is a concern, data-free knowledge distillation becomes helpful. This approach addresses the issue by generating synthetic data, eliminating the need to store sensitive data. The key challenges in KD are determining what knowledge to transfer, selecting the appropriate algorithm, and designing the student and teacher architectures. A diagram of a general teacher-student KD method is shown in Figure 1.

With the significant growth in the number of published papers in the field of KD and its wide applications across various tasks and domains, several survey papers have been presented to review KD from different perspectives. The prominent work Gou et al. (2021) provides a comprehensive survey on KD, reviewing it from the aspects of knowledge types, algorithms, and schemes, while including comparisons between different methods. Another notable work, Wang & Yoon (2021), offers an overview of model compression based on the teacher-student architecture specifically for vision tasks. Alkhulaifi et al. (2021) presents a survey on KD and proposes a new metric for comparing distillation methods based on size and accuracy, while Yang et al. (2023c) categorizes existing approaches based on the type of knowledge source used for distillation. More recently, Hu et al. (2023a) delivers a comprehensive survey, exploring knowledge optimization objectives associated with various representations, and Moslemi et al. (2024) updates earlier surveys by providing an extensive review of KD methods. Additionally, they briefly investigate distillation in VLMs and discuss the challenges of distillation under limited data scenarios.

While each of these surveys provides a detailed summary of papers from different perspectives, they also have some drawbacks. First, although these surveys analyze existing approaches in each category, they fail to address recent advancements, particularly in feature-based distillation methods. Despite the significant growth of feature-based distillation, which now dominates state-of-the-art methods, many recent and impactful works are overlooked. Furthermore, adaptive distillation and contrastive distillation algorithms are rarely discussed. Second, with the recent emergence of foundation models and LLMs, these surveys have largely overlooked their potential for distilling knowledge into other models. One significant shortcomings of Gou et al. (2021), as the most prominent survey in the field, is its lack of explanation regarding the applications of distillation in newly introduced models such as foundation models and LLMs, which have gained significant attention in recent years. Third, none of the existing surveys have investigated KD for 3D inputs like point clouds. As 3D tasks garner increasing attention in top research venues, the lack of sufficient approaches and models in comparison to the image domain highlights the importance of KD as an effective method for training the models in this area. Table 1 shows a summarized comparison between existing surveys and this work.

In this work, a comprehensive survey on knowledge distillation is presented. This includes reviewing existing KD methods from different perspectives: the source of distillation, distillation algorithms, distillation schemes, distillation by modalities, and applications of KD.

The sources reviewed include logit-based, feature-based, and similarity-based distillation methods. In contrast to existing surveys, a more comprehensive classification of these methods is presented, highlighting recent advancements, particularly in feature-based distillation.

Regarding algorithms, methods in attention-based, adversarial, multi-teacher, cross-modal, graph-based, adaptive, and contrastive distillation are reviewed. Notably, adaptive and contrastive distillation are two recent categories that, despite their importance, have yet to be presented in previous works.

In terms of schemes, offline, online, and self-distillation methods are covered. Compared to previous surveys, a new section is introduced that classifies existing KD methods by modalities such as video, speech, text, multi-view, and 3D data.

In applications, a comprehensive exploration is conducted on the applications of KD in important areas including self-supervised learning, foundational models, transformers, diffusion models, visual recognition tasks, and LLMs. Finally, a quantitative comparison of prominent methods is provided, along with a discussion of current challenges and future directions.

Figure 2 shows the organization of this paper. In summary, the main contributions of this work are:

Table 1: Comparison between existing KD surveys and the current survey. New topics not covered in previous works are highlighted in bold to indicate the progress and scope of this survey.

| Paper | Source | Algorithm | Modality | Application |
|---|---|---|---|---|
| Gou et al. (2021) | Logit, Feature, Similarity | Adversarial, Attention, Cross-modal, Data-free, Graph, Lifelong, Multi-teacher, NAS, Quantized | Image, Text, Speech, Video | Visual Recognition |
| Wang & Yoon (2021) | Logit, Feature, Similarity | Adversarial, Cross-modal, Data-free, Few-shot, Graph, Incremental, Multi-teacher, Reinforcement | Image, Video | Visual Recognition, Self-supervised Learning |
| Hu et al. (2023a) | Logit, Feature, Similarity, Mutual Information | Cross-modal, Federated, Graph, Multi-teacher | Image, Text, Speech, Video | Visual Recognition, Ranking, Generation, Regression |
| Moslemi et al. (2024) | Logit, Feature, Similarity | Adversarial, Attention, Cross-modal, Data-free, Graph, Lifelong, Multi-teacher, NAS, Quantized | Image, Text | Vision-Language Models, Medical Image |
| This paper | Logit, Feature, Similarity | **Adaptive**, Adversarial, Attention, **Contrastive**, Cross-modal, Graph, Multi-teacher | **3D/Multi-view**, Image, Text, Speech, Video | Visual Recognition, **Foundation Models**, **Transformers**, **LLMs**, **Diffusion Models**, Self-supervised Learning |

- Presenting a comprehensive survey on knowledge distillation as an essential area of research. This includes reviewing existing methods based on distillation sources, distillation algorithms, distillation schemes, modalities, and applications of distillation.

- Classifying recent distillation methods by sources, with a particular focus on feature distillation methods due to their importance and widespread application.

- Introducing two new distillation algorithms: adaptive distillation and contrastive distillation. These important categories have gained significance in distillation, especially with the advent of foundation models like CLIP, where contrastive distillation plays an increasingly crucial role.

- Reviewing the distillation methods for multi-view and 3D data. Due to the significant role of KD in 3D tasks nowadays, exploring distillation in the 3D domain is crucial, yet it has been overlooked in previous comprehensive surveys.

- Exploring the applications of KD in self-supervised learning, foundation models, transformers, diffusion models, and LLMs. Distillation from foundation models is a hot topic, and distillation in LLMs is particularly critical due to their large number of parameters.

- Comparing prominent distillation methods and discussing the existing challenges and future directions of KD.

## 2 Sources

The primary component of a distillation method is the source of knowledge being distilled. Various sources of knowledge can be utilized for distillation. In the initial concept of distillation (Hinton, 2015), logits were employed as the teacher's final output. Alongside logit distillation (Ba & Caruana, 2014; Mirzadeh et al., 2020; Hao et al., 2023; Sun et al., 2024a), the activations (Heo et al., 2019b), neurons, and features (Romero et al., 2014; Zagoruyko & Komodakis, 2016) of intermediate layers play a crucial role in guiding the student network. Moreover, relationships and similarities among channels (Liu et al., 2021c), instances (Tung & Mori, 2019), and classes (Wang et al., 2020d) provide higher-order information that facilitates the transfer of

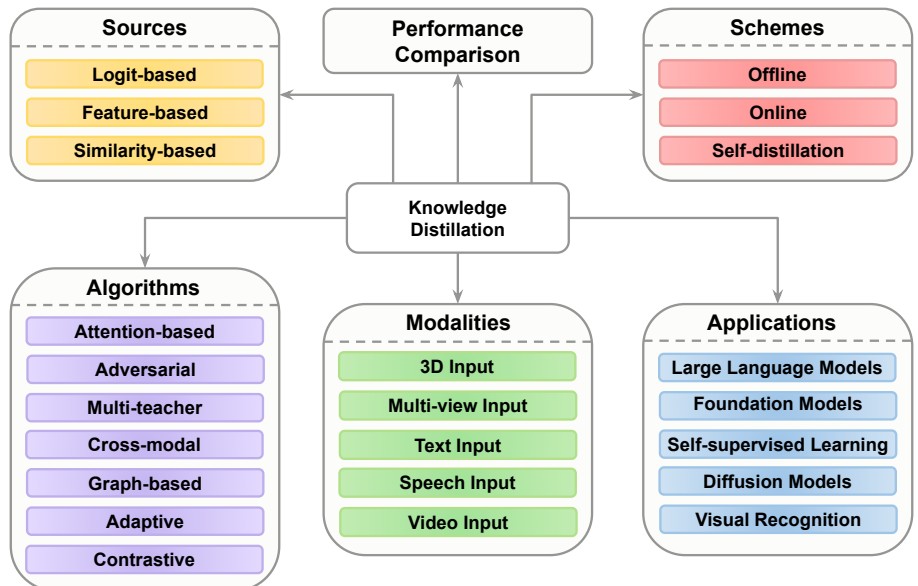

Figure 2: Diagram of the contents of this paper. Knowledge distillation is reviewed from different aspects, including sources, schemes, algorithms, modalities, and applications.

knowledge from the teacher to the student. In this section, KD methods based on their sources of distillation are examined. Specifically, existing methods are classified into three categories: Logit-based, Feature-based, and Similarity-based as shown in Figure 3. Table 2 summarizes logit-based and similarity-based distillation methods and Table 3 presents detailed information on existing feature distillation methods. Subsequently, each type of distillation is explained in detail.

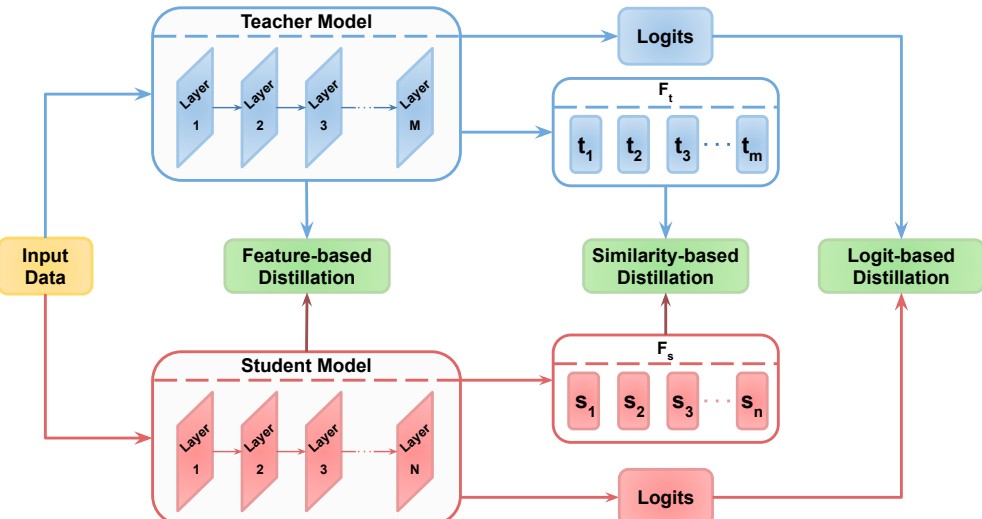

Figure 3: Illustration of the sources for distillation, including logits, features, and similarities. Logits are the model's last layer output, features are intermediate outputs of the model, and similarities are relationships between features, channels, samples, etc.

## 2.1 Logit-based Distillation

The most straightforward source for KD is logits. Logits are outputs of the last layer of the model, and the idea is to force the student model to mimic the final predictions of the teacher, which are supposed to contain informative dark knowledge (Hinton, 2015) of the teacher. The prediction from the last layer can vary for each task. For example, logits in classification contain predictions for each class, in detection they contain predicted coordinates, and in pose estimation, they can represent heatmaps. In general, let $Z_T$ and $Z_S$ be logits of the last layers of the teacher and student, respectively. Then the logit-based distillation loss ($L_{ld}$) is defined as

$$L_{ld}(Z_T, Z_S) = \ell_{logit}(Z_T, Z_S) \tag{1}$$

where $\ell_{logit}$ is a metric for measuring the difference between the logits.

The first papers on KD were proposed for image classification, where soft-labels were introduced by applying softmax on logits (Hinton, 2015; Ba & Caruana, 2014). These output distributions, or soft-labels, provide information on the probability of the input belonging to each class. The distillation loss function for soft-labels is expressed as

$$p(Z_i, \tau) = \frac{exp(Z_i/\tau)}{\Sigma_i exp(Z_i/\tau)} \tag{2}$$

$$L_{ld}(p(Z_t, \tau), p(Z_s, \tau)) = KL(p(Z_t, \tau), p(Z_s, \tau)) \tag{3}$$

where $exp$ is softmax function, $\tau$ is the temperature factor that controls the softening of logits, and $KL(\cdot)$ is the Kullback–Leibler Divergence (KL) function. By optimizing the above objective, the student can mimic the output distribution of the teacher and make better predictions.

Although the initial concept of KD was simple and effective, many different logit-based methods have been proposed in recent years (Sau & Balasubramanian, 2016; Mirzadeh et al., 2020; Song et al., 2023b; Hao et al., 2023; Xu et al., 2023d; Lu et al., 2025). These methods mainly aim to normalize logits before distillation (Guo et al., 2020a; Chi et al., 2023; Miles & Mikolajczyk, 2024; Zheng & Yang, 2024; Sun et al., 2024a), soften or smoothen the logits (Sau & Balasubramanian, 2016; Li et al., 2023a; Yuan et al., 2024b), or decouple the logit-distillation loss (Zhao et al., 2022a; Li et al., 2024a; Ding et al., 2024; Cui et al., 2024; Wei et al., 2024b; Liu et al., 2024d). Guo et al. (2020a) finds that the magnitude of confidence is not necessary for KD and proposes spherical KD to reduce the gap between teacher and student. Chi et al. (2023) shows that it is not feasible to soften all the samples with a constant temperature and proposes NormKD to customize the temperature for each sample according to the characteristics of the sample's logit distribution. Miles & Mikolajczyk (2024) and Zheng & Yang (2024) investigate the role of projectors and temperature in the KD process respectively. Sun et al. (2024a) suggests logit standardization by setting the temperature as the weighted standard deviation of the logit and performing a plug-and-play Z-score pre-process. On the other hand, among methods that try to soften the logits, Sau & Balasubramanian (2016) perturbs the logits for distillation with some noises, and Yuan et al. (2024b) softens the logits before KD and utilizes a learning simplifier with an attention module. However, most existing methods aim to decouple the distillation loss. For instance, Zhao et al. (2022a); Li et al. (2024a); Ding et al. (2024) separate the distillation of target-class and non-target class distributions. Wei et al. (2024b) reshapes the logits for multi-scale logit distillation, Liu et al. (2024d) uses different weights for distilling edges and bodies of objects, and Cui et al. (2024) decouples the KL to a weighted MSE and a cross-entropy loss that incorporates soft-labels.

In summary, logit distillation is a simple and straightforward method that can be likened to label smoothing (Kim & Kim, 2017) and regularization (Müller et al., 2019; Ding et al., 2019), aiming to enhance the student network's performance by preventing overfitting (Gou et al., 2021). However, due to the reliance of logits

on predefined classes, its application is limited to supervised learning. Moreover, mimicking the last-layer predictions does not provide insights into the intermediate layers of the teacher network. It can be challenging for the student to learn effectively solely from the final outputs of the teacher, especially when teacher and student networks have different architectures. Therefore, in addition to logit distillation, there is a necessity to distill intermediate outputs and similarities from the teacher network to facilitate effective knowledge transfer.

Table 2: Summary of logit-based and similarity-based distillation methods.

| Source | Sub-category | Description | Reference |
|---|---|---|---|
| Logit | Logit Normalization | Standardizes the logits before distillation | KD (Hinton, 2015), SphericalKD (Guo et al., 2020a), NormKD (Chi et al., 2023), Miles et al. (Miles & Mikolajczyk, 2024), TTM (Zheng & Yang, 2024), Sun et al. (Sun et al., 2024a) |
| | Logit Softening | Smooths the logits by adding noise | Sau et al. (Sau & Balasubramanian, 2016), Li et al. (Li et al., 2023a), SFKD (Yuan et al., 2024b) |
| | Decoupling | Splits the logit distillation loss into separate terms | DKL (Cui et al., 2024), BPKD (Liu et al., 2024d), SDD (Wei et al., 2024b), Ding et al. (Ding et al., 2024), NTCE (Li et al., 2024a) |
| | Revisiting Logit Distillation | Reduces the gap between teacher and student | TAKD (Mirzadeh et al., 2020), VanillaKD (Hao et al., 2023), EffDstl (Song et al., 2023b), Xu et al. (Xu et al., 2023d), BDCKD (Lu et al., 2025) |
| Similarity | Instance-level Similarity | Pairwise similarity between different data samples | MTKD (You et al., 2017), RKD (Park et al., 2019), SPKD (Tung & Mori, 2019), IRGKD (Liu et al., 2019f), Chen et al. (Chen et al., 2020b), CSKD (Chen et al., 2020f), Wang et al. (Wang et al., 2024g), Cheng et al. (Cheng et al., 2024a), GIRKD (Hu et al., 2024b) |
| | Feature/Channel-level Similarity | Pairwise similarity between features/channels | Yim et al. (Yim et al., 2017), ICKD (Liu et al., 2021c), CCD (Wang et al., 2023d) |
| | Class-level Similarity | Pairwise similarity between classes | CSKD (Chen et al., 2020f), DSD (Feng et al., 2021), DistKD (Huang et al., 2022a), IDD (Zhang et al., 2022d), AICSD (Mansourian et al., 2025), Dist+ (Huang et al., 2025a), IKD (Wang et al., 2025a) |
| | Others | Similarity between regions/ pixels/ neighbors/ views | SKD (Liu et al., 2019d), CIRKD (Yang et al., 2022c), PRRD (Wang et al., 2023b), BCKD (Wang et al., 2023k), NRKD (Xin et al., 2024), CRLD (Zhang et al., 2024g) |

## 2.2 Feature-based Distillation

DNNs are proven to extract different levels of features in their layers, and these multi-level representations from the intermediate layers of a network make them a valuable source for distillation, known as feature distillation. These intermediate representations provide step-by-step information that leads to the final prediction, which can be more valuable, especially in tasks like representation learning where the output of the model is a representation rather than logits.

The concept of feature distillation was first explored by Romero et al. (2014), where they proposed providing the student network with hints from the teacher. More specifically, they suggested minimizing the differences in features between the teacher and student to align them. Generally, let $F_s$ and $F_t$ be intermediate features of the teacher and student, respectively. A general feature distillation loss ($L_{fd}$) between teacher and student can then be defined as follows

$$L_{fd} = \ell_{feature}(\Phi_t(F_t), \Phi_s(F_s)) \tag{4}$$

where $\ell_{feature}$ is a similarity function for aligning the features, and $\Phi$ is a transformation function that matches the spatial size or number of channels between the features of the teacher and the student.

Inspired by the idea of FitNet, several feature-based distillation methods have been proposed. Initially, AT(Zagoruyko & Komodakis, 2016) introduced attention transfer by minimizing the $L_2$ norm of the difference between the feature maps of the teacher and student. Huang & Wang (2017) focused on matching the distributions of neuron selectivity patterns between the teacher and student by minimizing the Maximum Mean Discrepancy (MMD). Heo et al. (2019b) transferred the activation boundaries formed by hidden neurons, while Kim et al. (2018) transferred the factors of the features. Heo et al. (2019a) presented a comprehensive overhaul of feature distillation, proposing to distill the pre-activation feature maps, and Shu et al. (2021) introduced channel-wise distillation by applying softmax to channels of features and matching the distributions between teacher and student.

Subsequently, some methods explored cross-layer feature distillation. SimKD (Chen et al., 2022a) employed a reused teacher classifier. Chen et al. (2021d) introduced Knowledge Review, where the feature maps of each layer of the teacher should match the student's feature maps in the corresponding layer and all features from the previous layers. Chen et al. (2021a) proposed a more general framework where each feature from the student should match all of the teacher's features, learning weights to determine the connection between the feature maps. This framework is a generalized version of FitNet, where the weights for corresponding features are set to one.

Recently, the focus of many papers has shifted towards the transformation function of the features, highlighting its role in feature matching beyond just matching feature sizes. Heo et al. (2019b) trains an autoencoder to transform the student's features for better alignment with the teacher's features. Yang et al. (2022f) randomly masks some pixels of the student's feature map and uses two convolution layers to transform the masked features, aligning them with the corresponding features in the teacher's model to encourage the student to produce feature maps similar to those of the teacher. Liu et al. (2023c) demonstrated that masking is not necessary and a simple Multi-Layer Perceptron (MLP) layer for transforming the student's features is sufficient.

Most recently, Huang et al. (2023b) combined the idea of diffusion (Ho et al., 2020) with KD, training a diffusion model on the teacher's feature maps and using it to denoise the student's feature maps. It considers the student's features as a noisy version of the teacher's features and employs diffusion as a transformation for the student's features. In a concurrent work, Mansourian et al. (2024) used Convolutional Block Attention Mechanism (CBAM) (Woo et al., 2018) as a transformation, applying channel and spatial attention to the teacher's and student's features to highlight the most discriminative parts. It defines an MSE loss between the refined feature maps of the teacher and student.

Concurrent with the research path outlined, several other methods have been proposed. Liu et al. (2024g) rethinks raw feature alignment and decomposes the loss function of FitNets (Romero et al., 2014) into a magnitude difference term and an angular difference term. Yuan et al. (2024a) matches the features of the teacher with augmented versions of the student's features, while Zhang et al. (2024i) distills features in the frequency domain. Liu et al. (2023a) proposes N-to-one matching between the features of the student and teacher, respectively, and Passban et al. (2021) suggests that distilling corresponding features results in the disregard of some layers of the teacher. Instead, they propose to fuse all the features of the teacher for distillation.

In summary, feature-based distillation methods are more generalizable as the student can access information from the intermediate layers of the teacher, providing richer knowledge. While most state-of-the-art methods are feature-based and lead to improved performance, selecting the appropriate layers for distillation, aligning corresponding features from the teacher and student, and matching feature sizes pose significant challenges in feature distillation. This is especially true in cases where the teacher and student have different architectures, a common scenario with the recent growth of foundation models. In such cases, there can be a significant

decrease in performance due to the capacity gap between the teacher and student networks. Furthermore, comparing different aspects of feature-based methods is not straightforward, as their performance can vary significantly depending on factors like the selection of layers, feature alignment, and feature transformation components. The sensitivity of these methods to such factors complicates their comparison and evaluation.

Table 3: Summary of feature distillation methods.

| Method | Teacher Transformation | Student Transformation | Knowledge Type | Distillation Loss |
|---|---|---|---|---|
| FitNet (Romero et al., 2014) | – | $1 \times 1$ Conv | Feature Representation | $\mathcal{L}_2(\cdot)$ |
| AT (Zagoruyko & Komodakis, 2016) | Channel Aggregation | Channel Aggregation | Attention Map | $\mathcal{L}_2(\cdot)$ |
| NST (Huang & Wang, 2017) | – | $1 \times 1$ Conv | Neuron Selectivity Pattern | $\mathcal{L}_{MMD}(\cdot)$ |
| Jacobian (Srinivas & Fleuret, 2018) | Gradient | Gradient | Jacobian of Feature | $\mathcal{L}_2(\cdot)$ |
| FT (Kim et al., 2018) | Auto-encoder | Auto-encoder | Paraphraser | $\mathcal{L}_1(\cdot)$ |
| AB (Heo et al., 2019b) | Binarization | $1 \times 1$ Conv | Activation Boundary | Marginal $\mathcal{L}_2$ |
| Heo et al. (Heo et al., 2019a) | Margin ReLU | $1 \times 1$ Conv | Pre-ReLU Feature | Partial $\mathcal{L}_2$ |
| He el al. (He et al., 2019) | Auto-encoder | $3 \times 3$ Conv | Adapted Feature | $\mathcal{L}_1(\cdot)/\mathcal{L}_2(\cdot)$ |
| FN (Xu et al., 2020b) | $L_2$ Normalization | $L_2$ Normalization | Feature Representation | $\mathcal{L}_{CE}(\cdot)$ |
| CWD (Shu et al., 2021) | Softmax($\cdot$) | Softmax($\cdot$) | Activation Map | $KL(\cdot)$ |
| MGD (Yang et al., 2022f) | – | Masking + Conv | Feature Representation | $\mathcal{L}_2(\cdot)$ |
| MLP (Liu et al., 2023c) | – | MLP | Feature Representation | $\mathcal{L}_2(\cdot)$ |
| DiffKD (Huang et al., 2023b) | – | Diffusion Process | Denoised Feature | $KL(\cdot)/\mathcal{L}_2(\cdot)$ |
| FAKD (Yuan et al., 2024a) | – | Feature Augmentation | Activation Boundary | $KL(\cdot)$ |
| LAD (Liu et al., 2024g) | – | $1 \times 1$ Conv | Angular Information of Feature | $\mathcal{L}_2(\cdot)$ |
| AttnFD (Mansourian et al., 2024) | Channel/Spatial Attention | Channel/Spatial Attention | Refined Feature | $\mathcal{L}_2(\cdot)$ |

## 2.3 Similarity-based Distillation

In addition to logit and feature distillation, similarity distillation is another form of distillation that equips the student with the structural knowledge and relationships derived from the data learned by the teacher. Rather than relying solely on exact predictions or feature values, similarity distillation transfers a higher order of knowledge by considering the pairwise similarities between features or instances.

One of the pioneering works was Yim et al. (2017), which introduced the Flow of Solution Procedure (FSP). This method involved computing the inner product between features from two layers in the teacher network and defining the squared L2 norm as the cost function for each pair of layers between the teacher and student networks. In a related work, You et al. (2017) suggested distilling the relative dissimilarity between intermediate representations of different examples. Subsequent studies proposed distilling the 8-neighborhood similarity of logits (Xie et al., 2018) and distilling similarity of features and logits among multiple teachers (Zhang & Peng, 2018).

In general, the similarity distillation loss ($L_{SD}$) is defined between a relation in the teacher and student as follows

$$L_{SD} = \ell_{similarity}(\phi(F_t^i, F_t^j), \phi(F_s^i, F_s^j)) \tag{5}$$

where, $F_t$ and $F_s$ represent the features/logits of the teacher and student, respectively, and $F^i$ and $F^j$ denote different features/logits from either two different layers or two different samples. The function $\phi$ calculates the similarity knowledge between the features/logits of the teacher and student, and $\ell_{similarity}$ is a correlation function that measures the similarities between the teacher and student.

Recently, similarity distillation has garnered increased attention, with numerous methods proposed, each focusing on different levels of similarity such as instance/sample-level similarity (Park et al., 2019; Tung & Mori, 2019; Liu et al., 2019f; Peng et al., 2019; Chen et al., 2020b; Cheng et al., 2024a; Hu et al., 2024b; Wang et al., 2024g; Xin et al., 2024), feature/channel-level similarity (Wang et al., 2020d; Liu et al., 2021c; Wang et al., 2023d), and class-level similarity (Chen et al., 2020f; Feng et al., 2021; Zhang et al., 2022d; Huang et al., 2022a; Mansourian et al., 2025; Huang et al., 2025a; Wang et al., 2025a). Remarkable works have emerged in instance-level methods. For example, RelationKD (Park et al., 2019) distills the distance and anglewise correlations between features of different samples. Tung & Mori (2019) adopts a similar approach by distilling the correlation of samples within a batch through the inner-product calculation of features of each sample. Liu et al. (2019f) leverages an instance relationship graph where features are represented as nodes and relations as edges for each sample. CIRKD (Yang et al., 2022c), a popular method, extends instance similarity from the batch-level by utilizing a memory bank to consider global relations between samples. Hu et al. (2024b) conducts similar work by distilling pairwise similarity relations based on stored features to reveal more complete instance relations.

In feature/channel-level similarity, IFVD (Wang et al., 2020d) and ICKD (Liu et al., 2021c) are two pioneering works. The former defines a prototype for each class and distills the distances of pixels from each prototype, while the latter creates a channel correlation matrix using the dot-product of a feature map with its transpose for channel-level similarity distillation. Wang et al. (2023d) compels the student to imitate the teacher by minimizing the distance between the channel correlation maps of the student and the teacher.

In class-level similarity, DistKD (Huang et al., 2022a; 2025a) is a popular method that significantly enhances the student's performance by distilling inter- and intra-class similarities. Feng et al. (2021) distills the pairwise similarity of classes by multiplying the logits with its reshaped version. Zhang et al. (2022d) suggests distilling position information and inter-class distances for semantic segmentation. AICSD (Mansourian et al., 2025) distills inter-class similarities by introducing intra-class distributions for each class and subsequently calculating pairwise similarity of these distributions using KL for semantic segmentation. IKD (Wang et al., 2025a) facilitates knowledge sharing within the same class to ensure consistent predictions.

In addition to the similarity levels mentioned, spatial similarity is also utilized for distilling pixel-to-pixel or pixel-to-region (Liu et al., 2019d; Yang et al., 2022c; Wang et al., 2023a;b;k; Zhang et al., 2024g) similarities. Liu et al. (2019d) was among the first to propose pooling features into nine regions and distilling the similarity of these regions. CIRKD (Yang et al., 2022c) focused on pixel-to-pixel and pixel-to-region similarities across entire images. Wang et al. (2023b) transfers the multi-scale pixel-region relation, Wang et al. (2023k) distills correlations between adjacent blocks of the teacher, Wang et al. (2023a) employs patch-level similarity for distillation, and Zhang et al. (2024g) introduces distillation of the logits' similarity from local and global perspectives.

## 3 Schemes

After the knowledge source, the distillation scheme is one of the key choices in a teacher-student architecture. Based on the architecture of the teacher and its training mode, distillation methods can be categorized into three main schemes: offline distillation, online distillation, and self-distillation. In the offline scheme, which includes most of the distillation methods, the teacher is a larger pre-trained network. In the online scheme, the teacher and student are trained simultaneously, and in self-distillation, the teacher and student are the same network. Each of these schemes has its own applications: offline is preferred when a larger pre-trained teacher is available, online is most suitable when access to a pre-trained model is not available, and self-distillation plays an important role in self-supervised scenarios where labeled data is not available. Figure 4 shows the overall diagram of each scheme. In the following sections, the details of each scheme are explained.

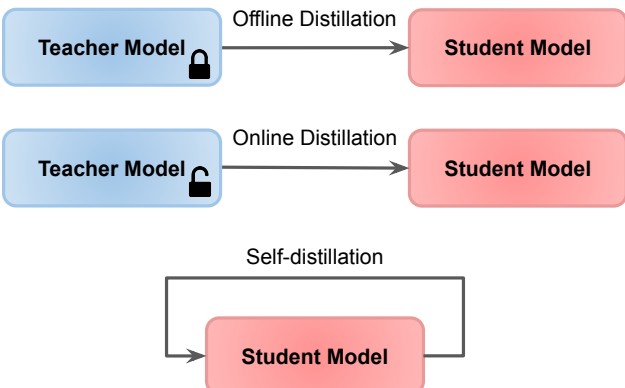

Figure 4: Illustration of distillation schemes. The teacher model in offline distillation is pre-trained and frozen, while in online distillation, the teacher and student are trained simultaneously. In self-distillation, the teacher and student are the same network.

### 3.1 Offline Distillation

The most straightforward approach to knowledge distillation is offline distillation, which consists of two stages. First, a large teacher model is pre-trained on a dataset. Then, during the training of the student model, the teacher's knowledge is transferred while its weights remain frozen.

Most existing distillation methods follow this offline approach, and the majority of the studies mentioned in the previous section adopt this technique. Offline distillation was first introduced in Hinton (2015), and numerous methods have since been proposed to extend and improve it. However, offline distillation has some limitations. One major challenge is the memory and computational overhead required to load a large teacher model, which can be a significant issue in resource-constrained environments. Additionally, since the teacher model remains fixed, the student model's performance is inherently constrained by the teacher's capabilities. Any biases present in the teacher can also be transferred to the student, potentially affecting its generalization (Guo et al., 2020a; Huang et al., 2022a).

To address these issues, incorporating adaptive distillation techniques alongside offline distillation can help improve student performance (Cho & Hariharan, 2019; Shu et al., 2021; Mansourian et al., 2025). Moreover, in LLM distillation, where the teacher model is proprietary and only accessible via an API, offline distillation remains the only practical approach. In summary, despite its challenges, offline distillation continues to be the most widely used method in knowledge distillation due to its simplicity and effectiveness.

### 3.2 Online Distillation

Unlike offline distillation, the training of online distillation occurs in a single stage (Chen et al., 2020a). Wang et al. (2024a) showed that online distillation offers advantages by using a group of students as a virtual teacher, eliminating the need for a large pretrained model. Early approaches, such as Guo et al. (2020b) and Zhu et al. (2018), utilized an ensemble of multi-branch logits to create an on-the-fly teacher and trained the student accordingly. In addition, Wu & Gong (2021) extended this approach by using the mean of previous predictions as an auxiliary guide to enhance learning. Li et al. (2021b) introduced variation in branch depth, with each Hourglass network mimicking both the final heatmap and the fused heatmap of all branches. Yang et al. (2023b) proposed a framework for mutual contrastive learning, where contrastive embeddings from the same or different networks were used with the anchor during the training phase.

Recent works have focused on improving other aspects of online KD. Rao et al. (2023) explores methods to enhance KD by reducing the sharpness and variance gap in the logit space between the teacher and student. Zhang et al. (2023h) aims to achieve flatter loss minima to improve the generalization and stability of student

models. Jacob et al. (2023) enhances multi-task networks by leveraging single-task networks as teachers and employing adaptive feature distillation with online task weighting. Similarly, Ypsilantis et al. (2024) applies this approach to multi-domain tasks using multiple teachers, where each teacher model specializes in a specific domain.

### 3.3 Self-distillation

Self-distillation is a unique form of online distillation where there is no larger pre-trained teacher; instead, the teacher and student are identical networks (Furlanello et al., 2018; Zhang et al., 2019b; Lan et al., 2019; Xu & Liu, 2019; Mobahi et al., 2020; Zhang et al., 2021c; Kim et al., 2021a). The concept was initially introduced in Furlanello et al. (2018), where a teacher is initially trained on labels, and subsequently, a new identical model is initialized from a different random seed and trained under the guidance of the former teacher. In a similar approach, Lan et al. (2019) first trains the network with labels and then utilizes labels and features of the model trained so far for subsequent training steps. Kim et al. (2021a) incorporates a linear combination of hard targets and predictions from the model in the previous epoch (teacher) to train the network in the current epoch (student). Zhang et al. (2019b) partitions the model into several sections and transfers knowledge from deeper sections to shallower ones, while Zhang et al. (2021c) employs multiple shallow classifiers at different layers and transfers knowledge from the deepest classifier to shallower ones. Mobahi et al. (2020) is the first to provide a theoretical analysis of self-distillation, and Zhang & Sabuncu (2020) establishes a theoretical link between self-distillation and label smoothing.

Recently, a variety of diverse self-distillation algorithms have been proposed (Yang et al., 2023g; Zhang et al., 2023b; Xiao et al., 2023b; Zheng et al., 2024b; Wu et al., 2024a; Chen et al., 2024a; Lin et al., 2024b; Li et al., 2024i; Liu et al., 2024e; Liang et al., 2024; Qin et al., 2025; Wang et al., 2025c), each offering a new perspective on the problem. Some methods leverage the deeper layers of the model as teachers to distill knowledge to shallower parts of the model (Liang et al., 2024; Qin et al., 2025). Other approaches utilize predictions from previous epochs as teachers to supervise the model in subsequent epochs (Liu et al., 2024e; Li et al., 2024i; Lin et al., 2024b). Chen et al. (2024a) and Zheng et al. (2024b) employ iterative pruning and discarding of parts of the model to create a student for self-distillation and MSKD(Wang et al., 2025c) proposes a self-distillation method for multi-label learning. Wu et al. (2024a) and Zhang et al. (2023b) integrate the concept of self-distillation with graphs, while Xiao et al. (2023b) combines data augmentation, deep contrastive learning, and self-distillation. In this method, different views of the same sample are input to the model for enhanced learning.

## 4 Algorithms

Aside from the source and scheme, different algorithms have been proposed to effectively transfer knowledge from the teacher to the student. In this section, the most important algorithms are reviewed. These include attention-based distillation, adversarial distillation, multi-teacher distillation, cross-modal distillation, graph-based distillation, adaptive distillation, and contrastive distillation.

### 4.1 Attention-based Distillation

Attention-based distillation focuses on transferring rich information from the teacher to the student using attention mechanisms, as attention can reflect neuron selectivity. Attention maps highlight crucial regions of the features in the teacher model and enable the student to automatically and effectively focus on mimicking the important information from the teacher. Instead of exact prediction matching, this approach allows the student to concentrate on the more critical regions of the image, similar to the teacher model.

The first work exploring attention distillation was Attention Transfer (AT) (Zagoruyko & Komodakis, 2016), which proposed the aggregation of feature channels in the student and teacher models. By minimizing the loss between attention maps of the teacher and student, the attention of the teacher can be effectively transferred to the student. Feng et al. (2021) transfers residual attention maps of features from two different layers, while An et al. (2022) employs a self-attention module to adaptively aggregate context information

from the student and teacher feature maps. Yuan et al. (2024b) uses self-attention to simplify the learning process by softening the logits before distillation.

Several methods have focused on utilizing Class Activation Maps (CAM) (Zhou et al., 2016) or Gradient-Weighted Class Activation Maps (GradCAM) (Selvaraju et al., 2017) for attention distillation by transferring regions with more attention (Bavandpour & Kasaei, 2020; Cho & Kang, 2022; Guo et al., 2023d). Karine et al. (2024) applies channel self-attention and position self-attention on the features for distillation, Gou et al. (2023) employs different types of attention to transfer knowledge at various levels of deep representation learning for KD, and Guo et al. (2024b) computes attention using student features as queries and teacher features as key values, implementing sparse attention values through random deactivation. These sparse attention values are then used to reweight the feature distance of each teacher-student feature pair to prevent negative transfer.

Recently, powerful attention distillation methods have been proposed (Yang et al., 2023d; Huang et al., 2023b; Mansourian et al., 2024). DiffKD (Huang et al., 2023b) suggests training a diffusion model on the teacher's features and using it to denoise the student's features to highlight important areas for distillation. AttnFD (Mansourian et al., 2024) investigates the role of attention in distillation for semantic segmentation and employs CBAM (Woo et al., 2018) attention mechanism to refine features before distillation by considering both channel and spatial attention. This refinement focuses on transferring crucial information and emphasizes important regions, particularly enhancing the student network's performance in scenarios involving small objects.

## 4.2 Adversarial Distillation

A promising direction for leveraging teacher information during distillation is to embed the process within an adversarial framework inspired by the min-max game of Generative Adversarial Networks (GANs) (Goodfellow et al., 2014). Three distinct approaches emerge in this context. First, generative networks can be employed to synthesize the teacher's training data when the original data is unavailable. Second, a discriminator is integrated to provide refined guidance by distinguishing between teacher and student outputs during knowledge transfer. Third, adversarial training is applied to minimize discrepancies between teacher and student models when confronted with perturbed samples, thus pushing the student to learn from more challenging examples. Collectively, these methods aim to enhance the performance and robustness of the distilled student model, and the following sections will elaborate on these three approaches in detail.

### 4.2.1 Adversarial Data-Free Knowledge Distillation

Adversarial distillation is applied in various ways, depending on the goal of the distillation process. One approach involves data-free distillation (Chen et al., 2019a; Ye et al., 2020; Fang et al., 2019; Choi et al., 2020; Liao et al., 2024; Patel et al., 2023; Wang et al., 2024o; Do et al., 2022; Zhao et al., 2022b; Zhou et al., 2024d; Liu et al., 2018; Micaelli & Storkey, 2019; Zhang et al., 2022b), where adversarial examples are generated in the absence of access to the teacher's original training data, often due to privacy concerns, transmission issues, or legal constraints (Fang et al., 2019). Adversarial samples are synthesized either to mimic the original dataset or to augment it, thereby enhancing the distillation process. Methods in this category (Chen et al., 2019a; Choi et al., 2020; Wang et al., 2024o; Ye et al., 2020; Zhao et al., 2022b; Micaelli & Storkey, 2019) often leverage the pre-trained teacher network and its internal statistics as a discriminator to guide the generator. Other techniques address distribution shifts in generated samples by proposing the use of an Exponential Moving Average (EMA) of the student as a more reliable learning source (Do et al., 2022) or framing the process as a meta-learning problem that balances knowledge acquisition and retention (Patel et al., 2023). Self-adversarial strategies (Zhou et al., 2024d) have also been explored to extend robustness in data-free settings. Lastly, Liao et al. (2024) investigated the challenges of data-free distillation with imbalanced datasets, where the teacher model is pre-trained on skewed class distributions, introducing class-aware strategies to mitigate this imbalance.

### 4.2.2 Discriminator in the Loop

The use of a discriminator has proven effective in better aligning the prediction distributions between the teacher and student networks. Leveraging this intuition, one of the prevalent approaches in adversarial distillation involves incorporating a discriminator network to align the teacher and student logits (Xu et al., 2017; Liu et al., 2019a; Ghorbani et al., 2020; Croitoru et al., 2024) or intermediate feature representations (Liu et al., 2020a; Wang et al., 2020a; Chung et al., 2020; Wang et al., 2020d; Zhang et al., 2020b; Li et al., 2020b; Liu et al., 2019d; Shen et al., 2019d; Belagiannis et al., 2018; Wang et al., 2018c). Additionally, some methods employ a three-player framework for discrimination (Wang et al., 2019; 2018b), enhancing the effectiveness of the distillation process.

Adversarial distillation has also been extended beyond standard networks to the compression of GANs (Aguinaldo et al., 2019; Wang et al., 2020c; Li et al., 2020a; Zhang et al., 2022a), facilitating the distillation of heavy generator networks into more efficient ones. Notably, the discrimination module is not limited to the GAN framework and has been adapted for diffusion models to reduce the number of inference steps, significantly improving their inference efficiency (Sauer et al., 2024; Liu et al., 2024a; Kong et al., 2024; Lin et al., 2024c). Furthermore, advanced techniques have been introduced to utilize the score function instead of the final outputs or logits for distillation, leading to approaches such as adversarial score distillation (Wei et al., 2024a) and variational score distillation (Wan et al., 2024). These methods employ score matching and variational approximations to achieve improved robustness and efficiency in knowledge transfer.

### 4.2.3 Adversarial Robust Knowledge Distillation

As previously discussed, KD transfers knowledge from a large, pre-trained teacher network to a smaller student network, yet it does not inherently guarantee robustness against adversarial attacks. Adversarial Robust Distillation (ARD) (Goldblum et al., 2020) addresses this limitation by combining adversarial training with distillation. In ARD, the objective is to minimize the discrepancy between the student's predictions and the teacher's predictions under adversarial perturbations, defined as follows

$$\min_{\theta} \mathbb{E}_{(X,y)\sim\mathcal{D}} \Big[ \underbrace{\alpha\tau^2 D_{\mathrm{KL}}(S_\theta^\tau(X+\delta_\theta), T^\tau(X))}_{\text{Adversarially Robust Distillation loss}}$$
$$+ \underbrace{(1-\alpha)\ell(S_\theta^t(X), y)}_{\text{Classification loss}} \Big], \tag{6}$$

where $S$ and $T$ denote the student and teacher networks, $\tau$ is the temperature constant, and $\delta_\theta$ (computed as $\delta_\theta = \arg\max_{\|\delta\|_p < \epsilon} \ell(S_\theta^t(X), y)$) represents the adversarial perturbation.

Building upon ARD, several approaches have been developed to enhance the transfer of robust knowledge. One group refines teacher prediction reliability: RSLAD (Zi et al., 2021) replaces the classification loss in eq. (6) with a KL loss between teacher and student predictions to capitalize on robust soft labels from adversarially trained teachers, while introspective adversarial distillation (IAD)(Zhu et al., 2021a) and MTARD(Zhao et al., 2022e) further allow selective trust in teacher outputs and employ dual teachers for clean and adversarial scenarios, respectively. Another group enhances alignment and adaptability by synchronizing gradients and adaptively deriving perturbations through IGDM (Lee et al., 2023a) and AdaAD (Huang et al., 2023a); additional refinements are achieved in SmaraAD (Yin et al., 2024b) and IGKD-BML (Wang et al., 2021b) via Spearman Correlation, Class Activation Mapping, and attention-guided distillation to better emulate the teacher's decision-making process.

Advanced strategies tackle limitations in robustness transfer by introducing novel training schemes. DAR-WIN (Dong et al., 2024b) incorporates intermediate adversarial samples along with a triplet-based loss to balance natural, adversarial, and intermediate distributions, while PeerAiD (Jung et al., 2024) trains a peer model alongside the student for dynamic adaptation. Furthermore, DGAD (Park & Min, 2024) partitions the dataset into standard and adversarial distillation groups with consistency regularization to manage data

imbalance, and both STARSHIP (Dong et al., 2024a) and TALD (Nguyen-Duc et al., 2023) further bolster robustness by transferring statistical attributes and sampling diverse adversarial examples via a Teacher Adversarial Local Distribution using Stein Variational Gradient Descent. These grouped techniques collectively advance the robustness and overall performance of student models in adversarial environments.

## 4.3 Multi-teacher Distillation

In contrast to typical distillation scenarios where the student is trained using a single teacher, multi-teacher algorithms aim to train the student network by amalgamating knowledge from multiple teachers. This approach has the potential to enhance the student's performance by leveraging an ensemble of teachers with diverse knowledge, thereby bolstering the student's generalization capabilities. This concept was initially proposed in Hinton (2015), which utilized the average of logits from multiple teachers as softened labels. Subsequent works have explored variations on this theme: You et al. (2017) employed a voting strategy, while Sau & Balasubramanian (2016) recommended perturbing the teacher's logits by adding noise, treating these perturbed outputs as distinct teachers that can act as a regularizer. TeKAP (Hossain et al., 2025) proposed an augmentation technique that generates multiple synthetic teacher knowledge by perturbing the knowledge of a single pretrained teacher. Fukuda et al. (2017) utilized a pool of teachers, randomly selecting a teacher for distillation each time. Papernot et al. (2016a) trained multiple teachers, with each focusing on a subset of the dataset, and employed a voting system to distill the knowledge of each expert teacher. Iordache et al. (2025) distilled knowledge from multiple teachers trained on distinct datasets. Furlanello et al. (2018) adopted a step-by-step strategy in which the student, at each stage, served as a teacher for the student in the subsequent step.

In recent years, a variety of multi-teacher methods have been introduced, each addressing distinct issues. Managing the knowledge aggregation adaptively poses a challenge due to the capacity gap between the student and each teacher (Li & Bilen, 2020; Wang et al., 2023e; Zhang et al., 2023a; Shang et al., 2023b; Lin et al., 2024d; Cheng et al., 2024b; Wang & Xu, 2024; Li et al., 2024e). Lin et al. (2024d) employs adaptive temperature and a diverse aggregation strategy to enhance distillation performance, while Cheng et al. (2024b) separates the vanilla KD loss and assigns adaptive weights to each teacher based on the entropy of their predictions. Wang & Xu (2024) leverages data relation knowledge to dynamically allocate weights to teachers, and Zhang et al. (2023a) employs a meta-network for weighting each teacher effectively.

Other methodologies have been proposed to effectively aggregate the knowledge of teachers for distillation (Luo et al., 2019; Chen et al., 2019b; Park & Kwak, 2020; Asif et al., 2020; Cao et al., 2023). Chen et al. (2019b) trains two distinct teachers, one for distilling intricate features and another for transferring general decision features. Park & Kwak (2020) and Asif et al. (2020) introduce additional branches to the student network for learning the features of teachers, while Cao et al. (2023) suggests a progressive approach for distilling knowledge from multiple teachers. Another significant research avenue in multi-teacher distillation involves using ensemble pre-trained teachers, referred to as Knowledge Amalgamation (KA) (Shen et al., 2019a;b; Vongkulbhisal et al., 2019; Luo et al., 2020; Amirkhani et al., 2021; Xu et al., 2022b; Zhang et al., 2023c; Li et al., 2024b). Through KA, publicly available trained networks can be repurposed in the context of multi-teacher distillation. Some approaches involve employing multiple teachers, each trained on different datasets (Dong et al., 2024b) or tasks (Shen et al., 2019a; Luo et al., 2019). Shen et al. (2019b) initially extracts task-specific knowledge from heterogeneous teachers sharing the same sub-task and then combines this extracted knowledge to construct the student network. Li et al. (2024b) proposes a KA framework based on uncertainty suppression, and Vongkulbhisal et al. (2019) suggests training a unified classifier from pre-trained classifiers from each source along with an unlabeled set of generic data.

In summary, multi-teacher distillation furnishes the student network with a broader range of diverse and enriched information from multiple teachers, ultimately enhancing the student's generalization capabilities. Nonetheless, the effective aggregation of teachers' knowledge, determining the individual contributions of each teacher in distillation, and managing the complexity and computational costs are the primary challenges associated with multi-teacher distillation methods.

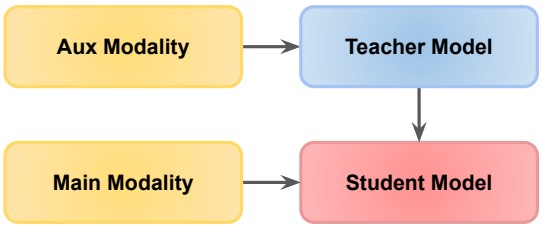

Figure 5: General overview of a cross-modal distillation method.

### 4.4 Cross-modal Distillation

In many scenarios, some modalities have rich annotations, while others lack sufficient labels. Cross-Modality Knowledge Distillation (CMKD) addresses this challenge by leveraging a well-annotated modality to improve the learning of a less-annotated one. This technique enables the model to benefit from the additional modality during training while remaining functional, even when that modality is unavailable at inference time. Gupta et al. (2016) introduced an innovative framework to transfer knowledge from a high-resource modality to a low-resource one, a method that has since been widely adopted in various studies. Figure 5 illustrates the overall concept of cross-modal distillation and Table 4 summarizes existing methods.

The early approach for CMKD was to use two parallel streams for two modalities during training, followed by a single stream during inference to produce feature maps similar to the teacher's modality.Garcia et al. (2018; 2019) applied this method to generate depth map from RGB. Later works focused on improving the imitation of the student's output to match the teacher's and reducing the gap between the modalities (Huo et al., 2024), with Thoker & Gall (2019) employing two student networks, each attempting to replicate the other's output. Wang et al. (2021d) adopted bidirectional distillation between events and frames.

The model proposed in Liu et al. (2021g) learned to generate 3D features from paired 2D inputs during the training phase and used them during inference. The structures in Hong et al. (2022b); Zhou et al. (2023b); Wang et al. (2023p); Klingner et al. (2023) consisted of separate encoders and decoders to extract features from LiDAR and RGB inputs, aiming to align the different features from the two modalities and distill spatial knowledge from LiDAR to RGB. Zhao et al. (2024b) used the same technique, replacing LiDAR with Radar, which provided lower spatial information, alongside RGB. Instead of using voxels for alignment, Zhang et al. (2024c) generated depth and semantic features from both networks and aimed to make them similar. Zhang et al. (2023i) employed a shared model to extract features from Thermal and RGB inputs in the student network, while using two heavier models for the teacher. The method proposed in Lee et al. (2023b) decomposed the process into task-specific components and enabled the distillation of knowledge from optical flow effects on the right part of the network.

To distill knowledge between text and image modalities, Andonian et al. (2022) extracted features separately and trained them on aligned instances, using soft alignment to supervise the student network. Wang et al. (2024j) applied masking techniques and used text features to guide the network in predicting masked images. Sarkar & Etemad (2024) masked audio and video frames, attempting to reconstruct them to achieve a good representation for both modalities and align the two features effectively.

### 4.5 Graph-based Distillation

Graph-based knowledge distillation extends traditional KD methods by incorporating higher-order relational information between features or instances. Unlike approaches that primarily rely on pairwise similarity, graph-based methods utilize graph structures as a generic framework to capture intra-dependent relationships. In such methods, features or instances are represented as vertices, while the dependencies between them are modeled through edge weights. These graph structures encapsulate both local and global relationships, enabling the transfer of richer and more structured knowledge from teacher models to student

Table 4: Summary of Cross-modal Distillation Methods.

| Method | Train Modality | Inference Modality |
|---|---|---|
| Gupta et al.(Gupta et al., 2016) | RGB | Depth/Optical Flow |
| Garcia et al.(Garcia et al., 2019) | RGB + Depth | RGB |
| EvDistill(Wang et al., 2021d) | RGB + Event | Event |
| Hong et al.(Hong et al., 2022b) DistillBEV(Wang et al., 2023p) | RGB + LiDAR | RGB |
| UniDistill(Zhou et al., 2023b) | RGB + LiDAR | LiDAR |
| Lee et al.(Lee et al., 2023b) | RGB + Optical Flow | RGB |
| Radocc(Zhang et al., 2024c) | Multi-view + LiDAR | Mult-view |
| CRKD(Zhao et al., 2024b) | Multi-view + LiDAR | Multi-view + Radar |
| Andonian et al. (Andonian et al., 2022) CM-MaskSD (Wang et al., 2024j) | RGB + Text | RGB + Text |
| XKD(Sarkar & Etemad, 2024) | Video | RGB |

models (Zhang & Peng, 2018; Chen et al., 2020b; Park et al., 2019; Tung & Mori, 2019; Peng et al., 2019; Liu et al., 2019f; Lee & Song, 2019; Passalis et al., 2020; Zhou et al., 2021a; Chen et al., 2021e; Ghorbani et al., 2021; Lee & Song, 2021). Furthermore, graph structures can be effectively employed to model the flow of information during distillation, particularly in scenarios involving multiple teacher models, by regulating the knowledge transfer process (Luo et al., 2018; Minami et al., 2020).

In recent years, the advent of Graph Neural Networks (GNNs) has garnered significant attention, particularly in the fields of data mining and knowledge graph modeling. GNN distillation is primarily utilized to achieve two main objectives (Tian et al., 2023a): enhancing the performance of the original teacher model (performance improvement) (Zhang et al., 2020a; 2021e; Chen et al., 2020e; Rezayi et al., 2021; Guo et al., 2023c; Huo et al., 2023; Zhu et al., 2024b; Yu et al., 2022a; Guo et al., 2022) or creating a lightweight version of the GNN (compression). The latter involves distilling a larger and more complex GNN into a compact and efficient GNN (Dong et al., 2023c; Yang et al., 2022b; He et al., 2022; Deng & Zhang, 2021) or even an MLP (Yang et al., 2021; Zhang et al., 2021d; Zheng et al., 2021; Tian et al., 2022a), making the distilled model suitable for real-time applications. To accomplish these goals, the information transferred between the teacher and the student typically includes logits (Guo et al., 2023c; Huo et al., 2023; Zhang et al., 2021e; 2020a; 2021d; Yang et al., 2021; Tian et al., 2022a; Dong et al., 2023c; He et al., 2022; Deng & Zhang, 2021), feature representations (Yu et al., 2022a; Huo et al., 2023; Rezayi et al., 2021; Zhang et al., 2020a; He et al., 2022), and the graph structure itself (Guo et al., 2022; Rezayi et al., 2021; Chen et al., 2020e; Zhang et al., 2021e; Zheng et al., 2021; Tian et al., 2022a; Yang et al., 2022b), which encapsulates the connectivity between the elements of the graph. These components collectively enable an effective transfer of structured knowledge, ensuring that the distilled model retains essential characteristics while meeting its respective performance or efficiency objectives.

## 4.6 Adaptive Distillation

Adaptive distillation has recently garnered increased attention. By dynamically adjusting the parameters of the distillation process instead of maintaining a constant setup, the effectiveness of knowledge transfer can be enhanced. Table 5 summarizes adaptive distillation methods in different categories.

Adaptive distillation can manifest in various levels and forms, such as adaptive loss definition (Park & Heo, 2020; Wang et al., 2024g; Ma et al., 2024), adaptive loss weighting (Cho & Hariharan, 2019; Zhou et al., 2020; Mansourian et al., 2025), adaptive teacher pruning (Mirzadeh et al., 2020; Sarridis et al., 2025; Lee et al., 2024; Passalis et al., 2020; Guo et al., 2024a), and dynamically weighting different aspects of distillation based on sample importance (Huang et al., 2022b; Tian et al., 2022c; Sun et al., 2023b; 2024b; Kim et al., 2024c; Gou et al., 2024; Huang et al., 2025b).

One primary concern is that the teacher network may produce incorrect outputs that should not be directly transferred to the student. To address this issue, some approaches have introduced adaptive losses. Park & Heo (2020) suggests an adaptive Cross Entropy loss that replaces incorrect probability maps with ground truth labels. Wang et al. (2024g) proposes a calibrated mask to prevent the teacher model's erroneous representations from interfering with the student model's training, while Ma et al. (2024) integrates the positive prediction distribution of two teacher networks based on their correctness and cross-entropy magnitudes to provide a more accurate output distribution for guiding the student network.

Moreover, certain studies have delved into the effects of altering the distillation loss throughout the distillation process. Cho & Hariharan (2019) was among the first to explore strategies where reducing the influence of the teacher network on the student can enhance performance. Zhou et al. (2020) indicates that the impact of the distillation loss should diminish as training progresses, with the student gradually assuming control of the training process towards the end. Mansourian et al. (2025) introduces a weight decaying process to merge similarity distillation with vanilla distillation.

Given the capacity disparity between teacher and student models, some works aim to narrow this gap. A notable example is TAKD (Mirzadeh et al., 2020), which employs a teaching assistant network as an intermediary teacher to facilitate better knowledge comprehension by the student model. Similarly, Passalis et al. (2020) utilizes an auxiliary teacher model before distillation, while Lee et al. (2024) and Guo et al. (2024a) advocate for channel and feature pruning, respectively, before distillation. Sarridis et al. (2025) implements filter pruning to reduce channels and applies a curriculum learning strategy to distill layers from easy to challenging levels.

Lastly, certain methodologies adaptively conduct distillation based on the significance of a region or sample. Tian et al. (2022c) extracts detailed context-specific information from each training sample, Sun et al. (2024b) adaptively identifies confusing samples, and Kim et al. (2024c) adjusts the temperature parameter based on each sample's energy level. Huang et al. (2022b) incorporates adaptive weights for distilling each mask in feature distillation, while Gou et al. (2024) employs ground truth information to mask crucial regions for logit distillation. Sun et al. (2023b) calculates channel divergences between the teacher and student networks, converting them into distillation weights, and Kang et al. (2024) assigns greater weight to losses in layers where training discrepancies between the teacher and student models are more pronounced during distillation. CALD (Huang et al., 2025b) leverages a meta-network to generate class-adaptive weights for distillation.

Table 5: Summary of adaptive distillation methods.

| Method | Reference |
|---|---|
| Adaptive Loss Definition | Park et al.(Park & Heo, 2020), Wang et al.(Wang et al., 2024g), CAG-DAKD (Ma et al., 2024), Dist+ (Huang et al., 2025a) |
| Adaptive Loss Scaling | Cho et al.(Cho & Hariharan, 2019), Zhou et al.(Zhou et al., 2020), AICSD(Mansourian et al., 2025) |
| Adaptive Teacher Pruning | TAKD(Mirzadeh et al., 2020), InDistill(Sarridis et al., 2025), Passalis et al.(Passalis et al., 2020), PCD(Lee et al., 2024), AFMPM(Guo et al., 2024a), ADG-KD(Li et al., 2025b) |
| Adaptive Sample Importance | MasKD(Huang et al., 2022b), APD(Tian et al., 2022c), HWD(Sun et al., 2023b), CS-KD (Sun et al., 2024b), Energy KD(Kim et al., 2024c), Gou et al.(Gou et al., 2024), CALD(Huang et al., 2025b) |

## 4.7 Contrastive Distillation

Contrastive learning seeks to learn representations by contrasting similar (positive) and dissimilar (negative) samples in the feature space (Hadsell et al., 2006). In KD, most research employs feature-based contrastive learning approaches, where anchor, positive, and negative sets are defined using features extracted by teacher and student models as shown in Figure 6. The contrastive distillation loss can be formulated as

$$\mathcal{L}_{\text{contrastive}} = -\log \frac{\exp(\text{sim}(F_i^s, F_i^t)/\tau)}{\sum_{j \in \mathcal{N}} \exp(\text{sim}(F_i^s, F_j)/\tau)}, \tag{7}$$

where anchors $(F_i^s)$ are typically features from the student's embeddings, positives $(F_i^t)$ are corresponding features from the teacher for the same input, and negatives $(F_j)$ are unrelated features from other data points. These methods employ contrastive loss to align the student's feature space with the teacher's, enabling efficient knowledge transfer. Other studies (Gao et al., 2021; Fan et al., 2023) innovate by introducing new contrastive loss functions that address challenges such as noisy negatives and capacity mismatches between teacher and student models, enhancing the robustness and effectiveness of the distillation process.

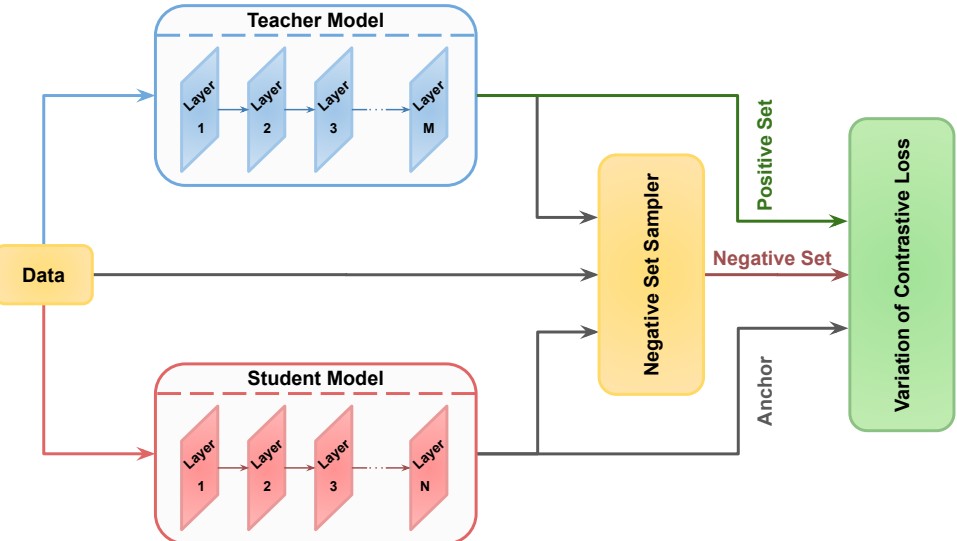

Figure 6: General overview of a contrastive distillation method. Embeddings of the teacher and student are considered as the positive set and anchor, respectively. The negative set can consist of embeddings from the teacher, student, or input data.

Contrastive distillation has gained significant attention for its ability to enhance knowledge transfer through contrastive learning techniques. (Tian et al., 2019) pioneered this approach by employing a contrastive loss to align the features of the teacher and student models, Expanding upon this foundation, WCoRD (Chen et al., 2021b) introduced a method that incorporates both primal and dual forms of the Wasserstein distance. While the dual form ensures global knowledge transfer by maximizing mutual information, the primal form focuses on local feature alignment within mini-batches, providing a balanced approach to feature alignment and distillation.

In the field of semantic segmentation, Fan et al. (2023) addresses the challenge of dense pixel-level representations by leveraging contrastive loss to align the teacher and student feature maps. Similarly, Liu et al. (2024b) focuses on the domain of single-image super-resolution. Their approach utilizes contrastive loss to distill statistical information from intermediate teacher features into a lightweight student network, enabling high-resolution image reconstruction with minimal computational resources.

The problem of distilling large language-image pretraining models, such as CLIP (Radford et al., 2021), is tackled by Chen et al. (2024g), which develops a novel framework incorporating image feature alignment and educational attention modules. This method effectively aligns multi-modal features through contrastive loss. To address challenges in sentence embedding, DistilCSE (Gao et al., 2021) proposes a method which compresses large contrastive sentence embedding models.

In the domain of medical imaging, CRCKD (Xing et al., 2021) combines class-guided contrastive distillation and categorical relation preserving techniques to enhance intra-class similarity and inter-class divergence. The use of class centroids and relational graphs ensures effective knowledge transfer, even in imbalanced datasets.

In the area of weakly supervised visual grounding, Wang et al. (2021e) uses pseudo labels generated by object detectors. By applying contrastive loss for region-phrase matching, their method eliminates the need for object detection during inference, simplifying the process. Zhu et al. (2021b) tackles the problem of relation-based knowledge transfer with CRCD. By maximizing mutual information between anchor-teacher and anchor-student relation distributions, their framework aligns inter-sample relations through a relation contrastive loss. Most recently, CRD (Zhu et al., 2025) has leveraged contrastive learning to align class-wise semantics by preserving sample-wise similarity for intra-sample alignment, while capturing structural semantics for inter-sample contrast.

In conclusion, the incorporation of contrastive learning into distillation demonstrates exceptional capabilities in feature alignment, relational knowledge preservation, and the integration of local and global information. Proficiency of contrastive distillation in addressing challenges such as multi-modal alignment, dense feature mapping, and efficient model compression underscores its versatility. Moreover, the inherent flexibility of contrastive learning-based distillation allows it to effectively address critical issues, including imbalanced datasets, noisy labels, and computational constraints.

## 5 Modalities

Although most distillation methods have been proposed for computer vision tasks and work with images, KD has been effectively applied to other modalities, such as 3D/multi-view data, text, speech, and video. This section provides a review of distillation methods for each modality, categorizing them based on their respective tasks.

### 5.1 3D Input

Knowledge distillation techniques contribute significantly to enhancing various 3D-related tasks, including object detection, semantic segmentation, shape generation, and shape classification. This section explores the application of KD in 3D data, highlighting innovative approaches and categorizing contributions based on task domains. Figure 7 illustrates the key tasks, domains, and types of 3D data where KD techniques have been successfully applied. Table 6 summarizes distillation methods for each task based on the source of distillation.

#### 5.1.1 3D Object Detection

3D object detection involves identifying and localizing objects within three-dimensional spaces using data such as point clouds, voxel grids, and images from multiple angles. KD enhances this process by transferring knowledge from more complex models, leveraging various approaches and modalities, to simpler ones. This approach improves accuracy and computational efficiency in tasks including autonomous driving and multi-camera detection. Recent advancements in KD-based 3D object detection have focused on three major areas: LiDAR-based, camera-only (single and multi-camera), and cross-modal, each addressing unique challenges in 3D perception.

**LiDAR:** Precise spatial and depth sensing are essential for autonomous systems, , yet challenges such as sparsity and occlusion often degrade performance in 3D object detection. To solve these, several KD frameworks have been proposed (Wei et al., 2022; Yang et al., 2022d; Zhang & Liu, 2023; Li et al., 2023g; Zhang et al., 2023d; Cho et al., 2023; Gambashidze et al., 2024; Bang et al., 2024; Huang et al., 2024e). X-Ray Distillation (Gambashidze et al., 2024) trains a teacher model on object-complete frames that are derived from aggregated LiDAR scans, to transfer knowledge to a student model that improves its ability to handle sparse data and occlusions. Similarly, RadarDistill (Bang et al., 2024) aligns radar and LiDAR features to handle sparsity and noise. Furthermore, RDD(Li et al., 2023g) reduces the mismatch between the teacher and student models by aligning feature representations and refining outputs.

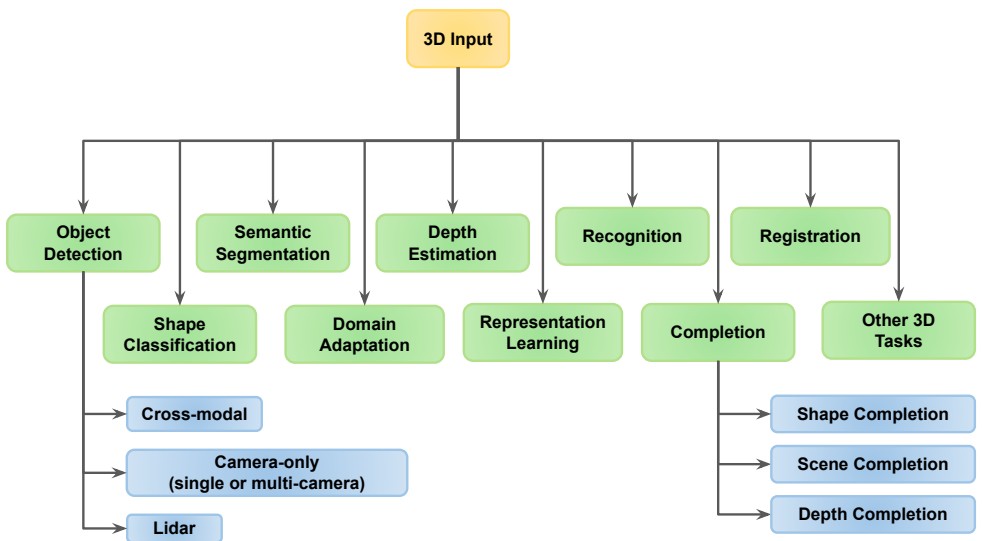

Figure 7: Overview of 3D domains and tasks.

In addition, PointDistiller (Zhang et al., 2023d) focuses on local geometric structures in point clouds, using dynamic graph convolution and reweighted learning to focus on important points and voxels. itKD (Cho et al., 2023) applies feature reduction and mutual refinement to transfer coarse and fine-grained geometric features. Additionally, SRKD (Huang et al., 2024e) bridges domain gaps caused by weather variations by aligning instance features through density and shape similarity and ensuring consistent predictions between the teacher and student models. LiDAR Distillation (Wei et al., 2022) also addresses domain gaps caused by varying LiDAR beam densities by performing distillation from a high-beam LiDAR teacher to a low-beam student.

Moreover, another study (Yang et al., 2022d) introduces two novel techniques: pivotal position logit KD, which focuses on key areas for enhanced distillation, and teacher-guided initialization, which transfers the teacher's feature extraction capabilities to the student through weight inheritance. However, intrinsic challenges such as sparsity, randomness, and varying density limit the effectiveness of normal distillation algorithms. To overcome these limitations, PVD (Zhang & Liu, 2023) leverages the advantages of both point-level and voxel-level knowledge to address these challenges in LiDAR object detection tasks.

**Camera-only (single or multi-camera):** Achieving accurate 3D object detection using only cameras is challenging due to the absence of depth information. To address this, FD3D (Zeng et al., 2023) uses selective masked generative distillation and query-based focal distillation to enhance object-specific learning in perspective and Bird's-Eye View (BEV) spaces. Similarly, X3KD (Klingner et al., 2023) integrates cross-task, cross-modal, and cross-stage distillation with techniques such as LiDAR feature alignment and adversarial training for dense supervision.

**Cross-modal:** Transferring spatial and depth knowledge from LiDAR-based models to cost-effective camera-radar or camera-only alternatives enhances multi-sensor systems, improving detection performance while resolving sensor inconsistencies. CRKD (Zhao et al., 2024b) transfers knowledge from LiDAR-camera teacher to a camera-radar student in the BEV space. Similarly, DistillBEV (Wang et al., 2023p) transfers 3D geometric knowledge from a LiDAR-based teacher to a multi-camera BEV student using region decomposition, adaptive scaling, spatial attention, and multi-scale distillation. In addition, CMKD (Hong et al., 2022b) transfers spatial knowledge from LiDAR-based teachers to monocular students, tackling depth estimation challenges using BEV features and teacher predictions. In addition UniDistill (Zhou et al., 2023b) proposes a universal KD framework which supports multiple modality pairs (LiDAR-to-camera, camera-to-

Table 6: Summary of 3D distillation methods based on their tasks and sources of distillation.

| Task | Feature-based | Similarity-based | Logit-based |
|---|---|---|---|
| Object Detection | FD3D (Zeng et al., 2023), X3KD (Klingner et al., 2023), Gambashidze et al. (Gambashidze et al., 2024), PointDistiller (Zhang et al., 2023d), itKD (Cho et al., 2023), CRKD (Zhao et al., 2024b), DistillBEV (Wang et al., 2023p), RDD (Li et al., 2023g), RadarDistill (Bang et al., 2024), SRKD (Huang et al., 2024e) | FD3D (Zeng et al., 2023), X3KD (Klingner et al., 2023), Gambashidze et al. (Gambashidze et al., 2024), itKD (Cho et al., 2023), CRKD (Zhao et al., 2024b) | X3KD (Klingner et al., 2023), Gambashidze et al. (Gambashidze et al., 2024), PointDistiller (Zhang et al., 2023d), CRKD (Zhao et al., 2024b), DistillBEV (Wang et al., 2023p), RDD (Li et al., 2023g) |
| Classification | FAD (Lee & Wu, 2023), HSD (Zhou et al., 2024a) | – | FAD (Lee & Wu, 2023), JGEKD (Zheng et al., 2024a), PointViG-Distil (Tian et al., 2023b) |
| Segmentation | Liu et al. (Liu et al., 2021g), Qiu et al. (Qiu et al., 2023), CMDFusion (Cen et al., 2023), Jiang et al. (Jiang et al., 2023a), Smaller3d (Adamyan & Harutyunyan, 2023), Seal (Liu et al., 2024h) | PVD (Hou et al., 2022), Qiu et al. (Qiu et al., 2023) | Genova et al. (Genova et al., 2021), PSD (Zhang et al., 2021f), PVD (Hou et al., 2022), Qiu et al. (Qiu et al., 2023), LGKD (Yang et al., 2023f), Smaller3d (Adamyan & Harutyunyan, 2023), PartDistill (Umam et al., 2024) |
| Depth Estimation | MVP-Net (Chen et al., 2024b), LiRCDepth (Sun et al., 2025) | LiRCDepth (Sun et al., 2025) | MVP-Net (Chen et al., 2024b), KD-MonoRec (Xiao et al., 2024), LiRCDepth (Sun et al., 2025) |
| Representation | PPKT(Liu et al., 2021d), Fu et al.(Fu et al., 2022), SLidR(Sautier et al., 2022), PointCMT(Yan et al., 2022), RECON(Qi et al., 2023), MVNet(Yan et al., 2023), LiDAR2Map(Wang et al., 2023m), C2P(Zhang et al., 2023k), HVDistill(Zhang et al., 2024f), Yao et al.(Yao et al., 2024), Text4Point(Huang et al., 2024c), Diff3F(Dutt et al., 2024) | – | Yu et al.(Yu et al., 2022b), LiDAR2Map(Wang et al., 2023m), I2P-MAE(Zhang et al., 2023g) |
| Domain Adaptation | Cardace et al. (Cardace et al., 2023), Wu et al. (Wu et al., 2023e), SEN (Li et al., 2025a) | – | SEN (Li et al., 2025a) |
| Recognition | DistilVPR (Wang et al., 2024h), PointMCD (Zhang et al., 2023f) | – | – |
| Completion | SCPNet (Xia et al., 2023), RaPD (Fan et al., 2022), Lin et al. (Lin et al., 2024a), VPNet (Wang et al., 2025b), Huang et al. (Huang et al., 2024a) | ADNet (Kim et al., 2024b) | Hwang et al. (Hwang et al., 2021), MonDi (Liu et al., 2022b), Zhou et al. (Zhou et al., 2024b), Zhang et al. (Zhang et al., 2024e), HASSC (Wang et al., 2024i), Huang et al. (Huang et al., 2024a) |
| Registration | Jiang et al. (Jiang et al., 2024a) | – | DiReg (Löwens et al., 2024) |
| Other 3D Tasks | DTC123 (Yi et al., 2024b), CSD (Decatur et al., 2024), Van et al. (Van Vo et al., 2024) | Van et al. (Van Vo et al., 2024) | PCDNet (Huang et al., 2022c) |

LiDAR, fusion-to-camera, and fusion-to-LiDAR). The model employed BEV as a shared representation to align teacher and student detectors across modalities.

### 5.1.2   3D Shape Classification

Point cloud classification is another critical task where the teacher-student framework can be applied. FAD (Lee & Wu, 2023) introduces a novel loss function that combines logit distillation and feature distillation. JGEKD (Tian et al., 2023b) proposes a loss function based on joint graph entropy to address the challenges of non-independent and identically distributed 3D point cloud data in classification tasks. To improve the efficiency Zheng et al. (2024a) introduced an offline distillation framework that incorporates a negative-weight self-distillation approach. Zhou et al. (2024a) employs a self-distillation framework for incomplete point cloud classification.

### 5.1.3   3D Semantic Segmentation

KD enhances the accuracy and robustness of point cloud segmentation by effectively transferring knowledge from a high-capacity teacher model to a lightweight student model (Genova et al., 2021; Liu et al., 2021g; Zhang et al., 2021f; Hou et al., 2022; Qiu et al., 2023; Yang et al., 2023f; Cen et al., 2023; Jiang et al., 2023a; Adamyan & Harutyunyan, 2023; Karine et al., 2024; Umam et al., 2024; Liu et al., 2024h).

In order to address challenges such as sparsity, randomness and varying density in segmentation, Hou et al. (2022) proposes point-to-voxel KD which distills the probabilistic outputs of the teacher at both the point level (fine-grained details) and voxel level (coarse structural information). Qiu et al. (2023) proposes a multi-to-single KD framework that fuses multi-scan information for hard classes and employs a multilevel distillation strategy, including feature, logit, and instance-aware similarity distillation.

Limited labeled data is another major challenge in the semantic segmentation of large-scale point clouds. Despite the task of annotating point-wise labels for such datasets being expensive and time-consuming,

existing methods often rely on fully supervised approaches that require dense annotations. PSD (Zhang et al., 2021f) addresses this issue by introducing perturbed branches and leveraging graph-based consistency in a self-distillation manner. Seal (Liu et al., 2024h) is a self-supervised learning framework that distills knowledge from off-the-shelf vision foundation models for point cloud segmentation. PartDistill (Umam et al., 2024) proposes a method to distill knowledge from a vision-language model to a 3D segmentation network.

### 5.1.4    3D Domain Adaptation

Unsupervised Domain Adaptation (UDA) struggles to perform effectively when the target domain differs from the labeled source domain, due to factors such as noise, occlusions, or missing elements. Cardace et al. (2023) proposes a self-distillation UDA technique to generate discriminative representations for the target domain and utilizes GNN-based online pseudo-label refinement. Wu et al. (2023e) introduces a cross-modal feature fusion in UDA for semantic segmentation. Li et al. (2025a) proposes a self-ensembling network for domain adaptation in 3D point clouds, which leverages a semi-supervised learning framework for effective knowledge transfer from a labeled source domain to an unlabeled target domain.

### 5.1.5    3D Depth Estimation

3D depth estimation is crucial for accurately understanding the geometry of a scene, enabling perception systems to reconstruct spatial relationships and object structures. MVP-Net (Chen et al., 2024b) enhances reconstruction by using depth estimation in a multi-view, cross-modal distillation approach for better point cloud upsampling. KD-MonoRec (Xiao et al., 2024) leverages monocular RGB images as input and employs KD along with point cloud optimization to improve depth estimation. LiRCDepth (Sun et al., 2025) utilizes RGB images and radar point clouds and proposes a lightweight depth estimation model via KD.

### 5.1.6    3D Representation Learning

Representation learning focuses on extracting meaningful representations from data. Collecting a large and high-quality annotated dataset in 3D is a costly and time-consuming process, and pre-training 3D models requires access to large-scale 3D datasets, which is not a trivial task. However, some works, such as DCGLR (Fu et al., 2022), have effectively utilized knowledge distillation (KD) with only 3D data to address this challenge. However, given that numerous large-scale datasets and powerful foundation models are available in the 2D domain, several methods (Liu et al., 2021d; Sautier et al., 2022; Yu et al., 2022b; Yan et al., 2022; Wang et al., 2023m; Zhang et al., 2023g; Yan et al., 2023; Yao et al., 2024; Zhang et al., 2024f; Dutt et al., 2024) have leveraged these capabilities to learn rich representations for 3D point clouds.

Liu et al. (2021d) introduces contrastive learning to transfer 2D semantic knowledge to 3D networks. SLidR (Sautier et al., 2022) introduces a self-supervised 2D-to-3D representation distillation framework that uses superpixel-driven contrastive loss to align image and LiDAR features, enabling effective pre-training of 3D perception models for autonomous driving. Qi et al. (2023) proposes a framework that can be used in either a single-modal or cross-modal setting.

3D representations can be learned with the help of text (Huang et al., 2024c) or through sequences of point clouds, as demonstrated in Zhang et al. (2023k), which develops a self-supervised method for learning 4D point cloud representations using KD.

### 5.1.7    3D Recognition

Visual place recognition is an important task in computer vision and robotics, aiming to identify previously visited locations based on visual input such as camera image and LiDAR point clouds. PointMCD (Zhang et al., 2023f) and DistilVPR (Wang et al., 2024h) are two innovative cross-modal KD frameworks in this field, with PointMCD employing a multi-view framework, as discussed further in Section 5.2.

### 5.1.8 3D Completion

Point cloud completion refers to the process of restoring missing geometric details. KD helps transfer learned priors from unpaired data, reducing the need for large annotated datasets and improving completion performance in data-scarce scenarios.

**Shape Completion:** Shape completion aims to reconstruct a complete point cloud from occluded input point cloud. RaPD (Fan et al., 2022) is the first semi-supervised point cloud completion method that utilizes prior distillation. Lin et al. (2024a) uses loss distillation and Zhou et al. (2024b) propose a hierarchical self-distillation point cloud completion method.

**Scene Completion:** LiDAR sensors often capture incomplete point clouds due to occlusions from objects or their surroundings, making large-scale 3D scene interpretation challenging. KD can enhance their performance as mentioned in Xia et al. (2023); Wang et al. (2024i); Zhang et al. (2024e); Wang et al. (2025b).

**Depth Completion:** Depth completion is the process of estimating a dense depth map from sparse or incomplete depth measurements, such as those obtained from LiDAR sensors. KD can aid in predicting missing depth information (Hwang et al., 2021; Liu et al., 2022b; Kim et al., 2024b; Huang et al., 2024a).

### 5.1.9 3D Registration

KD can enhance point cloud registration by transferring knowledge from a large, complex model to a smaller, efficient one while preserving performance (Löwens et al., 2024; Jiang et al., 2024a).

### 5.1.10 Other 3D Tasks

KD is applied to various other 3D tasks with different types of distillation. This includes generation (Yi et al., 2024b), stylization (Decatur et al., 2024), sampling (Huang et al., 2022c) and accordance detection, as in (Van Vo et al., 2024), which utilizes all types of KD.

## 5.2 Multi-view Input

In recent years, multi-view learning has emerged as a powerful approach for addressing complex tasks, such as 3D object detection and reconstruction. While leveraging information from multiple perspectives enables models to achieve more accurate and robust results, the diversity of data modalities and the challenge of transferring knowledge across them present significant challenges. To address these, several studies have explored novel KD techniques that allow models to share and refine knowledge across different views or modalities. This section reviews recent advancements in multi-view distillation methods, categorizing them by task and the modality. The existing works can broadly be divided into two main groups: 3D object detection and 3D shape recognition, with other tasks represented by individual studies.

### 5.2.1 3D Object Detection

3D object detection has been a focal point in multi-view learning, with numerous studies proposing novel distillation techniques to enhance performance (Li et al., 2022c; Chen et al., 2022b; Klingner et al., 2023; Wang et al., 2023p; Jang et al., 2023; Zhao et al., 2024a; Jiang et al., 2024e). To this end, SimDistill (Zhao et al., 2024a) introduces a LiDAR-camera fusion-based teacher and a simulated fusion-based student for multi-modal learning. By maintaining identical architectures and incorporating a geometry compensation module, the student learns to generate multi-modal features solely from multi-view images.

Klingner et al. (2023) proposes a comprehensive framework for multi-camera 3D object detection by leveraging cross-task and cross-modal distillation. Cross-task distillation from an instance segmentation teacher avoids ambiguous error propagation, while cross-modal feature distillation and adversarial training refine 3D representations using a LiDAR-based teacher. Cross-modal output distillation further enhances detection accuracy.

Wang et al. (2023p) proposes a KD framework for aligning a BEV-based student's features with a LiDAR-based teacher's, addressing the depth and geometry inference issue compared to LiDAR-based methods.

Table 7: Summary of Knowledge Distillation in Multi-View Tasks.

| Method | Task | Teacher Modality | Student Modality | Short Description |
|---|---|---|---|---|
| KD-MVS(Ding et al., 2022b) | Multi-view Stereo Depth Reconstruction | Multi-view Images | Multi-view Images | Self-supervised MVS distillation for depth reconstruction. |
| MTS-Net(Tian et al., 2022b) | Multi-view Image Classification | Multi-view Images | Multi-view Images | Multi-view learning and distillation for image classification. |
| BEV-LGKD(Li et al., 2022c) | Multi-view BEV 3D Object Detection | LiDAR | Multi-view RGB Images | LiDAR-guided distillation with foreground masks and depth distillation for BEV perception. |
| BEVDistill(Chen et al., 2022b) | Multi-view 3D Object Detection | LiDAR | Multi-view Images | Cross-modal BEV distillation for unifying image and LiDAR features. |
| STXD(Jang et al., 2023) | Multi-view 3D Object Detection | LiDAR | Multi-view Images | Structural and temporal distillation with cross-correlation regularization and response distillation. |
| PointMCD(Zhang et al., 2023f) | 3D Shape Recognition | Multi-view Images | 3D Point Cloud | Multi-view cross-modal distillation for 3D shape recognition. |
| Acar et al.(Acar et al., 2023) | Vision-based Robotic Manipulation | Multi-view Images | Single-camera | Distillation from multi-camera to single-camera for robust manipulation. |
| Zhang et al.(Zhang et al., 2023e) | Multi-view BEV Detection | Multi-view BEV Images | Multi-view BEV Images | Structured distillation for efficient BEV detection. |
| MVKD(Wang et al., 2023c) | Semantic Segmentation | Multi-view Images | Single-view Images | Multi-view distillation with co-tuning and feature/output distillation losses. |
| MKDT(Lin & Tseng, 2023) | Human Action Recognition | Multi-view Videos | Single-view Videos | Multi-view distillation with hierarchical vision transformer for incomplete data. |
| 3X3KD(Klingner et al., 2023) | Multi-camera 3D Object Detection | LiDAR | Multi-view Images | Cross-modal distillation for multi-camera 3D object detection. |
| DistillBEV(Wang et al., 2023p) | 3D Object Detection for Autonomous Driving | LiDAR | Multi-view BEV Images | Distillation from LiDAR to BEV for enhanced 3D perception. |
| FSD-BEV(Jiang et al., 2024e) | 3D Object Detection for Autonomous Driving | Self-distillation | Self-distillation | Foreground self-distillation for single-model distillation. |
| SimDistill(Zhao et al., 2024a) | Multi-modal 3D Object Detection | LiDAR-camera Fusion | Multi-view Images | Distillation for 3D object detection with modality gap compensation. |
| GMViT(Xu et al., 2024b) | 3D Shape Recognition | Multi-view Images | Multi-view Images | Knowledge distillation with GMViT for efficient 3D shape analysis. |
| MT-MV-KDF(Qiang et al., 2024) | Myocardial Infarction Detection and Localization | Multi-view ECG | Single-view ECG | Multi-task, multi-view distillation with CNN and attention modules. |

Cross-modal KD often suffers from feature distribution mismatches. FSD scheme (Jiang et al., 2024e) eliminates the need for pre-trained teachers and complex strategies.

BEV-LGKD (Li et al., 2022c) proposes a LiDAR-guided KD framework for multi-view BEV 3D object detection. By transforming LiDAR points into BEV space and generating foreground masks, the method guides RGB-based BEV models without requiring LiDAR at inference. Depth distillation is also incorporated to improve depth estimation, enhancing BEV perception performance.

BEVDistill (Chen et al., 2022b) introduces a cross-modal BEV distillation framework for multi-view 3D object detection by unifying image and LiDAR features in BEV space and adaptively transfers knowledge across non-homogeneous representations in a teacher-student paradigm.

STXD (Jang et al., 2023) proposes a structural and temporal cross-modal distillation framework for multi-view 3D object detection. It reduces redundancy in the student's feature components by regularizing cross-correlation while maximizing cross-modal similarities. Additionally, it encodes temporal relations across frames using similarity maps and employs response distillation to enhance output-level knowledge transfer.

### 5.2.2   3D Shape Recognition

3D shape recognition is another critical area where multi-view distillation has been applied to bridge the gap between 2D and 3D representations (Zhang et al., 2023f; Xu et al., 2024b). PointMCD (Zhang et al., 2023f) bridges 2D visual and 3D geometric domains through a multi-view cross-modal distillation framework, by distilling knowledge from the teacher to a point encoder student. Group Multi-View Transformer (GMViT) (Xu et al., 2024b) addresses the limitations of view-based 3D shape recognition methods, which often struggle with large model sizes that are unsuitable for memory-limited devices. To overcome this, GMViT introduces a large high-performing model designed to enhance the capabilities of smaller student models

### 5.2.3  Other Tasks

In addition to 3D object detection and 3D shape recognition, multi-view distillation techniques have been applied to a range of other tasks, including multi-view image classification (Tian et al., 2022b), vision-based robotic manipulation (Acar et al., 2023), multi-view BEV detection (Zhang et al., 2023e), multi-view stereo depth reconstruction (Ding et al., 2022b), semantic segmentation (Wang et al., 2023c), human action recognition (Lin & Tseng, 2023), and medical applications (Qiang et al., 2024). These studies are summarized as follows:

MTS-Net (Tian et al., 2022b) integrates KD with multi-view learning, redefining the roles of the teacher and student models. This end-to-end approach optimizes both view classification and knowledge transfer, with extensions like MTSCNN refining multi-view feature learning for image recognition.

In robotic manipulation, Acar et al. (2023) improves vision-based reinforcement learning by transferring knowledge from a teacher policy trained with multiple camera viewpoints to a student policy using a single viewpoint.

Zhang et al. (2023e) uses a spatio-temporal distillation and BEV response distillation by aligning the student's outputs with those of the teacher, addressing the computational complexity of multi-view BEV detection methods.

Supervised multi-view stereo methods face challenges due to the scarcity of ground-truth depth data. The self-supervised KD-MVS framework (Ding et al., 2022b) uses a teacher trained with photometric and featuremetric consistency to probabilistically transfer knowledge to a student.

MVKD (Wang et al., 2023c) introduces a multi-view KD framework for efficient semantic segmentation. The framework aggregates knowledge from multiple teacher models and transfers it to a student model. To ensure consistency among the multi-view knowledge, MVKD employs a multi-view co-tuning strategy. Additionally, it proposes multi-view feature distillation loss and multi-view output distillation loss to effectively transfer knowledge from multiple teachers to the student.

MKDT (Lin & Tseng, 2023) introduces a multi-view KD transformer framework for human action recognition, addressing the challenge of incomplete multi-view data. The framework consists of a teacher network and a student network, both utilizing a hierarchical vision transformer with shifted windows to effectively capture spatio-temporal information.

MT-MV-KDF (Qiang et al., 2024) introduces a novel multi-task multi-view KD framework for myocardial infarction detection and localization. Multi-view learning extracts ECG feature representations from different views, while multi-task learning captures similarities and differences across related tasks.

In summary, multi-view distillation techniques have shown significant potential across a broad range of tasks, from 3D object detection and shape recognition to specialized applications such as robotic manipulation and medical diagnostics. The variety of approaches, as highlighted in Table 7, emphasizes the adaptability of these methods to various modalities and challenges.

### 5.3  Text Input

Knowledge distillation has been widely applied across various NLP tasks. This technique is particularly valuable in NLP due to the large scale of state-of-the-art models, namely BERT and GPT. While these models are highly accurate, they can be impractical for deployment in resource-constrained environments. By distilling the rich representations learned by these large models into lightweight versions, KD helps maintain high accuracy in tasks such as neural machine translation, question answering, text generation, event detection, document retrieval, text recognition, named entity recognition, text summarization, natural language understanding, sentiment analysis, and text classification.

**Neural Machine Translation (NMT)**: While NMT has significantly outperformed traditional statistical methods, large models like transformers are computationally intensive. KD addresses this challenge by distilling the teacher's output to the student, achieving comparable translation quality with reduced com-

putational costs. This enables the development of more efficient and scalable NMT solutions (Kim & Rush, 2016; Tan et al., 2019; Wang et al., 2021a; Liang et al., 2022a; Jin et al., 2024)

**Question Answering (QA)**: In QA, KD transfers both answer predictions and intermediate representations, such as attention distributions, from a large teacher model to a smaller student model. This process helps maintain high accuracy while minimizing computational costs, making QA systems more efficient and suitable for real-world applications (Hu et al., 2018; Izacard & Grave, 2020; Yang et al., 2020; Liu et al., 2021b; Yin et al., 2024a).

**Text Generation:** In text generation, KD enables the transfer of a teacher's generative capabilities to a smaller student model, training it to mimic the vocabulary distributions and language patterns of the teacher. Chen et al. (2019c) focuses on leveraging BERT's contextual understanding for efficient text generation, while Haidar & Rezagholizadeh (2019) enhances fluency and diversity by combining distillation with GANs.

**Event Detection:** KD is applied in event detection to distill knowledge from larger, more complex models to smaller, more efficient ones. Ren et al. (2021) improves event detection across languages by distilling knowledge from high-resource to low-resource models, while Yu et al. (2021) uses distillation to retain learned knowledge while adapting to new events. Meanwhile, Liu et al. (2019a) employs adversarial imitation to enhance detection accuracy.

**Document Retrieval:** In document retrieval, KD transfers the ranking and matching capabilities of large teacher models to smaller student models, typically replicating relevance scores and interaction features (Shakeri et al., 2019; Chen et al., 2021c).

**Text Recognition:** KD in text recognition facilitates the transfer of both visual and contextual features from a teacher to a student, allowing the student to maintain high recognition accuracy with reduced computational cost (Bhunia et al., 2021; Wang & Du, 2021).

**Named Entity Recognition (NER):** In NER, KD helps a smaller student model learn from the teacher's contextual understanding and classification abilities (Liang et al., 2021; Ge et al., 2024).

**Text Summarization:** KD in text summarization transfers summarization capabilities from teacher to student, training the student to replicate output sequences and probability distributions (Chen et al., 2017; Liu et al., 2020b; Nguyen & Luu, 2022).

**Natural Language Understanding (NLU):** KD in NLU transfers the rich linguistic understanding of the teacher models to the student. In NLU, the student learns to replicate the teacher's output probabilities and internal representations across various tasks, ensuring high performance while reducing computational costs (Liu et al., 2019c; Tang et al., 2019; FitzGerald et al., 2022).

**Sentiment Analysis:** In sentiment analysis, KD distills the sentiment classification capabilities of the teacher to the student, training it to mimic the teacher's probability distributions and decision patterns (Lehečka et al., 2020; Malki, 2023).

**Text Classification**: In text classification, KD transfers classification capabilities from the teacher to the student, enabling the student to mimic the teacher's feature representations and output probabilities (Xu & Yang, 2017; Li & Li, 2021).

### 5.4   Speech Input

Deep neural acoustic models have achieved remarkable success in speech recognition tasks; however, their complexity poses challenges for real-time deployment on devices with limited computational resources. As the demand for efficient and rapid processing increases on embedded platforms, conventional large-scale models often increases on embedded platforms. To this end, KD has garnered significant attention in various speech-related tasks, including speech recognition, speech enhancement, speaker recognition and verification, speech translation, speech synthesis, speech separation, spoken language identification and understanding, deepfake speech and spoofing detection, audio classification and tagging, spoken question answering and conversational AI, and audio captioning and retrieval.

**Speech Recognition (ASR):** KD is important in enhancing ASR models by transferring expertise from high-precision teacher models to student models, supporting rapid inference with minimal loss in accuracy. This also improves domain adaptation, making ASR effective in poor acoustic environments or in expert domains, including medical transcription (Chebotar & Waters, 2016; Markov & Matsui, 2016; Takashima et al., 2018; Mun'im et al., 2019; Kurata & Saon, 2020; Yoon et al., 2021; You et al., 2021a; Yoon et al., 2022; Zhao et al., 2022c; Kim et al., 2023b; Park et al., 2023; Zhang et al., 2023j).

**Speech Enhancement:** KD in speech enhancement enables denoising through high-fidelity speech representations and model compression (Watanabe et al., 2017; Wu et al., 2019; Kim & Kim, 2021; Shin et al., 2023; Park et al., 2024).

**Speaker Recognition and Verification:** In speaker recognition and verification, KD helps reduce model complexity while preserving speaker identity features, allowing for fast and secure authentication. It also enhances robustness against adversarial attacks and cross-domain variations, improving performance in noisy or multilingual environments (Mingote et al., 2020; Peng et al., 2022; Liu et al., 2022a; Jin et al., 2023b; Mingote et al., 2023; Truong et al., 2024; Hoang et al., 2024).

**Speech Translation:** KD improves the efficiency of multilingual translation models by enabling knowledge sharing between high-resource and low-resource languages (Liu et al., 2019e; Gaido et al., 2020; Inaguma et al., 2021; Lei et al., 2023). This enhances translation fluency while maintaining low latency for real-time applications.

**Speech Synthesis (Text-to-Speech):** In speech synthesis, KD reduces the complexity of deep generative models while retaining natural-sounding speech output (Oord et al., 2018; Xu et al., 2020a; Li et al., 2024f). It also helps distill expressive features, making synthetic voices more human-like with lower computation costs.

**Speech Separation:** KD compresses complex separation models into smaller networks while preserving speaker distinction (Chen et al., 2018; Zhang et al., 2021b). It enables real-time speaker and cocktail-party effect reduction on low-power devices.

**Spoken Language Identification and Understanding:** In this task, KD enables language adaption in low-resource languages through the transfer of information between high-resource multilingual teacher models (Shen et al., 2018; 2019c; 2020; Kim et al., 2021b; Cappellazzo et al., 2022; Mao & Zhang, 2023; Cappellazzo et al., 2023). As a result, such mechanism empowers lighter, accelerated models to achieve high performance in language identification and speech dialogue comprehension.

**Deepfake Speech and Spoofing Detection:** With growing concerns about synthetic voice fraud, the need for reliable approaches to deepfake detection has increased. KD strengthens anti-spoofing by transferring knowledge between DNNs trained on a range of deepfake speech datasets and lightweight detection architectures (Liao et al., 2022; Xue et al., 2023; Ren et al., 2023c;b; Lu et al., 2024; Wani & Amerini, 2025).

**Audio Classification and Tagging:** KD strengthens feature learning relevant to sound event detection, enhancing model efficiency and allowing deployment on embedded platforms such as IoT and edge AI platforms (Chang et al., 2020; Yin et al., 2021; Schmid et al., 2023; Liang et al., 2022b; Dinkel et al., 2023; Tang et al., 2024).

**Spoken Question Answering and Conversational AI:** KD enables efficient question-answer through the distillation of contextual information in deep, lengthy transformer models, mapping them onto efficient, quick-answer-producing models (You et al., 2020a;b; 2021a;b).

**Audio Captioning and Retrieval:** In this task, KD enables deep audio-text embedding models to be compressed while maintaining generalization, making them suitable for efficient multimedia search architectures (Xu et al., 2024c; Primus & Widmer, 2024).

## 5.5 Video Input

Knowledge distillation has been widely applied in image and language tasks, but its potential in video-based applications is gaining increasing attention. Recent studies have explored KD for video action recognition

Table 8: Summary of video knowledge distillation methods based on their sources of distillation.

| Category | Method |
|---|---|
| **Feature-based** | Wang et al.(Wang et al., 2024c), Liu et al.(Liu et al., 2019b), V2I-DETR (Jiang et al., 2024c), MaskAgain(Li et al., 2023e) |
| **Similarity-based** | RSKD(Wang & Tang, 2023), Bhardwaj et al.(Bhardwaj et al., 2019), OOKD (Kim et al., 2024a), DL-DKD(Dong et al., 2023a) |
| **Logit-based** | Camarena et al.(Camarena et al., 2024), Wu et al.(Wu & Chiu, 2020), PKD (Soufleri et al., 2024), MobileVOS(Miles et al., 2023), V2I-DETR(Jiang et al., 2024c), DTO(Monti et al., 2022), MVD (Wang et al., 2023l), RankDVQA-mini(Feng et al., 2024a), VideoAdviser (Wang et al., 2023o), Afouras et al.(Afouras et al., 2020), Perez et al.(Perez et al., 2020) |

(Liu et al., 2019b; Wu & Chiu, 2020; Camarena et al., 2024; Wang et al., 2024c; Soufleri et al., 2024), video classification (Bhardwaj et al., 2019; Wang & Tang, 2023), video object segmentation (Miles et al., 2023; Jiang et al., 2024c), video instance segmentation (Kim et al., 2024a), Partially Relevant Video Retrieval (PRVR) (Dong et al., 2023a), trajectory forecasting (Monti et al., 2022), and representation learning through masked video modeling (Li et al., 2023e; Wang et al., 2023l). Table 8 summarizes video distillation methods.

The main challenge in deep learning models, particularly in applications such as video quality estimation (Feng et al., 2024a) and attention prediction (Fu et al., 2020), is the high computational cost and complexity of large models, which restrict their execution on edge devices. KD effectively addresses this by enabling a smaller, more efficient model (student) to learn from a larger, complex model (teacher) with minimal performance reduction while significantly decreasing computational and memory requirements. MobileVOS (Miles et al., 2023) specifically focuses on semi-supervised video object segmentation on resource-constrained devices, for example mobile phones, and proposes a pixel-wise representation distillation loss to efficiently transfer structural information from the teacher to the student while ensuring feature consistency across frames. In Kim et al. (2024a), the authors propose a real-time approach that distills similarity information from an offline teacher model, which processes entire video sequences, to an online model, which processes individual frames. Mullapudi et al. (2019) introduces an online approach that applies feature distillation on live videos for low-cost semantic segmentation.

Furthermore, KD is applicable in multi-modal learning, enabling cross-modal knowledge transfer, such as video-to-image (Jiang et al., 2024c), video-to-text (Wang et al., 2023o), text-to-video (Dong et al., 2023a), audio-to-video (Afouras et al., 2020), and video-to-audio (Perez et al., 2020). Notably, (Pan et al., 2020) leverages KD in a spatio-temporal graph model for the video captioning task, making the model more robust against spurious correlations. V2I-DETR enhances medical video lesion detection by distilling temporal knowledge from a video-based teacher model to an image-based student model, achieving real-time inference speed with high accuracy (Jiang et al., 2024c).

# 6 Applications

Knowledge distillation can be applied across various fields and has important applications, including distillation in LLMs, distillation in foundation models and in vision transformers, distillation in self-supervised learning, distillation in diffusion models, and distillation in visual recognition tasks. The following provides a detailed explanation of each application.

## 6.1 Distillation in Large Language Models

LLMs have recently revolutionized the field of NLP, with significant applications across various domains in both academia and industry. However, these models pose critical challenges due to their large number of parameters, which limits their deployment in real-time scenarios. Additionally, these giant models, with a considerable number of attention blocks, have been shown to be overparameterized (Kaplan et al., 2020;

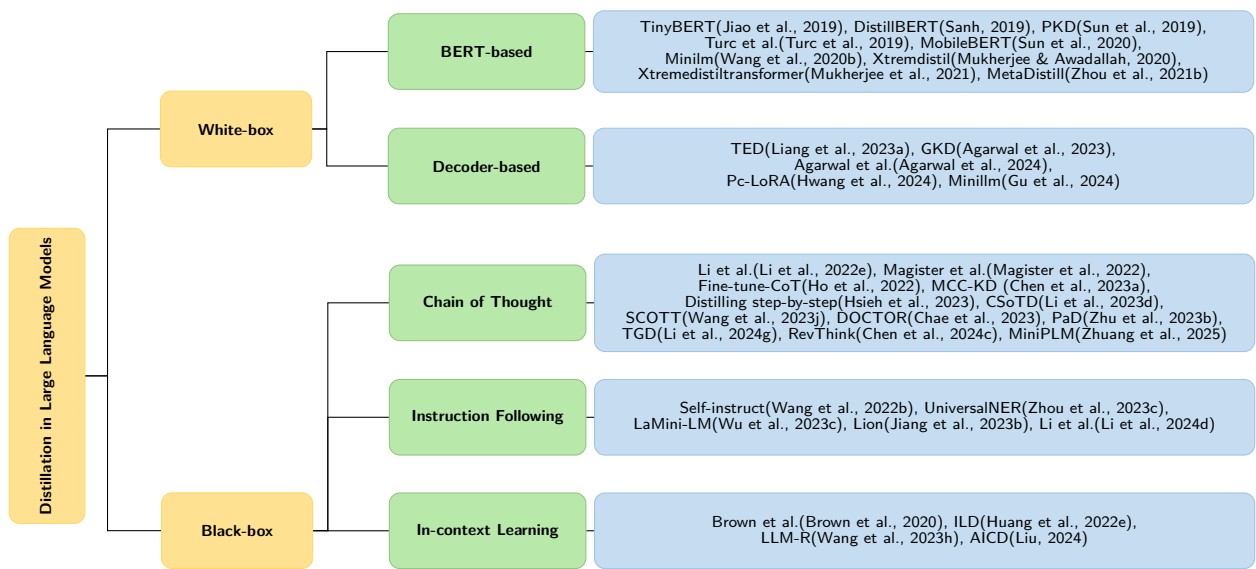

Figure 8: Classification of distillation methods for large language models.

Fischer et al., 2024). Therefore, KD plays a crucial role in compressing LLMs into Small Language Models (SLMs). Existing KD methods for LLMs are mainly classified into two major categories: white-box distillation and black-box distillation. In white-box KD, logits and intermediate outputs of the LLMs are accessible, enabling the student to benefit from the internal information of the teacher. This information can be either logits (Hinton, 2015) or intermediate features (Romero et al., 2014), such as outputs of each attention block or attention scores. On the other hand, in black-box distillation, only the predictions of the teacher are available for distillation, which is common in many closed-source LLMs like GPT-4 (Achiam et al., 2023) and Gemini (Team et al., 2023), limiting the distillation process to only the predictions. Black-box methods are categorized into In-Context Learning (ICL) (Huang et al., 2022e), Chain of Thought (CoT) (Li et al., 2022e), and Instruction Following (IF) (Wang et al., 2022b). Figure 8 summarizes the black-box and white-box distillation methods for LLMs.

Most white-box KD methods focus on the BERT (Kenton & Toutanova, 2019) model (Turc et al., 2019; Jiao et al., 2019; Sanh, 2019; Sun et al., 2019; 2020; Wang et al., 2020b; Mukherjee & Awadallah, 2020; Mukherjee et al., 2021). DistilBERT (Sanh, 2019) was one of the works that initialized the student with a pre-trained teacher and minimized the soft probabilities between the student and the teacher during the pre-training phase. In a concurrent work, MiniLM (Wang et al., 2020b) proposed distilling the self-attention module of the last transformer layer of the teacher. MobileBERT (Sun et al., 2020) first trains a special teacher and then uses feature and attention transfer to distill knowledge to the student. TinyBERT (Jiao et al., 2019) introduced a KD method for both pre-training and task-specific training of a smaller version of BERT by distilling the logits, attention matrices, hidden states, and the output of the teacher's embedding layer. PKD (Sun et al., 2019) introduces two strategies for incremental distillation, while Mukherjee & Awadallah (2020) and Mukherjee et al. (2021) propose architecture-agnostic and task-agnostic distillation methods, respectively.

Recently, several new white-box KD methods have been proposed (Zhou et al., 2021b; Agarwal et al., 2023; Liang et al., 2023a; Hwang et al., 2024; Agarwal et al., 2024; Gu et al., 2024; Anshumann et al., 2025), primarily designed for decoder-based LLMs. MetaDistill (Zhou et al., 2021b) utilizes a feedback mechanism within a meta-learning framework, GKD (Agarwal et al., 2023) trains the student on its self-generated output sequences, and PC-LoRA (Hwang et al., 2024) concurrently conducts progressive model compression and fine-tuning. One of the most notable recent works is MiniLLM (Gu et al., 2024), which suggests using the reverse of the Kullback-Leibler Divergence to prevent student models from overestimating the low-

probability distribution of the teacher. CKA(Dasgupta & Cohn, 2025) matches hidden states of different dimensionality, allowing for smaller students and higher compression ratios, and Gong et al. (2025) Alignings feature dynamics for effective KD.

Furthermore, white-box KD has been successfully employed in industry to train small language models in pre-training stage (Liu et al., 2023b; Sreenivas et al., 2024; Muralidharan et al., 2024; Wang et al., 2024b; Team et al., 2025; Yang et al., 2025). Liu et al. (2023b) proposes a data-free, quantization-aware training approach in which a quantized version of the teacher is trained as a student using data-free distillation. Pruning-aware distillation has also been adopted by NVIDIA (Muralidharan et al., 2024; Sreenivas et al., 2024). Muralidharan et al. (2024) prunes the teacher's neurons and uses a small portion of the original data for post-training the student via KD. In a related approach, Sreenivas et al. (2024) additionally introduces a teacher correction step prior to distillation. Gemma3 (Team et al., 2025) and Qwen3 (Yang et al., 2025) employ logit sampling and weak-to-strong distillation, respectively.

Black-box KD has recently garnered increased attention due to the rise of closed-source LLMs. The first category of black-box KD is ICL, initially introduced in GPT-3 (Brown et al., 2020). In ICL, the teacher receives a task description, a few task examples, and a query, where the teacher predicts a response that the student aims to mimic. Huang et al. (2022e) combines in-context learning objectives with language modeling objectives to distill both the ability to interpret in-context examples and task knowledge into smaller models. AICD (Liu, 2024) explores the simultaneous distillation of in-context learning and reasoning, leveraging the autoregressive nature of LLMs to optimize the likelihood of all rationales in context. Wang et al. (2023h) introduces a novel framework to iteratively train dense retrievers capable of identifying high-quality in-context examples for LLMs.

The second category of black-box KD is IF, which seeks to enhance the zero-shot capabilities of LLMs through instruction prompts. Specifically, in IF, the teacher generates task-specific instructions on which the student network is fine-tuned (Jiang et al., 2023b; Zhou et al., 2023c; Li et al., 2024d; Wang et al., 2022b; Wu et al., 2023c). Self-Instruct (Wang et al., 2022b) generates instructions, input, and output samples, filters out invalid samples, and then fine-tunes the student model. UniversalNer (Zhou et al., 2023c) explores mission-focused instruction tuning to train student models capable of excelling in a broad application class. Lion (Jiang et al., 2023b) utilizes an adversarial framework to generate challenging instructions for students, while LaMini-LM (Wu et al., 2023c) compiles a large dataset containing 2.58 million instructions based on both existing and newly generated instructions.

The last and most popular black-box KD approach is COT, where the teacher model generates rationales alongside predictions, providing the student network with intermediate inference steps. In COT, the student is expected to generate predictions and reasonings similar to the teacher, similar to feature learning where the student mimics the teacher's intermediate steps for better predictions. This concept was first introduced by Li et al. (2022e) and has been expanded upon in subsequent works (Ho et al., 2022; Hsieh et al., 2023; Chen et al., 2023a; Magister et al., 2022; Li et al., 2023d; Wang et al., 2023j; Zhu et al., 2023b; Chae et al., 2023; Li et al., 2024g; Chen et al., 2024c; Zhuang et al., 2025). Magister et al. (2022) explores the trade-off between model and dataset size, demonstrating that the reasoning ability of LLMs can be transferred to SLMs. MCC-KD (Chen et al., 2023a) generates multiple reasonings for each sample and enforces consistency and diversity among them. Hsieh et al. (2023) employs a multi-task learning framework to train the student with the teacher's reasonings. In contrast to other approaches that overlook reasonings with incorrect labels, Li et al. (2024g) utilizes both positive and negative data. Most recently, Chen et al. (2024c) demonstrated that generating backward questions and reasoning, and training the student on both forward and backward reasoning, significantly enhances the generalization ability of the student.

In summary, white-box KD methods possess greater efficacy in transferring knowledge by granting the student access to the internal information of the teacher. However, they typically come with high computational costs when distilling knowledge during student training. Moreover, most powerful LLMs are currently not open-source, rendering the use of white-box KD infeasible for these models. On the other hand, black-box methods enable KD from any model. Nevertheless, as black-box methods lack access to the internal workings of the teacher and solely rely on training the student with data generated by the teacher, they may not encompass all possibilities and could exhibit lower generalizability. One potential strategy for

leveraging white-box distillation could involve initially distilling knowledge from a closed-source model to an open-source one using black-box KD, and subsequently applying a white-box method for further distillation.

## 6.2 Foundation Model Distillation

Foundation Models (FMs) are large-scale, pre-trained architectures designed to generalize across multiple downstream tasks. KD serves as a critical technique in optimizing these models by transferring their knowledge into smaller, task-specific, or efficient student models. KD addresses challenges such as computational costs and deployment on resource-constrained devices, making it indispensable for scaling FMs' applications.

Prominent FMs, including VLMs such as CLIP (Radford et al., 2021), conversational agents such as Chat-GPT (Brown et al., 2020; Radford, 2018; Baktash & Dawodi, 2023), and generative models such as DALL-E (Ramesh et al., 2021), illustrate the diverse domains from which knowledge can be distilled using KD. KD is utilized to distill the complex, multi-modal representations from these models into simpler ones optimized for specific tasks. In addition, vision-specific models such as SAM (Kirillov et al., 2023) and DINO (Caron et al., 2021) serve as valuable sources of knowledge that can be distilled for various applications, including segmentation and unsupervised object discovery. Transformer-based models, including ViT (Dosovitskiy, 2020), DETR (Carion et al., 2020), Swin Transformer (Liu et al., 2021e), and DeIT (Touvron et al., 2021) also provide foundational knowledge that can be transferred through KD to enhance performance in vision-related tasks.

To provide a comprehensive overview of these FMs, this section explores KD for each model individually. Figure 9 presents a structured overview of the topics covered in this section, outlining the different types of FMs and their respective applications. To complement this, Figure 10 illustrates the architectures of these prominent models, highlighting key points where data is commonly extracted and distilled for downstream tasks.

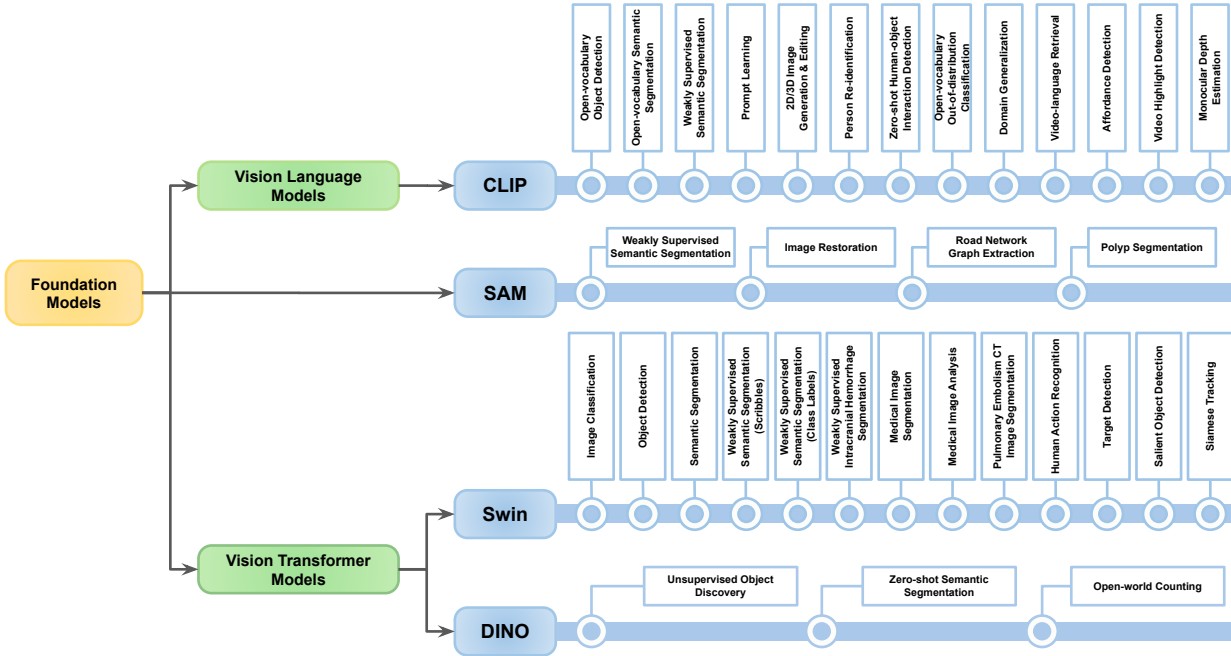

Figure 9: Categorization of foundation models based on their type, and the tasks for which their knowledge is distilled.

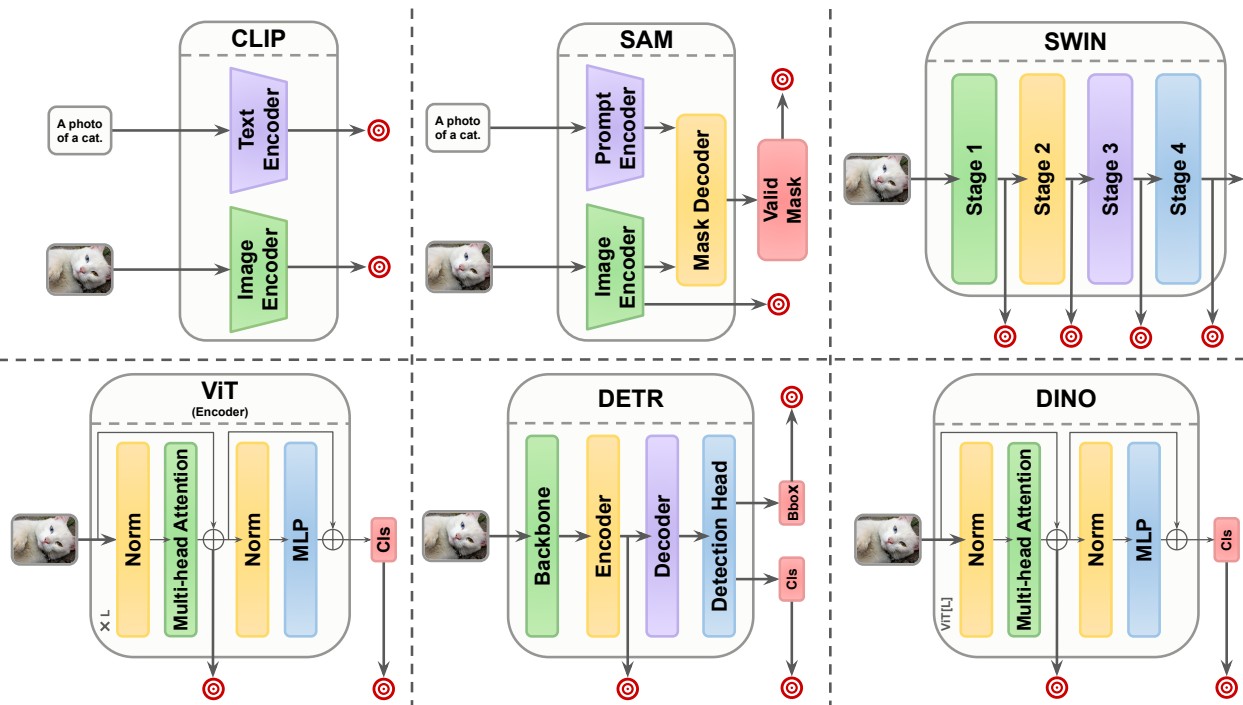

Figure 10: Architectural overview of prominent foundation models, indicating key points for data extraction and distillation to downstream tasks.

### 6.2.1 Vision-language Models and Knowledge Distillation

VLMs bridge visual and textual data through joint representation learning, offering capabilities to align these data types effectively. KD plays a central role in refining these models (Chen et al., 2024i; Wang et al., 2023h; Minderer et al., 2022), enabling their general-purpose knowledge to be adapted for more compact or task-specific models, enhancing scalability and reducing computational overhead.

CLIP (Contrastive Language-Image Pre-training) (Radford et al., 2021), one of the most significant VLMs, exemplifies this integration. Trained on paired image and text data, CLIP achieves remarkable generalization by aligning global visual and textual representations. Through KD, these capabilities are transferred into student models designed for downstream tasks, including several key areas:

**Open-vocabulary Tasks** In conventional vision tasks such as semantic segmentation and object detection, a dataset-focused methodology is traditionally employed, where the most successful techniques depend on a training dataset that has been manually annotated for a predefined and narrow range of categories. However, the rise of advanced VLMs, namely CLIP, is driving a shift towards an open-world paradigm. These models are trained through a simple yet scalable process that aligns image-text pairs, leveraging broad, loosely descriptive captions that can be collected in vast amounts with little human intervention. In object detection, KD techniques (Gu et al., 2021b; Ma et al., 2022; Bangalath et al., 2022; Zang et al., 2022; Xie & Zheng, 2022; Kim et al., 2023a; Wu et al., 2023d; Wang et al., 2023i; Du et al., 2022; Feng et al., 2022; Gao et al., 2022b; Huynh et al., 2022; Long et al., 2023; Zhao et al., 2022f; Lin et al., 2022a; Kuo et al., 2022; Zhou et al., 2022d; Yao et al., 2022) distill CLIP's global visual-textual knowledge into models capable of detecting objects outside fixed vocabularies, while in semantic segmentation (Lüddecke & Ecker, 2022; Ding et al., 2022a; Li et al., 2022a; Zabari & Hoshen, 2021; Zhou et al., 2022a; 2023d; Ma et al., 2022; Liang et al., 2023b; Xu et al., 2022a; Ghiasi et al., 2022; Shin et al., 2022a; Qin et al., 2023; Wysoczańska et al., 2024a; Chen et al.,

2023b), and part segmentation (Choi et al., 2024) KD aids in transferring CLIP's textual alignment to dense pixel-level tasks, enabling fine-grained scene understanding. Furthermore, Wei et al. (2025) has explored open-vocabulary customization from CLIP through Data-Free Knowledge Distillation (DFKD), introducing a framework that enables CLIP model adaptation without requiring the original training data.

**Weakly Supervised Semantic Segmentation**  The utilization of CLIP has recently been extensively explored for weakly supervised semantic segmentation (Xie et al., 2022; Lin et al., 2023b; Xu et al., 2023b; Deng et al., 2023; Murugesan et al., 2024; Zhang et al., 2024a; Zhu et al., 2024a; Jang et al., 2024; Zhou et al., 2025) where it enables the generation of segmentation masks with minimal labeled data, by leveraging its ability to align textual and visual representations within a shared embedding space. Through this approach, text prompts representing class labels guide the identification of relevant regions in an image by querying CLIP's pre-trained image-text alignment capabilities. The visual features extracted by CLIP's image encoder are matched to the semantic meaning encapsulated in text embeddings, producing coarse class activation maps that localize objects and regions associated with the given labels.

**Prompt Learning**  Prompt learning is a method that enables the use of a large pre-trained model, such as CLIP, for specific tasks without having to retrain the entire model. This approach suggests adjusting the model's representations for particular tasks by using learnable soft prompts, either text-based or visual, rather than relying on pre-designed hard prompts. Several methods (Zhou et al., 2022c;b; Li et al., 2024h; Ge et al., 2023; Rao et al., 2022) have been proposed to implement this approach.

**Generation and Editing**  Applications such as generation and editing of 2D (Abdal et al., 2022; Gal et al., 2022; Patashnik et al., 2021) and 3D images (Hyung et al., 2023; Canfes et al., 2023; Hong et al., 2022a; Jain et al., 2022; Kobayashi et al., 2022; Michel et al., 2022; Sanghi et al., 2022; Wang et al., 2022a), where text-guided modifications are required, also benefit significantly from CLIP-guided KD.

**Other Applications**  The range of tasks addressed by KD in VLMs, particularly CLIP, is broad, including person re-identification (Liu et al., 2024c), zero-shot human-object interaction (HOI) detection (Wan & Tuytelaars, 2024), open-vocabulary out-of-distribution classification (Li et al., 2023f), domain generalization (Huang et al., 2023c), video-language retrieval (Pei et al., 2023), affordance detection (Cuttano et al., 2024), video highlight detection (Han et al., 2024a), and monocular depth estimation (Son & Lee, 2024). These applications demonstrate how KD transforms large-scale VLMs into accessible, efficient tools for specialized needs, with some works (Yang et al., 2023a; Nair, 2024; Wu et al., 2023b; Dong et al., 2023b) tackling multiple tasks simultaneously.

### 6.2.2 Segment Anything (SAM)

The Segment Anything Model (SAM) (Kirillov et al., 2023) is a versatile and general-purpose model designed for efficient and accurate object segmentation across diverse image domains. Developed with a focus on flexibility, SAM employs a powerful vision-transformer architecture that can segment any object in an image based on prompts, such as points, boxes, or masks. Its pre-trained nature enables SAM to generalize well across tasks without task-specific fine-tuning, making it a valuable resource for many applications. Some works (Zhang et al., 2024e; Hetang et al., 2024; Rahman et al., 2024) leverage SAM's pre-trained knowledge by distilling its segmentation capabilities into lighter, task-specific models, enabling the transfer of its robust segmentation performance to applications with constrained data annotations, computational resources or domain-specific requirements.

**SAM for Weakly Supervised Semantic Segmentation**  SAM is utilized for weakly supervised semantic segmentation (Kweon & Yoon, 2024; Jiang & Yang, 2023; Chen & Sun, 2023; Chen et al., 2023c; Sun et al., 2023a) by leveraging its pre-trained capabilities to generate high-quality pseudo-labels from minimal or weak supervision signals, such as image-level tags, points, or bounding boxes. These pseudo-labels serve as a foundation for training more efficient models, bridging the gap between limited supervision and fully labeled data. The robustness of SAM's outputs allows semantic segmentation models to learn complex patterns and

object boundaries without extensive manual annotation. This approach significantly reduces labeling costs while maintaining competitive performance.

### 6.2.3 Multi-model Applications

In some instances, research has explored synergistic use of multiple FMs, combining their strengths through KD to enhance performance in segmentation, detection, and multi-modal applications. This collaborative use of KD fosters the development of unified models that inherit the complementary strengths of various teacher architectures. General studies have demonstrated the effectiveness of integrating diverse FMs across tasks (Yin & Jiang, 2024; Ranzinger et al., 2024; Sun et al., 2023c; Ye et al., 2023a; Shang et al., 2024; Gao et al., 2024; Englert et al., 2024; Rai et al., 2024), highlighting improvements in both efficiency and accuracy. One common approach involves combining models with complementary strengths. For instance, (SAM + CLIP) (Wang et al., 2024d; Aleem et al., 2024; Yang & Gong, 2024) leverages SAM's robust image segmentation alongside CLIP's semantic understanding, enhancing performance in various vision tasks. Similarly, (DINO + CLIP) (Wysoczańska et al., 2024b; Kerr et al., 2023) has been explored, taking advantage of DINO's self-supervised learning capabilities and CLIP's strong vision-language alignment. Extending this idea, (DETR + CLIP) has been proposed in Jiang et al. (2024b) for open-vocabulary object detection, where DETR serves as the detection model while CLIP provides text-based prompts. Additionally, research (Ren et al., 2024; Das et al., 2024) has investigated using the complementary strengths of (SAM + GDINO) for object detection and segmentation.

### 6.2.4 Vision Transformers

Vision Transformers (ViTs) leverage the self-attention mechanism to capture long-range dependencies in images, enabling them to model complex and global visual features. This capability makes them highly effective for tasks such as image classification, segmentation, and detection. However, the high computational cost of training and deploying ViTs highlights the need for distilling the knowledge of pre-trained transformer models for more specialized tasks. The following section discusses prominent ViTs and their distillation methods.

**Vision Transformer** ViT (Dosovitskiy, 2020) is a deep learning model that extends the transformer architecture (Vaswani, 2017), originally developed for natural language processing, to image analysis. Several works (Lin et al., 2023a; Ren et al., 2023a; Shang et al., 2023a; Kang et al., 2023b; Bai et al., 2023; Zhao et al., 2023a; Yang et al., 2024) have explored the application of KD to ViT, investigating a range of strategies aimed at achieving diverse objectives, as summarized in Table 9.

Table 9: Summary of Methods Applying Knowledge Distillation to the Vision Transformer Model.

| Method | Contribution |
|---|---|
| SMKD (Lin et al., 2023a) | Explores the tendency of ViTs to overfit and degrade in performance under few-shot learning, due to the lack of CNN-like inductive biases, and proposes a KD method to address this issue. |
| TinyMIM (Ren et al., 2023a) | Explores KD techniques for transferring the benefits of large MIM (Masked Image Modeling) pre-trained models to smaller ones. |
| Incrementer (Shang et al., 2023a) | Introduces a class-incremental semantic segmentation framework that employs ViT instead of traditional CNNs, emphasizing the need for global context modeling and incorporating convolutional layers to support new class predictions in incremental learning settings. |
| CST (Kang et al., 2023b) | Uses self-distillation by leveraging attention maps from a frozen ViT to generate pseudo-labels for few-shot, weakly supervised tasks. |
| DMAE (Bai et al., 2023) | Suggests distilling intermediate features, rather than logits, from a large ViT model to a smaller one. |
| CSKD (Zhao et al., 2023a) | Presents a method for enhancing ViT performance through KD from CNN models. |
| ViTKD (Yang et al., 2024) | Introduces a method for distilling knowledge between ViT feature maps instead of logits, using DeiT (Touvron et al., 2021) as the base model. |

**Distillation with No Labels (DINO)** DINO (Caron et al., 2021) is a self-supervised vision transformer that learns rich visual representations from unlabeled data. Due to its strong ability to capture meaningful features, many works leverage Dino's pretrained features for object localization for various downstream

tasks, such as unsupervised object discovery (Kara et al., 2024; Siméoni et al., 2021; 2023; Wang et al., 2023n; 2022b; Liu et al., 2024f), open-world counting (Amini-Naieni et al., 2024), and zero-shot semantic segmentation (Karazija et al., 2023).

**DEtection TRansformer (DETR)**   DETR (Carion et al., 2020) is an end-to-end object detection model built on ViT, capable of directly predicting bounding boxes and class labels within a unified framework, thereby eliminating the need for anchors or complex post-processing. Its ability to effectively capture global context and model interactions between objects has positioned it as a leading backbone in object detection tasks. However, DETR's large model size and substantial computational demands pose significant challenges for deployment in real-world scenarios with limited computational budgets. To overcome these limitations, recent works (Chang et al., 2023; Wang et al., 2024m) have explored compressing DETR through KD, aiming to retain its strong performance while reducing computational overhead for real-time and resource-limited applications.

**Swin Transformer**   Swin Transformer (Liu et al., 2021e) is a powerful architecture that effectively combines the strengths of CNNs and ViTs through its hierarchical design and efficient window-based attention mechanism. By employing local window attention within each window and shifting the windows between layers, Swin captures both local and global context, enabling it to excel in modeling fine-grained details as well as long-range dependencies. This rich feature representation has made Swin a preferred choice in various computer vision tasks. Recent works (Ahmadi & Kasaei, 2024; Zhang et al., 2021a; Cheng et al., 2021; Liu et al., 2021f; Lin et al., 2022b; Shi et al., 2022; Li et al., 2022f; Tang et al., 2022b; Rasoulian et al., 2023; Heidari et al., 2023; Chen & Mo, 2023; Wang et al., 2023f; Chen et al., 2024h) have leveraged Swin Transformer as an encoder, utilizing its features across different vision tasks, as summarized in Table 10.

Table 10: Summary of Methods Applying Knowledge Distillation to the Swin Transformer Model.

| Method | Task |
|---|---|
| MaskFormer (Cheng et al., 2021) | Semantic Segmentation |
| DFR (Zhang et al., 2021a) | Weakly Supervised Semantic Segmentation (Scribbles) |
| SwinNet (Liu et al., 2021f) | Salient Object Detection |
| SwinTrack (Lin et al., 2022b) | Siamese Tracking |
| SwinF (Li et al., 2022f) | Target Detection |
| Swin-UNETR (Tang et al., 2022b) | Medical Image Analysis |
| Ssformer (Shi et al., 2022) | Semantic Segmentation |
| HGI-SAM (Rasoulian et al., 2023) | Weakly Supervised Intracranial Hemorrhage Segmentation |
| HiFormer (Heidari et al., 2023) | Medical Image Segmentation |
| Swin-Fusion (Chen & Mo, 2023) | Human Action Recognition |
| P.Swin (Wang et al., 2023f) | Image Classification and Object Detection |
| SCUNet++ (Chen et al., 2024h) | Pulmonary Embolism CT Image Segmentation |
| SWTformer (Ahmadi & Kasaei, 2024) | Weakly Supervised Semantic Segmentation (Class Labels) |

## 6.3   Distillation in Self-supervised learning

Self-Supervised Learning (SSL) is an important field due to its independence from labeled data, and self-distillation is the basis of the most prominent works in SSL, especially in the realm of computer vision. The most prominent work in SSL was based on self-distillation (Chen et al., 2020c). SimCLR (Chen et al., 2020c) proposed a shared network that receives two different augmentations of an image and minimizes their extracted embeddings. Following SimCLR, different extensions have been proposed (He et al., 2020; Chen et al., 2020d; Grill et al., 2020; Misra & Maaten, 2020; Caron et al., 2020; Xu et al., 2021). MOCO (He et al., 2020; Chen et al., 2020d) and BYOL (Grill et al., 2020) are among successful methods where they propose to use the same network but with a different training process. The student is trained using backpropagation, and the teacher is an EMA of the student. SwAP (Caron et al., 2020) uses some prototypes and assigns a code to the augmentations of the same image. The student is then trained to predict the same class ID as

the teacher. DINO (Caron et al., 2021) and DINOv2 (Oquab et al., 2023) are the most recent important applications of KD in SSL, where global and local patches of an image pass through similar teacher and student networks. The teacher is updated using EMA, and the representations of the teacher and the student are forced to be similar. This local-to-global approach makes it ideal for segmentation tasks.

Recently, more complex forms of SSL based on KD have been proposed (Misra & Maaten, 2020; Xu et al., 2021; Wang et al., 2023l; Song et al., 2023a; Gao et al., 2022c; Gu et al., 2021a). BINGO (Xu et al., 2021) uses a pre-trained SSL method to group the images of a dataset into a bag of samples. Then, in each epoch, samples are taken from the bag and the inter and intra-sample losses are minimized. Wang et al. (2023l) trains two SSL teachers for images and videos using masked modeling and follows a scenario similar to SimCLR, minimizing the spatial features with the image teacher and the spatio-temporal features with the video teacher. Song et al. (2023a) uses a parallel SSL approach and defines consistency losses between them, and Gao et al. (2022c) integrates a known KD approach into an SSL framework using an SSL pre-trained teacher.

Apart from the mentioned papers, some studies attempt to distill the knowledge of a pre-trained SSL network into a smaller network (Abbasi Koohpayegani et al., 2020; Tejankar et al., 2021; Navaneet et al., 2022; Fang et al., 2021; Dadashzadeh et al., 2022). The gap between two similar networks, one trained fully-supervised and one trained self-supervised, is shown to decrease with a bigger model size (Abbasi Koohpayegani et al., 2020). Navaneet et al. (2022) shows the impact of an MLP layer after representations of the student, while Abbasi Koohpayegani et al. (2020) proposes a similarity-based distillation method for distilling the knowledge of a pre-trained network, which can outperform the corresponding fully-supervised trained network. In a similar approach, SEED (Fang et al., 2021) uses a queue of samples and minimizes the distance of the student's representation with anchors in the queue and the corresponding distances between the anchors and the teacher's representation. Dadashzadeh et al. (2022) proposes a novel approach to complement self-supervised pretraining via an auxiliary pretraining phase for small unlabeled datasets.

## 6.4 Diffusion Distillation

Diffusion models (Ho et al., 2020; Song et al., 2020) generate high-quality images, text, and 3D data through an iterative denoising process that can require hundreds or thousands of steps. This is computationally expensive, limiting deployment on low-resource hardware. KD mitigates this by transferring soft outputs and internal representations from a large teacher model to a smaller student model. The distilled model then achieves the same quality with far fewer sampling steps, requiring significantly less inference time and computation (Luo, 2023).

Several strategies have been developed to accelerate diffusion models. Feature-level methods have the student mimic the teacher's internal representations (Huang et al., 2023b; Hsiao et al., 2024; Feng et al., 2024b; Zhang et al., 2024b). Sampling process distillation reduces the iterative denoising by either merging multiple steps into one or matching a one-step student's output to that of the full teacher (Salimans & Ho, 2022; Yin et al., 2024c; Zhou et al., 2024e). Adversarial and data-free approaches align outputs using GAN losses or bootstrapping without requiring large synthetic datasets (Sauer et al., 2024; Mekonnen et al., 2024; Xie et al., 2024b). Consistency models reformulate the denoising into a direct, often single-step transformation, cutting down inference time (Song et al., 2023c; Song & Dhariwal, 2023; Geng et al., 2024; Luo et al., 2023a; Xie et al., 2024a; Lu & Song, 2024). In the following sections, each category is explained in detail.

**Feature-level:** Feature-level distillation accelerates diffusion model efficiency by bringing the student representations closer to the teacher and reducing inference steps. Recent methods reinforce feature alignment through denoising (Huang et al., 2023b), external guidance (Hsiao et al., 2024), and cross-sample consistency (Feng et al., 2024b), which facilitate convergence and improved sampling efficiency. Furthermore, combining feature distillation with model compression reduces model size and inference time (Zhang et al., 2024b). These techniques accelerate generation without compromising output quality, thereby making diffusion models more realistic.

**Sampling Process:** Distillation of the sampling procedure is performed to ease the naturally iterative denoising process characteristic of diffusion models. Progressive distillation (Salimans & Ho, 2022) achieves this by consolidating two steps into a single step; through repeated application of this simplification, the

duration of the sampling procedure can be drastically shortened. Distribution matching distillation (Yin et al., 2024c) attempts to condense the entire iterative procedure into a single step. This alignment is accomplished by matching the output distribution of a one-step student model directly with that of the combined multi-step teacher model.

**Adversarial and Data-free:** Adversarial methods (Sauer et al., 2024; Mekonnen et al., 2024) leverage GAN-based losses to enforce the student's output distribution to closely match that of the teacher. Alternatively, data-free methods like EM distillation (Xie et al., 2024b), employ bootstrapping between successive denoising steps, eliminating the need for large synthetic datasets during training.

**Consistency Models:** Consistency models (Song et al., 2023c; Song & Dhariwal, 2023; Geng et al., 2024) reformulate the denoising process as a direct, often single transformation from noisy data to its clean version. Their latent-space variants (Luo et al., 2023a; Xie et al., 2024a), along with improvements focusing on stability and scalability (Lu & Song, 2024), achieve significant speedups in inference times. However, training these models may require careful tuning to prevent quality collapse.

Furthermore, recent developments in the area of distillation techniques expanded their accessibility to multi-modal and multi-model settings, thus improving transferability across tasks and diverse architectures. ProlificDreamer (Wang et al., 2023q) and DREAMFUSION (Poole et al., 2022) utilize the methods in the realm of text-to-3D translation, and general-purpose systems like Diff-Instruct (Luo et al., 2023b) can facilitate generalization for the task.

In summary, various approaches offer distinct benefits: Sampling process distillation drastically accelerates generation but tends to require cumbersome training in order to maintain sample quality. Adversarial and data-free methods provide more flexibility, though sometimes at the cost of training stability. Consistency models enable low-latency generation, provided they are well-calibrated. Furthermore, combining cross-modal and universal approaches significantly expands the applicability of these distillation techniques across a wide range of generation tasks.

### 6.5 Visual Recognition Distillation

Knowledge distillation has been widely applied to computer vision tasks, enabling the training of compact student models replicating the behavior of larger, resource-intensive teacher models. Key applications of KD in visual recognition are summarized in Table 11, based on two main factors: the type of knowledge transferred (feature-based, logit-based, similarity-based, and combinations) and the distillation scheme employed (offline, online, and self-distillation). Several important visual recognition tasks are covered in this table, including depth estimation, face recognition, image segmentation, medical imaging, object detection, object tracking, pose estimation, super resolution, action recognition, and image retrieval.

Table 11: Summary of knowledge distillation methods in visual recognition tasks based on their sources and schemes of distillation.

| Task | Knowledge | Scheme | Reference |
|---|---|---|---|
| Depth Estimation | feature | offline | (López-Cifuentes et al., 2023) |
| | | self-distillation | (Zhou & Dong, 2022) |
| | logit | offline | (Hu et al., 2023b), (Xu et al., 2023c), (Shi et al., 2023), (Ka et al., 2024), (Wang & Liu, 2024) |
| | | self-distillation | (Peng et al., 2021), (Zhao et al., 2023b), (Marsal et al., 2024), (Hu et al., 2024a), (Boutros et al., 2024), (Dong et al., 2024c) |
| | similarity | offline | (Chen et al., 2024e) |
| | feature + logit | offline | (Wu et al., 2023f) |
| | | online | (Shao et al., 2024b) |

Table 11 – Continued

| Task | Knowledge | Scheme | Reference |
|---|---|---|---|
| Face Recognition | feature | offline | (Shin et al., 2022b), (Boutros et al., 2024), (Otroshi-Shahreza et al., 2023), (Neto et al., 2024), (Caldeira et al., 2024), (Li et al., 2023b), (Huang et al., 2024f), (Babnik et al., 2024) |
| | logit | offline | (Jung et al., 2021), (Liu et al., 2021a), (Xu et al., 2023a) |
| | similarity | offline | (Huang et al., 2022d) |
| | | self-distillation | (Di et al., 2024) |
| Image Segmentation | feature | offline | (Ji et al., 2022), (Yang et al., 2022f), (Kang et al., 2023a), (Shang et al., 2023a), (Yuan et al., 2024a), (Liu et al., 2024g), (Shu et al., 2021) |
| | | self-distillation | (Wu et al., 2024b) |
| | logit | offline | (Amirkhani et al., 2021), (Xu et al., 2023c), (Li et al., 2024c) |
| | | online | (Shen et al., 2023), (Berrada et al., 2024) |
| | similarity | offline | (Yang et al., 2022c), (Mansourian et al., 2025) |
| | | online | (Gao et al., 2022a) |
| | feature + logit | offline | (Tian et al., 2022c), (Phan et al., 2022), (Xiao et al., 2023a), (Zheng et al., 2025) |
| | | online | (Zhu et al., 2023a) |
| | feature + similarity | offline | (Feng et al., 2021), (Liu et al., 2024d) |
| | | online | (Wang et al., 2024p) |
| Medical Imaging | feature | offline | (Qin et al., 2021), (Li et al., 2022d), (Ghosh et al., 2023), (Wang et al., 2023g), (Cao et al., 2024), (Nasser et al., 2024), (Chen et al., 2024d), (Zhou et al., 2024c) |
| | | online | (Dong et al., 2025) |
| | | self-distillation | (Ye et al., 2022), (Yi et al., 2024a) |
| | logit | offline | (Feng et al., 2024c), (Wang et al., 2024l) |
| | | online | (Xie et al., 2023c) |
| | | self-distillation | (Zhong et al., 2023), (Huang et al., 2024b) |
| | similarity | online | (Xing et al., 2021) |
| | | self-distillation | (You et al., 2022), (Sharma et al., 2024) |
| | feature + logit | offline | (El-Assiouti et al., 2024) |
| | feature + similarity | offline | (Liu et al., 2022c), (Qi et al., 2024) |

Table 11 – Continued

| Task | Knowledge | Scheme | Reference |
|------|-----------|--------|-----------|
| Object Detection | feature | offline | (Dai et al., 2021b), (Guo et al., 2021a), (Guo et al., 2021b), (Kang et al., 2021), (Yang et al., 2022a), (Jang et al., 2022), (Yang et al., 2022f), (Lao et al., 2023), (Lan & Tian, 2024), (Wang et al., 2023i), (Liu et al., 2024i) |
| | | self-distillation | (Wu et al., 2023a), (Jia et al., 2024), (Wu et al., 2024b), (Zheng et al., 2023) |
| | logit | offline | (Yang et al., 2023e), (Jin et al., 2023a), (Li et al., 2023c) |
| | | self-distillation | (Zhou et al., 2023a) |
| | | online | (Nguyen et al., 2022) |
| | similarity | offline | (Ni et al., 2023) |
| | feature + logit | offline | (Tang et al., 2022a), (Du et al., 2021) |
| | | online | (Zhang & Ma, 2021), (Yao et al., 2021), (Yang et al., 2022e), (Li et al., 2022b), (Zhang & Ma, 2023), (Zhu et al., 2023c) |
| | feature + similarity | offline | (Zhang & Ma, 2021), (Yao et al., 2021), (Yang et al., 2022e), (Li et al., 2022b), (Zhang & Ma, 2023), (Zhu et al., 2023c) |
| | | self-distillation | (Zhang et al., 2022c) |
| | logit + similarity | offline | (Wang et al., 2024f) |
| Object Tracking | feature | offline | (Hou et al., 2024), (Zhang et al., 2024h) |
| | logit | offline | (Valverde et al., 2021) |
| | feature + similarity | offline | (Wang et al., 2024k) |
| Pose Estimation | feature | offline | (Nie et al., 2019) |
| | | self-distillation | (Chen et al., 2024f) |
| | logit | offline | (Guo et al., 2023b), (Ye et al., 2023b), (Wang et al., 2021c), (Aghli & Ribeiro, 2021), (Xu et al., 2022c), (Zhang et al., 2019a) |
| | | online | (Li et al., 2021b) |
| | | self-distillation | (Li & Lee, 2021) |
| | feature + logit | offline | (Fang et al., 2023b), (GUAN et al., 2023) |
| | | self-distillation | (Yang et al., 2023h) |
| Super Resolution | feature | offline | (Jiang et al., 2024d), (Liu et al., 2024b) |
| | | self-distillation | (Wang et al., 2024e), (Wang & Zhang, 2024) |
| | logit | offline | (Noroozi et al., 2024) |
| | | self-distillation | (Park, 2024) |
| | relation | offline | (Angarano et al., 2022), (Zhang et al., 2024j) |
| | feature + similarity | offline | (Xie et al., 2023a), (Fang et al., 2023a) |
| | | self-distillation | (Wang et al., 2021f) |
| | feature + logit | offline | (Zhang et al., 2021g), (Yu et al., 2023), (Wang et al., 2024n) |
| Action Recognition | feature | offline | (Zhao et al., 2022d), (Wang et al., 2024c), (Lee et al., 2023b), (Huang et al., 2024d), (Hong et al., 2021) |
| | | online | (Guo et al., 2023a) |
| | logit | offline | (Radevski et al., 2023) |
| | feature + similarity | offline | (Dai et al., 2021a), (Bian et al., 2021) |

Table 11 – Continued

| Task | Knowledge | Scheme | Reference |
|---|---|---|---|
| | feature | offline | (Jin et al., 2020), (Porrello et al., 2020), (Budnik & Avrithis, 2021) |
| Image Retrieval | similarity | offline | (Wu et al., 2022), (Xu et al., 2024a), (Tian et al., 2021), (Xie et al., 2024d) |
| | feature + logit | offline | (Xie et al., 2023b) |
| | feature + similarity | offline | (Suma & Tolias, 2023), (Xie et al., 2024c) |

## 7   Performance Comparison

KD enables the compression of a deep network into a smaller model while preserving strong performance. Due to the vast number of methods proposed in this field, a comprehensive comparison across existing methods in different settings is necessary. However, the complicated nature of KD, encompassing various schemes, algorithms, sources, and teacher-student architectures, makes it challenging to conduct a comprehensive and fair comparison of all existing methods. For example, the exact pre-trained teacher model along with factors such as the number of epochs and batch size, can influence the results of the distillation. These factors are highly dependent on the hardware resources employed in each method, which is the determining factor in the final results.

In this work, existing methods are compared for two well-known tasks: classification and semantic segmentation segmentation, as almost all the proposed distillation methods evaluate their results on these two tasks. For classification, the ImageNet and CIFAR-100 datasets, and for semantic segmentation, the Cityscapes and PascalVOC datasets are used, as these two datasets are widely employed in existing methods.

ImageNet is a large-scale dataset designed for visual object recognition tasks, consisting of over 1.2 million training images, 50,000 validation images, and 100,000 testing images across 1,000 object classes. CIFAR-100 is composed of $32 \times 32$ images taken from 100 classes, with 50,000/10,000 images for training/testing, and each class has the same number of training and testing images. Cityscapes is designed for understanding urban scenes and includes 2,975/500/1,525 images for training/validation/testing in 19 classes. PascalVOC dataset comprises 1464/1449/1456 images for train/val/test, with 21 classes.

Table 12: Performance comparison of different knowledge distillation methods on the CIFAR-100 classification dataset (* denotes that results are not reported from the original paper).

| Method | Knowledge | Teacher (baseline) | Student (baseline) | Accuracy | Code |
|---|---|---|---|---|---|
| FitNet (Romero et al., 2014) | Feature | WRN-40-2 (75.61) | WRN-40-1 (71.98) | 72.24 (+0.26)* | Link |
| | | ResNet32×4 (79.42) | ShuffleNetV2 (71.82) | 73.54 (+1.72) | |
| KD (Hinton, 2015) | Logit | WRN-40-2 (75.61) | WRN-40-1 (71.98) | 73.54 (+1.56)* | Link |
| | | ResNet32×4 (79.42) | ShuffleNetV2 (71.82) | 74.45 (+2.63) | |
| AT (Zagoruyko & Komodakis, 2016) | Feature | WRN-40-2 (75.61) | WRN-40-1 (71.98) | 72.77 (+1.79)* | Link |
| | | ResNet32×4 (79.42) | ShuffleNetV2 (71.82) | 72.73 (+0.91) | |
| CCKD (Peng et al., 2019) | Similarity | ResNet-110 (−) | ResNet-20 (68.40) | 72.40 (+4.00) | − |
| SPKD (Tung & Mori, 2019) | Similarity | WRN-40-2 (75.61) | WRN-40-1 (71.98) | 72.43 (+1.45)* | − |
| | | ResNet32×4 (79.42) | ShuffleNetV2 (71.82) | 74.56 (+2.74) | |
| CRD (Tian et al., 2019) | Feature | WRN-40-2 (75.61) | WRN-40-1 (71.98) | 74.14 (+2.16) | Link |
| | | ResNet32×4 (79.42) | ShuffleNetV2 (71.82) | 75.65 (+3.83) | |
| OFD (Heo et al., 2019a) | Feature | WRN-40-2 (75.61) | WRN-40-1 (71.98) | 74.33 (+2.35) | Link |
| | | ResNet32×4 (79.42) | ShuffleNetV2 (71.82) | 76.82 (+5) | |
| Chen et al. (Chen et al., 2020f) | Similarity + Logit | ResNet-101 (71.77) | ResNet-18 (65.64) | 69.14 (+3.49) | Link |
| TAKD (Mirzadeh et al., 2020) | Logit | ResNet-110 (−) | ResNet-8 (−) | 61.82 | Link |
| | | ResNet32×4 (79.42) | ShuffleNetV2 (71.82) | 74.82 (+3) | |
| SphericalKD (Guo et al., 2020a) | Logit | ResNet-32×4 (79.42) | ResNet-8×4 (72.50) | 76.40 (+3.90) | Link |
| RKD (Chen et al., 2021d) | Feature | WRN-40-2 (75.61) | WRN-40-1 (71.98) | 75.09 (+3.11) | Link |
| | | ResNet32×4 (79.42) | ShuffleNetV2 (71.82) | 77.78 (+5.96) | |
| SemCKD (Chen et al., 2021a) | Feature + Logit | WRN-40-2 (75.61) | MobileNetV2 (65.43) | 69.67 (+4.24) | Link |
| | | ResNet32×4 (79.42) | ShuffleNetV2 (72.60) | 77.62 (+5.02) | |
| ICKD (Liu et al., 2021c) | Similarity + Logit | WRN-40-2 (75.61) | WRN-40-1 (71.98) | 74.63 (+2.65) | Link |
| DKD (Zhao et al., 2022a) | Logit | WRN-40-2 (75.61) | WRN-40-1 (71.98) | 74.81 (+2.83) | Link |
| | | ResNet32×4 (79.42) | ShuffleNetV2 (71.82) | 77.07 (+5.25) | |
| DistKD (Huang et al., 2022a) | Similarity + Logit | WRN-40-2 (75.61) | WRN-40-1 (71.98) | 74.73 (+2.75) | Link |
| | | ResNet32×4 (79.42) | ShuffleNetV2 (71.82) | 77.35 (+5.53) | |
| Norm (Liu et al., 2023a) | Feature | WRN-40-2 (75.61) | WRN-40-1 (71.98) | 74.82 (+2.84) | Link |
| | | ResNet32×4 (79.42) | ShuffleNetV2 (71.82) | 78.07 (+6.25) | |
| SFKD (Yuan et al., 2024b) | Logit | ResNet-32×4 (79.55) | ResNet-8×4 (72.50) | 76.84 (+4.34) | − |
| | | ResNet32×4 (79.42) | ShuffleNetV2 (71.82) | 76.96 (+5.14) | |
| NormKD (Chi et al., 2023) | Logit | ResNet-32×4 (79.42) | ResNet-8×4 (72.50) | 76.57 (+4.07) | Link |
| | | ResNet32×4 (79.42) | ShuffleNetV2 (71.82) | 76.01 (+4.19) | |
| CRLD (Zhang et al., 2024g) | Similarity + Logit | WRN-40-2 (75.61) | WRN-40-1 (71.98) | 75.58 (+3.60) | Link |
| | | ResNet32×4 (79.42) | ShuffleNetV2 (71.82) | 78.27 (+6.45) | |
| NTCE-KD (Li et al., 2024a) | Logit | WRN-40-2 (75.61) | WRN-40-1 (71.98) | 76.44 (+4.46) | − |
| | | ResNet32×4 (79.42) | ShuffleNetV2 (71.82) | 79.43 (+7.61) | |
| TTM (Zheng & Yang, 2024) | Logit | WRN-40-2 (75.61) | WRN-40-1 (71.98) | 74.58 (+2.60) | Link |
| | | ResNet32×4 (79.42) | ShuffleNetV2 (71.82) | 76.57 (+4.75) | |
| Miles et al. (Miles & Mikolajczyk, 2024) | Logit | WRN-40-2 (76.46) | WRN-16-2 (73.64) | 76.14 (+2.50) | Link |
| | | ResNet32×4 (79.63) | ShuffleNetV2 (73.12) | 79.06 (+5.94) | |
| SDD (Wei et al., 2024b) | Logit | ResNet-32×4 (79.42) | MobileNetV2 (64.6) | 68.84 (+4.24) | Link |
| | | ResNet50 (79.34) | MobileNetV2 (64.60) | 71.46 (+6.86) | |
| Sun et al. (2024a) | Logit | WRN-40-2 (75.61) | ResNet-8×4 (72.50) | 77.11 (+3.14) | Link |
| | | ResNet32×4 (79.42) | ShuffleNetV2 (71.82) | 75.56 (+3.74) | |
| InDistill (Sarridis et al., 2025) | Feature | WRN-40-2 (75.61) | WRN-40-1 (71.98) | 75.09 (+3.11) | Link |
| IKD (Wang et al., 2025a) | Logit | WRN-40-2 (75.61) | WRN-40-1 (71.98) | 74.84 (+2.86) | − |
| | | ResNet50 (79.34) | MobileNetV2 (64.60) | 70.49 (+5.89) | |
| ADG-KD (Li et al., 2025b) | Logit | WRN-40-2 (75.61) | WRN-40-1 (71.98) | 75.84 (+3.86) | − |
| | | ResNet32×4 (79.42) | ShuffleNetV2 (71.82) | 78.96 (+7.14) | |
| TeKAP (Hossain et al., 2025) | Feature + Logit | WRN-40-2 (75.61) | WRN-40-1 (71.98) | 74.41 (+2.43) | Link |
| | | ResNet32×4 (74.64) | ShuffleNetV2 (70.36) | 77.38 (+7.02) | |

Table 13: Performance comparison of different knowledge distillation methods on the ImageNet classification dataset (* denotes that results are not reported from the original paper).

| Method | Knowledge | Teacher (baseline) | Student (baseline) | Accuracy | Code |
|---|---|---|---|---|---|
| FitNet (Romero et al., 2014) | Logit | ResNet-34 (73.31) | ResNet-18 (69.75) | 70.31 (+0.56)* | Link |
| KD (Hinton, 2015) | Logit | ResNet-34 (73.31) | ResNet-18 (69.75) | 70.62 (+0.87)* | Link |
| | | ResNet-50 (76.16) | MobileNetV1 (68.87) | 68.58 (-0.29)* | |
| AT (Zagoruyko & Komodakis, 2016) | Feature | ResNet-34 (73.31) | ResNet-18 (69.75) | 70.30 (+0.55)* | Link |
| | | ResNet-50 (76.16) | MobileNetV1 (68.87) | 69.56 (+0.69)* | |
| CRD (Tian et al., 2019) | Feature | ResNet-34 (73.31) | ResNet-18 (69.75) | 71.17 (+1.42) | Link |
| | | ResNet-50 (76.16) | MobileNetV1 (68.87) | 71.37 (+2.50) | |
| OFD (Heo et al., 2019a) | Feature | ResNet-34 (73.31) | ResNet-18 (69.75) | 70.81 (+1.06) | Link |
| | | ResNet-50 (76.16) | MobileNetV1 (68.87) | 71.25 (+2.38) | |
| SphericalKD (Guo et al., 2020a) | Logit | ResNet-34 (73.31) | ResNet-18 (69.75) | 72.30 (+2.55) | Link |
| RKD (Chen et al., 2021d) | Feature + Logit | ResNet-34 (73.31) | ResNet-18 (69.75) | 71.61 (+1.86) | Link |
| | | ResNet-50 (76.16) | MobileNetV1 (68.87) | 72.56 (+3.69) | |
| SemCKD (Chen et al., 2021a) | Feature + Logit | ResNet-34 (73.31) | ResNet-18 (69.75) | 70.87 (+1.12) | Link |
| ICKD (Liu et al., 2021c) | Similarity + Logit | ResNet-34 (73.31) | ResNet-18 (69.75) | 72.19 (+2.44) | Link |
| DKD (Zhao et al., 2022a) | Logit | ResNet-34 (73.31) | ResNet-18 (69.75) | 72.05 (+2.30) | Link |
| | | ResNet-50 (76.16) | MobileNetV1 (68.87) | 72.05 (+3.18) | |
| DistKD (Huang et al., 2022a) | Similarity + Logit | ResNet-34 (73.31) | ResNet-18 (69.75) | 72.07 (+2.32) | Link |
| | | ResNet-50 (76.16) | MobileNetV1 (70.13) | 73.24 (+3.11) | |
| CAT-KD (Guo et al., 2023d) | Feature | ResNet-34 (73.31) | ResNet-18 (69.75) | 71.26 (+1.51) | Link |
| | | ResNet-50 (76.16) | MobileNetV1 (70.13) | 72.24 (+3.37) | |
| NKD (Yang et al., 2023g) | Logit | ResNet-34 (73.31) | ResNet-18 (69.75) | 71.96 (+2.21) | Link |
| | | ResNet-50 (76.16) | MobileNetV1 (70.13) | 72.58 (+3.71) | |
| MLLD (Jin et al., 2023a) | Logit | ResNet-34 (73.31) | ResNet-18 (69.75) | 71.90 (+2.15) | Link |
| | | ResNet-50 (76.16) | MobileNetV1 (70.13) | 73.01 (+4.14) | |
| Norm (Liu et al., 2023a) | Feature | ResNet-34 (73.31) | ResNet-18 (69.75) | 72.14 (+2.39) | Link |
| | | ResNet-50 (76.16) | MobileNetV1 (68.87) | 74.26 (+5.39) | |
| SFKD (Yuan et al., 2024b) | Logit | ResNet-34 (73.31) | ResNet-18 (69.75) | 71.86 (+2.11) | – |
| | | ResNet-50 (76.16) | MobileNetV1 (68.87) | 72.24 (+3.37) | |
| NormKD (Chi et al., 2023) | Logit | ResNet-34 (73.31) | ResNet-18 (69.75) | 72.03 (+2.28) | Link |
| | | ResNet-50 (76.16) | MobileNetV1 (68.87) | 72.79 (+3.92) | |
| CRLD (Zhang et al., 2024g) | Similarity + Logit | ResNet-34 (73.31) | ResNet-18 (69.75) | 72.37 (+2.62) | Link |
| | | ResNet-50 (76.16) | MobileNetV1 (70.13) | 73.53 (+4.66) | |
| NTCE-KD (Li et al., 2024a) | Logit | ResNet-34 (73.31) | ResNet-18 (69.75) | 73.12 (+3.37) | – |
| | | ResNet-50 (76.16) | MobileNetV1 (70.13) | 74.90 (+6.03) | |
| TTM (Zheng & Yang, 2024) | Logit | ResNet-34 (73.31) | ResNet-18 (69.75) | 72.19 (+2.44) | Link |
| | | ResNet-50 (76.16) | MobileNetV1 (70.13) | 773.09 (+4.22) | |
| SDD (Wei et al., 2024b) | Logit | ResNet-34 (73.31) | ResNet-18 (69.75) | 72.33 (+2.58) | Link |
| Sun et al. (2024a) | Logit | ResNet-34 (73.31) | ResNet-18 (69.75) | 72.08 (+2.33) | Link |
| | | ResNet-50 (76.16) | MobileNetV1 (68.87) | 73.22 (+4.35) | |
| InDistill (Sarridis et al., 2025) | Feature | ResNet-34 (73.31) | ResNet-18 (69.75) | 71.34 (+1.59) | Link |
| IKD (Wang et al., 2025a) | Logit | ResNet-34 (73.31) | ResNet-18 (69.75) | 71.82 (+2.07) | – |
| ADG-KD (Li et al., 2025b) | Logit | ResNet-34 (73.31) | ResNet-18 (69.75) | 72.22 (+2.47) | – |
| | | ResNet-50 (76.16) | MobileNetV1 (68.87) | 73.43 (+4.56) | |

Furthermore, due to the importance of distillation in LLMs nowadays, compare existing methods for distillation in LLMs are compared. This includes comparing black-box and white-box methods. However, datasets and models used in LLMs vary significantly across different papers, specifically in white-box methods. Results for the datasets and models that are more commonly used in existing methods are reported, making the comparison easier and fair. The datasets used are: GLUE, Multilingual NER, DollyEval, IMDB, GSM8K, CrossFit, CommonsenseQA, SVAMP, Universal NER Benchmark, OpenBookQA, BBH, StrategyQA, and MATH.

Tables 12 and 13 summarize various knowledge distillation methods applied to classification tasks on the CIFAR-100 and ImageNet datasets. For each method, the source of knowledge, teacher and student architectures, and post-distillation accuracy are reported. In most cases, two scenarios are considered: one where the teacher and student share similar architectures, and another where their architectures differ.

Table 14: Performance comparison of different knowledge distillation methods on the PascalVoc segmentation dataset (* denotes that results are not reported from the original paper).

| Method | Knowledge | Teacher (baseline) | Student (baseline) | Accuracy | Code |
|---|---|---|---|---|---|
| FitNet (Romero et al., 2014) | Logit | DeepLab-R50 (76.21) | DeepLab-MV2 (70.57) | 71.30 (+0.73)* | Link |
| KD (Hinton, 2015) | Logit | DeepLab-R101 (77.85) | DeepLab-R18 (67.50) | 69.13 (+1.63)* | Link |
| | | DeepLab-R101 (77.85) | DeepLab-MV2 (63.92) | 66.39 (+2.47) | |
| AT (Zagoruyko & Komodakis, 2016) | Feature | DeepLab-R101 (77.85) | DeepLab-R18 (67.50) | 68.95 (+1.09)* | Link |
| | | DeepLab-R101 (77.85) | DeepLab-MV2 (63.92) | 66.27 (+2.35) | |
| SKD (Liu et al., 2019d) | Feature + Logit | PSPNet-R101 (78.52) | PSPNet-R18 (71.35) | 72.01 (+0.66) | Link |
| | | PSPNet-R101 (77.67) | DeepLab-R18 (71.98) | 73.03 (+1.05) | |
| He et al. (2019) | Feature + Similarity | DeepLab-R50 (76.21) | DeepLab-MV2 (70.57) | 72.50 (+1.93) | – |
| IFVD (Wang et al., 2020d) | Similarity + Logit | PSPNet-R101 (78.52) | PSPNet-R18 (71.35) | 73.49 (+2.14) | Link |
| | | PSPNet-R101 (77.67) | DeepLab-R18 (71.98) | 72.87 (+0.89) | |
| CWD (Shu et al., 2021) | Feature | PSPNet-R101 (78.52) | PSPNet-R18 (71.35) | 73.49 (+2.14) | Link |
| | | PSPNet-R101 (77.67) | DeepLab-R18 (71.98) | 73.78 (+1.79) | |
| DSD (Feng et al., 2021) | Similarity | DeepLab-R101 (79.10) | DeepLab-R18 (71.9) | 73.70 (+1.80) | – |
| CIRKD (Yang et al., 2022c) | Similarity + Logit | DeepLab-R101 (77.67) | DeepLab-R18 (73.21) | 74.50 (+1.29) | Link |
| | | DeepLab-R101 (77.67) | PSPNet-R18 (73.33) | 74.78 (+1.45) | |
| DIST (Huang et al., 2022a) | Similarity + Logit | DeepLab-R101 (77.85) | DeepLab-R18 (67.50) | 69.84 (+2.34) | Link |
| | | DeepLab-R101 (77.85) | PSPNet-R18(67.40) | 68.93 (+1.53) | |
| IDD (Zhang et al., 2022d) | Similarity | PSPNet-R101 (77.86) | PSPNet-R18 (70.80) | 76.70 (+5.9) | – |
| MGD (Yang et al., 2022f) | Feature | PSPNet-R101 (78.52) | PSPNet-R18 (71.35) | 74.98 (+3.63)* | Link |
| FAKD (Yuan et al., 2024a) | Feature | PSPNet-R101 (78.52) | PSPNet-R18 (70.52) | 73.97 (+3.45) | – |
| | | DeepLab-R101 (78.62) | DeepLab-MV2 (62.56) | 67.62 (+5.06) | |
| BPKD (Liu et al., 2024d) | Logit | PSPNet-R101 (78.52) | PSPNet-R18 (71.35) | 75.74 (+4.39) | Link |
| LAD (Liu et al., 2024g) | Feature + Logit | PSPNet-R101 (78.52) | PSPNet-R18 (71.35) | 75.74 (+4.39) | – |
| | | PSPNet-R101 (78.52) | DeepLab-R18 (71.98) | 76.33 (+4.35) | |
| AttnFD (Mansourian et al., 2024) | Feature | DeepLab-R101 (77.85) | DeepLab-R18 (67.50) | 73.09 (+5.59) | Link |
| | | DeepLab-R101 (77.85) | PSPNet-R18(67.40) | 70.95 (+3.55) | |
| AICSD (Mansourian et al., 2025) | Similarity + Logit | DeepLab-R101 (77.85) | DeepLab-R18 (67.50) | 70.58 (+3.08) | Link |
| | | DeepLab-R101 (77.85) | PSPNet-R18(66.60) | 68.29 (+1.69) | |

In a similar vein, Tables 14 and 15 report the results of various methods for the semantic segmentation task on the PASCAL VOC and Cityscapes datasets. For a fair comparison, the reported results are taken directly from the original papers. However, in cases where results on the specified datasets are not provided in the original work, we report results obtained from our own implementations (denoted by *).

Along the same line, Table 16 presents a comparison of distillation methods for LLMs. It shows the compression rate and the improvement over the teacher model for each method. As can be seen, distillation in LLMs is more crucial than in other fields, as it allows for compressing large models with a high compression rate while still maintaining comparable performance. It should be noted that in cases where the exact performance of the teacher model is not reported, performance of the student model before and after distillation is reported (denoted by *).

Table 15: Performance comparison of different knowledge distillation methods on the Cityscapes segmentation dataset (* denotes that results are not reported from the original paper).

| Method | Knowledge | Teacher (baseline) | Student (baseline) | Accuracy | Code |
|---|---|---|---|---|---|
| KD (Hinton, 2015) | Logit | DeepLab-R101 (77.66) | DeepLab-R18 (64.09) | 65.21 (+1.12)* | Link |
| | | DeepLab-R101 (77.66) | DeepLab-MV2 (63.05) | 64.03 (+0.98) | |
| AT (Zagoruyko & Komodakis, 2016) | Feature | DeepLab-R101 (77.66) | DeepLab-R18 (64.09) | 65.29 (+1.20)* | Link |
| | | DeepLab-R101 (77.66) | DeepLab-MV2 (63.05) | 63.72 (+0.67) | |
| Xie et al. (2018) | Similarity + Logit | DeepLab-R101 (70.90) | DeepLab-MV2 (67.30) | 71.90 (+4.60) | – |
| SKD (Liu et al., 2019d) | Feature + Logit | PSPNet-R101 (79.76) | PSPNet-R18 (72.65) | 74.23 (+1.58) | Link |
| | | PSPNet-R101 (79.76) | DeepLab-R18 (74.96) | 75.32 (+0.36) | |
| He et al. (2019) | Feature + Similarity | DeepLab-R101 (71.40) | DeepLab-MV2 (70.20) | 72.70 (+2.50) | – |
| IFVD (Wang et al., 2020d) | Similarity + Logit | PSPNet-R101 (79.76) | PSPNet-R18 (72.65) | 74.55 (+1.90) | Link |
| | | PSPNet-R101 (79.76) | DeepLab-R18 (74.96) | 76.01 (+1.05) | |
| CWD (Shu et al., 2021) | Feature | PSPNet-R101 (79.76) | PSPNet-R18 (72.65) | 75.91 (+3.26) | Link |
| | | PSPNet-R101 (79.76) | DeepLab-R18 (74.96) | 77.13 (+2.17) | |
| DSD (Feng et al., 2021) | Similarity | DeepLab-R101 (78.23) | DeepLab-R18 (69.42) | 72.24 (+2.82) | – |
| CIRKD (Yang et al., 2022c) | Similarity + Logit | DeepLab-R101 (78.07) | DeepLab-R18 (74.21) | 76.38 (+2.27) | Link |
| | | DeepLab-R101 (78.07) | PSPNet-R18 (72.55) | 74.73 (+2.18) | |
| DIST (Huang et al., 2022a) | Similarity + Logit | DeepLab-R101 (78.07) | DeepLab-R18 (74.21) | 77.10 (+2.89) | Link |
| | | DeepLab-R101 (78.07) | PSPNet-R18 (72.55) | 76.31 (+3.76) | |
| IDD (Zhang et al., 2022d) | Similarity | PSPNet-R101 (78.50) | PSPNet-R18 (70.09) | 77.59 (+7.5) | – |
| | | PSPNet-R101 (78.50) | ESPNet (61.40) | 68.87 (+7.47) | |
| MGD (Yang et al., 2022f) | Feature | PSPNet-R101 (78.34) | PSPNet-R18 (69.85) | 73.63 (+3.78) | Link |
| | | PSPNet-R101 (78.34) | DeepLab-R18 (73.20) | 76.31 (+3.11) | |
| MLP (Liu et al., 2023c) | Feature | PSPNe-R101 (78.34) | PSPNet-R18 (69.85) | 74.51 (+4.66) | – |
| | | PSPNet-R101 (78.34) | DeepLab-R18 (73.20) | 76.55 (+3.35) | |
| DiffKD (Huang et al., 2023b) | Feature | DeepLab-R101 (78.07) | DeepLab-R18 (74.21) | 77.78 (+3.57) | Link |
| | | DeepLab-R101 (78.07) | PSPNet-R18 (72.55) | 75.83 (+3.28) | |
| FAKD (Yuan et al., 2024a) | Feature | PSPNet-R101 (79.74) | PSPNet-R18 (68.99) | 74.75 (+5.76) | – |
| | | DeepLab-R101 (80.98) | DeepLab-MV2 (70.49) | 74.08 (+3.59) | |
| BPKD (Liu et al., 2024d) | Logit | PSPNet-R101 (79.74) | PSPNet-R18 (74.23) | 77.57 (+3.34) | Link |
| | | DeepLab-R101 (80.98) | DeepLab-MV2 (75.29) | 78.59 (+3.30) | |
| LAD (Liu et al., 2024g) | Feature + Logit | PSPNet-R101 (79.76) | PSPNet-R18 (72.65) | 76.86 (+4.21) | – |
| | | PSPNet-R101 (79.76) | DeepLab-R18 (74.96) | 77.23 (+2.27) | |
| FreeKD (Zhang et al., 2024i) | Feature | PSPNet-R101 (78.34) | PSPNet-R18 (69.85) | 74.40 (+4.55) | Link |
| | | PSPNet-R101 (78.34) | DeepLab-R18 (73.20) | 76.45 (+3.25) | |
| AttnFD (Mansourian et al., 2024) | Feature | DeepLab-R101 (77.66) | DeepLab-R18 (64.09) | 73.04 (+8.96) | Link |
| | | DeepLab-R101 (77.66) | PSPNet-R18 (65.72) | 68.86 (+3.14) | |
| AICSD (Mansourian et al., 2025) | Similarity + Logit | DeepLab-R101 (77.66) | DeepLab-R18 (64.09) | 70.96 (+6.88) | Link |
| | | DeepLab-R101 (77.66) | DeepLab-MV2 (63.05) | 69.51 (+6.56) | |

Table 16: Performance comparison of different Black-box and White-box distillation methods for LLMs (* denotes that the comparison is between the student model before and after distillation).

| Method | Distillation Type | Teacher Model | Compression Rate | Benchmark | Comparison with Teacher | | Code |
|---|---|---|---|---|---|---|---|
| **White-box Distillation** | | | | | | | |
| DistillBERT (Sanh, 2019) | Logit | $BERT_{base}$ | 1.66 | GLUE | 77.00 / 79.50 | 97.0% | – |
| TinyBERT (Jiao et al., 2019) | Feature | $BERT_{base}$ | 7.50 | GLUE | 77.00 / 79.50 | 97.0% | Link |
| PKD (Sun et al., 2019) | Logit + Feature | $BERT_{base}$ | 2.00 | GLUE | 77.70 / 84.90 | 92.0% | Link |
| PD (Turc et al., 2019) | Logit | $BERT_{base}$ | 2.00 | GLUE | 82.10 / 81.70 | 100.5% | Link |
| Xtremedistil (Mukherjee & Awadallah, 2020) | Feature | $mBERT_{base}$ | 35.00 | Multilingual NER | 88.60 / 92.70 | 95.0% | Link |
| MobileBERT (Sun et al., 2020) | Feature | $BERT_{base}$ | 4.30 | GLUE | 77.70 / 78.30 | 99.0% | Link |
| MINILM (Wang et al., 2020b) | Feature | $BERT_{base}$ | 1.65 | GLUE | 80.40 / 81.50 | 98.0% | – |
| xtremedistiltransformers (Mukherjee et al., 2021) | Feature | $BERT_{base}$ | 7.70 | GLUE | 81.70 / 83.70 | 97.6% | Link |
| MetaDistil (Zhou et al., 2021b) | Feature | $BERT_{base}$ | 2.00 | GLUE | 80.40 / 80.70 | 99.0% | Link |
| TED (Liang et al., 2023a) | Feature | $DeBERTaV3_{base}$ | 2.60 | GLUE | 87.50 / 88.90 | 98.0% | Link |
| MiniLLM (Gu et al., 2024) | Feature | GPT-2 | 2.00 | DollyEval | 57.40 / 58.40 | 94.0% | Link |
| PC-LoRA (Hwang et al., 2024) | Feature | $BERT_{base}$ | 3.73 | IMDB | 92.78 / 94.00 | 98.0% | – |
| GKD (Agarwal et al., 2024) | Logit | $T5_{XL}$ | 3.75 | GSM8K | 29.10 / 29.00 | 100.3% | – |
| **Black-box Distillation** | | | | | | | |
| ILD (Huang et al., 2022e) | ICL | $GPT2_{large}$ | 6.00 | CrossFit | 61.20 / 66.20 | 92.0% | – |
| LLM-R (Wang et al., 2023h) | ICL | $LLaMA_{13B}$ | 2.00 | CommonsenseQA | 48.80 / 49.60 | 98.0% | – |
| AICD (Liu, 2024) | ICL | GPT-3.5-Turbo | – | SVAMP | 51.70 / 20.70 | 250.0%* | – |
| UniversalNER (Zhou et al., 2023c) | IF | GPT-3.5-Turbo | – | UNIVERSAL NER | 41.70 / 34.90 | 119.0% | Link |
| LaMini-LM (Wu et al., 2023c) | IF | $Alpaca_{7B}$ | 9.00 | OpenBookQA | 34.00 / 43.20 | 79.0% | Link |
| Lion (Jiang et al., 2023b) | IF | $GPT_{175B}$ | 25.00 | BBH | 32.00 / 48.90 | 65.0% | Link |
| Fine-tune-CoT (Ho et al., 2022) | CoT | $GPT_{175B}$ | 26.00 | SVAMP | 30.33 / 64.67 | 47.0% | Link |
| Li et al. (Li et al., 2022e) | CoT | $GPT_{175B}$ | 58.00 | CommonsenseQA | 82.47 / 73.00 | 113.0% | – |
| Magister et al. (Magister et al., 2022) | CoT | $GPT_{175B}$ | 16.00 | StrategyQA | 63.77 / 65.40 | 97.0% | – |
| MCC-KD (Chen et al., 2023a) | CoT | GPT-3.5-Turbo | – | SVAMP | 68.66 / 75.14 | 91.0% | Link |
| CSoTD (Li et al., 2023d) | CoT | $GPT_{175B}$ | 135.00 | CommonsenseQA | 67.00 / 82.10 | 82.0% | – |
| SCOTT (Wang et al., 2023j) | CoT | $GPT neox_{20B}$ | 7.00 | CommonsenseQA | 74.70 / 60.40 | 124.0% | Link |
| Distilling step-by-step (Hsieh et al., 2023) | CoT | $Palm_{540B}$ | 700.00 | SVAMP | 58.40 / 72.30 | 81.0% | Link |
| PaD (Zhu et al., 2023b) | CoT | GPT-3.5-Turbo | – | SVAMP | 36.70 / 57.80 | 64.0% | – |
| TDG (Li et al., 2024g) | CoT | GPT-3.5-Turbo | – | MATH | 6.81 / 3.88 | 175.0%* | Link |
| RevThink (Chen et al., 2024c) | CoT | Gemini-1.5-Pro-001 | 30.00 | CommonsenseQA | 75.76 / 76.72 | 99.0% | – |

# 8 Discussion

## 8.1 Principles of Method Classification

To structure this survey, the reviewed methods are categorized according to their primary contributions across five dimensions: distillation sources, schemes, algorithms, modalities, and applications. While many approaches span multiple categories, each method is assigned based on its central innovation, that is, the main idea or technical novelty the paper introduces, with relevant cross-references provided where appropriate. This principled organization aims to balance clarity with the multi-dimensional nature of knowledge distillation research.

## 8.2 Guidelines for Selecting Knowledge Distillation Strategies

To complement the presented classification, the survey also offers practical guidance for selecting appropriate KD strategies based on common task-specific constraints. While modality and application are typically determined by the nature of the target domain, the selection of distillation source, scheme, and algorithm remains a critical aspect of model design. Table 17 provides a decision-oriented summary to map task scenarios to suitable KD components. This perspective bridges the gap between methodological classification and practical usage, offering a starting point for design decisions in KD, while recognizing that the optimal choice ultimately depends on the specific goals, constraints, and context of each use case.

## 8.3 Quantitative Comparison

Quantitative comparison of existing distillation methods is very challenging. The performance of each method is influenced by numerous hyperparameters and is highly dependent on the weights of the pre-trained teacher model. As a result, providing a clear and fair comparison is nearly impossible, and reported performance

Table 17: Summary of key task constraints in knowledge distillation, with corresponding guidance for selecting suitable strategies, including KD sources, schemes, and suitable algorithms.

| Task Requirement | KD Source(s) | KD Scheme(s) | KD Algorithm(s) |
|---|---|---|---|
| Real-time Inference | Logit-based | Online | Attention-based Adaptive |
| Data Scarcity / Few Shot | Feature-based | Self-distillation | Graph-based Contrastive |
| Model Interpretability | Feature-based | Offline | Attention-based Graph-based |
| Fine-grained Recognition Tasks | Feature-based Similarity-based | Offline | Attention-based Contrastive |
| Limited Compute Budget | Logit-based | Offline | Adaptive |
| Adversarial Robustness | Logit-based | Offline | Adversarial |
| Multimodal Fusion | Feature-based | Offline | Cross-modal |
| Continual/Incremental Learning | Logit-based Similarity-based | Online | Adaptive |
| Multi-task Learning | Logit-based Feature-based | Online | Adaptive Multi-teacher |
| Learning from Noisy Labels | Similarity-based | Self-distillation | Contrastive |

metrics often differ significantly from those in the original papers. Therefore, we have intentionally avoided making direct claims about which method performs better than others.

However, in each section of the survey, we have highlighted the strengths and advantages of each category over others. In addition, Tables 12, 13, 14, 15, and 16 report the performance of various distillation methods across different domains, tasks, and datasets. Where available, we also include results for cases where the teacher and student models have either similar or different architectures. This information enables meaningful comparison, as it demonstrates the impact of varying dataset sizes and the robustness of each method to architectural differences.

## 8.4 Practical and Ethical Aspects

KD can also be studied from various other perspectives, including its use cases beyond model compression, such as enhancing privacy, improving security, and addressing intellectual property concerns. While this is beyond the scope of this paper, it is important to highlight some of the key challenges in these areas.

Recent studies have explored the use of KD to enhance privacy and prevent data leakage. For instance, Fernandez et al. (2023) employed KD to mitigate data memorization in text-to-image generative models, Flemings & Annavaram (2024) applied it to reduce the risk of LLMs generating sensitive information, and Shao et al. (2024a) utilized KD to train a large teacher model without exposing the data of smaller edge devices. On the other hand, Zhao & Zhang (2025) demonstrated that techniques like Data-Free Knowledge Distillation (DFKD) do not inherently guarantee privacy, and Cui et al. (2025) showed that such approaches may even introduce new vulnerabilities in LLMs. KD can also be used to guide the student model—intended as the final model—toward more fair and equitable behavior. (Chai et al., 2022)

Another important real-world consideration of KD is its potential to reduce computational demands, particularly relevant in the context of LLMs, which typically require substantial processing power. By enabling student models to achieve comparable performance with significantly lower resource consumption, KD can contribute to more energy-efficient and environmentally sustainable AI systems (Yuan et al., 2024c). How-

ever, the use of commercial models as teachers in the distillation process may raise legal concerns, as it could violate copyright protections. This limitation must be carefully considered in both research and deployment contexts.

## 8.5 Challenges

While KD is an effective method for improving the performance of smaller networks, it comes with several challenges. Below, some of the most important challenges are discussed.

**Knowledge Extraction**: Finding an appropriate source of knowledge for distillation is one of the main challenges in the distillation process (Heo et al., 2019a; Hsu et al., 2022; Ojha et al., 2023). Although using logits is the most straightforward approach to distillation, its effectiveness varies across different scenarios. For example, in tasks where labels are not available, logit-based distillation is not feasible. Furthermore, with the emergence of foundation models like CLIP, distilling rich features and embeddings has become crucial, which is not achievable through logit-based distillation alone, requiring feature-based and similarity-based distillation methods. In LLMs, this issue is even more pronounced, as some recent LLMs are not open-source (Achiam et al., 2023; Team et al., 2023), restricting access to intermediate features or logits, and making black-box distillation particularly challenging. Existing methods often leverage a combination of different distillation sources to effectively transfer the teacher model's knowledge to the student model.

**Choosing Proper Distillation Scheme**: Selecting an appropriate teacher-student architecture is another challenge, as it depends on the specific context. In scenarios where a pre-trained large model is available, offline distillation tends to be more effective, whereas, in other cases, online distillation may prove to be a better alternative. In self-supervised learning, self-distillation schemes have shown to be highly effective. Additionally, using multiple teachers to train a smaller network is a promising approach, often referred to as a mixture of experts. However, this comes with its own challenges, including the need for more powerful memory and GPUs, as well as the complexity of loading different teachers onto separate resources. In most existing methods, offline distillation is more commonly used. Specifically, when distilling from foundation models or LLMs, training the teacher alongside the student is not feasible due to the immense number of parameters in the teacher model. Table 17 presents recommended distillation strategies tailored to specific conditions.

**Capacity Gap**: Another important challenge is the capacity gap between the teacher and student models. It may be assumed that a larger teacher always leads to higher performance improvement in the student network, but when the teacher and student networks have a considerable size difference, it can cause capacity gap issues (Cho & Hariharan, 2019; Guo et al., 2020a; Huang et al., 2022a). This is because the receptive fields of the teacher and student models differ significantly. This challenge is even more pronounced in foundation models, as they contain a tremendous number of parameters, making the distillation of their knowledge into a smaller model challenging. Furthermore, in black-box distillation of LLMs, where intermediate layers are not accessible, directly transferring knowledge from the teacher to the student is a serious challenge. Existing methods employ various algorithms to reduce this gap and make the teacher's knowledge more comprehensible for the student.

**Architectural Difference**: Differences in the architecture of the teacher and student models present another significant challenge, which can impact the performance of the distillation method. In scenarios where the teacher and student have different architectures, which is very common, transferring proper knowledge becomes a new challenge. This issue becomes apparent when distilling knowledge from a teacher to a student with a different architecture, often leading to performance degradation (Cho & Hariharan, 2019; Guo et al., 2020a). In foundation models, the student's architecture can differ significantly from that of the teacher's. Although logit-based distillation remains effective in such scenarios, feature-based and similarity-based distillation methods face additional challenges. More specifically, selecting appropriate layers with similar semantic information for feature distillation plays an important role in the performance of the method, as the teacher and student may even have different inductive biases. In black-box distillation for LLMs, CoT distillation can help address this problem, as it does not force the student to mimic the exact intermediate layers of the teacher. Instead, it provides the student with intermediate reasoning steps derived from the teacher. However, generating proper reasoning and addressing the capacity gap between the teacher

and student remain significant challenges. Tables 12, 13, 14, and 15 present the effect of teacher-student architectural differences on the performance of knowledge distillation methods.

## 8.6 Future Directions

Although KD is not a new concept and has been around since its initial proposal, the number of new methods being proposed is rapidly increasing, along with its expanding applications in various fields. This presents a significant opportunity for further research. In the following, potential future directions are discussed.

**Feature-based Distillation**: While logit-based and similarity-based distillation are effective, recent trends and results show that feature-based distillation can lead to even greater improvements in student performance. However, it can also be combined with other methods. Recent feature-based distillation approaches have demonstrated that instead of defining complex relationships, a simple transformation on the features of either the teacher or student is sufficient to achieve SOTA results in enhancing the student's performance (Liu et al., 2023c; 2024g). Furthermore, with the rapid growth of foundation models like CLIP, distilling embeddings from these large-scale models is gaining increasing attention.

**Adaptive Distillation**: Despite the existence of numerous distillation approaches, effectively distilling informative knowledge from the teacher to the student remains largely unexplored (Hsu et al., 2022; Ojha et al., 2023). Several adaptive distillation methods have been proposed to effectively transfer the dark knowledge of the teacher while ignoring unused knowledge. This line of research is particularly interesting, as it prevents the distillation of incorrect knowledge to the student and helps reduce the gap between the teacher and student networks. This process closely resembles human learning: a teacher transfers knowledge to students, but directly transferring all knowledge is not always effective or practical (Cho & Hariharan, 2019; Mirzadeh et al., 2020). Therefore, various forms of adaptive distillation can be employed, such as using a teacher assistant, decreasing the impact of distillation toward the end of training, or applying adaptive loss functions based on the importance of samples. All of these approaches align with real-world learning scenarios in human training processes.

**Distillation from Foundation Models**: With the emergence of a variety of foundation and multi-modal models, they have become a rich source of knowledge for distillation (Zhang et al., 2024d). For example, CLIP, with its image and text encoder, can provide student models with rich embeddings, where text can be used to improve vision tasks such as weakly-supervised semantic segmentation, making it a hot topic in research (Xie et al., 2022). Additionally, due to the general-purpose features of these large-scale models, they can be leveraged for open-vocabulary tasks (Liu et al., 2024f), where the task is not limited to predefined classes. These interesting applications of distillation present great potential for future research.

**Distillation in LLMs**: While distillation is a highly effective approach in vision, text, and speech tasks, its application in LLMs is even more important. As stated earlier, LLMs have been shown to be overparameterized (Kaplan et al., 2020; Fischer et al., 2024) with billions and even trillions of parameters, far exceeding the size of models in other fields. These weight matrices are often sparse (Hu et al., 2021), making distillation crucial in reducing model size while maintaining performance. Additionally, most powerful models are not open-source (Achiam et al., 2023; Team et al., 2023), increasing the importance of black-box distillation. CoT distillation is a particularly interesting technique for transferring reasoning capabilities from an LLM to an SLM. Unlike conventional distillation methods, CoT aims to enhance the generalization ability of the student model by teaching it how to reason. Results show that by distilling the knowledge of an LLM, the student model can achieve performance comparable to the teacher while significantly reducing model size. This is especially important because the teacher model may be too large to run on common GPUs, whereas the distilled student model can be efficiently deployed on common hardware, making advanced AI more accessible to to a wider audience.

Besides its role in training SLMs, a promising new direction is the use of KD during the pre-training stage of LLMs in industry (Yang et al., 2025; Team et al., 2025). Training LLMs of different sizes is highly cost-intensive. KD plays an important role in creating smaller versions of a pre-trained LLM, where techniques such as quantization (Liu et al., 2023b) or pruning (Sreenivas et al., 2024; Muralidharan et al., 2024) are applied to the large model to extract a compact student model. This process can be used in a post-training setting, requiring only a small fraction of the dataset originally used to train the teacher model.

**Applications of Distillation**: In addition to proposing novel distillation methods, distillation has been applied across various tasks, resulting in significant performance improvements. For example, distillation has been successfully applied in adversarial attacks (Papernot et al., 2016b; Goldblum et al., 2020), mitigating backdoor attacks (Li et al., 2021a; Han et al., 2024b), and anomaly detection (Salehi et al., 2021). Furthermore, the concept of distillation is utilized in dataset distillation (Wang et al., 2018a), making it an interesting research direction. Another application of distillation is its integration with other compression methods, such as quantization or low-rank factorization.

## 9   Conclusion

Knowledge distillation has revolutionized the field of model compression and has played a significant role in various research topics in recent years. In this work, a comprehensive survey on knowledge distillation was proposed, reviewing its methods from various perspectives, including: sources, schemes, algorithms, modalities, applications, and performance comparisons. In contrast to existing surveys, this survey presented recent advancements in the field and focuses on the most important topics in KD. Adaptive and contrastive distillation were presented as two important algorithms that have gained increased attention in recent years. Additionally, KD was reviewed across different modalities, particularly for 3D inputs such as point clouds, which have established important connections with KD in recent research. Furthermore, the applications of distillation in self-supervised learning, diffusion models, foundation models, and LLMs was explored. Self-supervised learning methods primarily rely on self-distillation techniques, diffusion models utilize distillation to reduce the number of steps in the generation phase, and LLMs are distilled into smaller versions KD was reviewed across different modalities, particularly for 3D inputs such as point clouds, which have established important connections with KD in recent research. Furthermore, with recent advancements in foundation models like CLIP, distilling their rich knowledge into other models has become increasingly widespread.

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
