# OpenReview forum: "A Comprehensive Survey on Knowledge Distillation"
_TMLR — Accepted by TMLR_

### Review · Reviewer_PGQU · 2025-06-08

**Summary Of Contributions:**

This paper presents a comprehensive survey of knowledge distillation (KD), a model compression technique that transfers knowledge from large, overparameterized teacher models to smaller student models for efficient deployment. It categorizes KD methods based on the source of distilled knowledge (logits, features, similarities), training schemes (offline, online, self-distillation), and algorithmic strategies (attention-based, adversarial, contrastive, multi-teacher, graph-based, cross-modal, and adaptive methods). The survey also extends prior work by including underexplored domains such as KD for 3D data, diffusion models, vision-language models, and large language models (LLMs). Despite its breadth and clear taxonomy, the paper lacks a critical comparison of methods, quantitative benchmarks, and deeper coverage of LLM-specific distillation challenges, which limits its utility for practitioners. Overall, it serves as a valuable, up-to-date reference but would benefit from more analytical depth and practical guidance.

**Audience:**

Yes

**Broader Impact Concerns:**

The paper currently lacks a discussion on broader societal or ethical impacts. I recommend the authors include a dedicated section addressing potential implications of knowledge distillation. In particular:

* Intellectual Property (IP) Risks: Many teacher models used in distillation, especially in the LLM domain, are proprietary or commercial (e.g., GPT-4, Claude, Gemini). Using these models for generating soft labels or synthetic data raises potential IP and licensing concerns, especially when the distilled student is redistributed or deployed in commercial settings.

* Privacy Concerns: When feature representations or internal activations of the teacher model are used as supervision, especially in medical or user-facing applications, privacy leakage can occur — either from training data embedded in the teacher or from the generation of teacher-augmented data. This risk is amplified when the teacher was trained on sensitive or non-public datasets.

* Synthetic Data Misuse: Data-free or synthetic data distillation approaches may also produce content that reflects biases or unintended memorization from the teacher, potentially leading to misuse or ethical concerns in downstream deployment.

A brief discussion of these risks — along with possible mitigations such as teacher model licensing, synthetic data filtering, and privacy-preserving KD — would significantly strengthen the paper's completeness and real-world relevance.

**Claims And Evidence:**

Yes

**Requested Changes:**

1. It would be good if the authors could add comparative summary table on the comparison of different methods (e.g. Accuracy gain, Computational cost, Training stability, Applicability across domains (CV, NLP, LLMs), Sensitivity to teacher-student mismatch, Data efficiency or privacy-friendliness).

2. Discuss their trade-offs and contextual recommendations of different methods.

3. Add the coverage of distillation in LLM (please kindly refer to the weakness)

**Strengths And Weaknesses:**

## Strengths:
* Extensive Coverage of Recent Work

    The survey provides a detailed taxonomy of KD, including sources (logit, feature, similarity), schemes (offline, online, self-distillation), and algorithms (attention-based, adversarial, multi-teacher, etc.).

    It goes beyond earlier surveys by including adaptive and contrastive distillation, as well as recent work in diffusion models, 3D tasks, foundation models, and LLMs — areas often neglected.

* Clear Taxonomy and Structured Presentation

    The paper has a well-organized hierarchical structure and figures (e.g., Figure 2 and 3) that make the categorization intuitive.

    Each subsection offers tables summarizing key methods with concise descriptions, making it easy for readers to compare them (table 2-5).

* Comparative Table of Past Surveys

    The paper includes a meta-comparison table (Table 1) contrasting this survey with prior surveys (e.g., Gou et al., 2021), clarifying its novelty and broader scope.

* Focus on Underexplored Modalities and Applications

    Inclusion of 3D data, multi-view inputs, and cross-modal KD fills a gap left by older surveys that largely focus on CV and NLP.

    Applications such as adversarial robustness, self-supervised learning, and knowledge distillation in LLMs are well-articulated.

## Weakness:
* The paper is descriptive rather than critical enough:

    There is no (enough) discussion of when/why some methods outperform others — e.g., when logit-based KD fails but feature-based succeeds.

* No analysis of trade-offs: e.g., effectiveness vs. computational cost; distillation depth vs. performance; teacher-student architecture mismatches.

    As a result, the reader gets a broad view of the field but lacks a clear decision-making guide.

* For LLM-related knowledge distillation, the paper lacks some important advances from industry (especially in pre-training stages).

    For example, "Compact Language Models via Pruning and Knowledge Distillation" (Nvidia Minitron, Neurips'24) introduces their design and effort in distilling the smaller LLMs in pretraining stage and explores different design choices. Also in technical reports of Gemma 3, Qwen-3, etc, they introduces their application of distillation to industrial LLMs.

---

> ### Author Response · Authors · 2025-08-05
> **Detailed Responses to the Reviewer’s Feedback**
>
> First, we would like to thank you for your valuable feedback on our paper. Please find our responses to your concerns below:
> > It would be good if the authors could add comparative summary table on the comparison of different methods (e.g. Accuracy gain, Computational cost, Training stability, Applicability across domains (CV, NLP, LLMs), Sensitivity to teacher-student mismatch, Data efficiency or privacy-friendliness).
>
> Thank you for your comment. We fully agree with the importance of the requested information. However, gathering all of it is not practically feasible due to the complexity of the task. Very few methods quantitatively discuss the training complexity and stability of their approach, making it extremely difficult to perform a fair and consistent comparison.
>
> Furthermore, reproducing results for all existing methods is not applicable, as each method involves many hyperparameters—particularly the **weights of the pre-trained teacher model** used during the distillation process. This variability is one of the main reasons why different papers often report inconsistent results for the same methods, deviating from those originally published.
>
> In addition, it is worth noting that the earliest distillation methods—and many subsequent ones—were primarily designed for image classification and segmentation tasks, and more recently has a growing body of work focused specifically on distillation techniques for LLMs.
>
> Despite these challenges, we have updated our tables and added two new ones. These tables **(Table 12, 13, 14, 15)** compare the performance of each method on two different datasets for classification and two different datasets for semantic segmentation. For each method, we report results for both similar and different teacher-student architectures. We believe this directly addresses two of your concerns: evaluating methods across **datasets of varying sizes**, and analyzing the robustness of distillation methods under **different architectural** configurations.
>
> > Discuss their trade-offs and contextual recommendations of different methods.
>
> Regarding your concern, we refer back to your initial comment. We have updated the tables **(Tables 12, 13, 14, 15)** to better highlight comparisons between methods under different scenarios. In addition, we have added a new table **(Table 17)** that provides **guidance on selecting the most suitable distillation method** based on various practical conditions, such as real-time inference, data scarcity or few-shot scenarios, model interpretability, and so on.
>
> Furthermore, a new subsection has been included in the Discussion section **(Subsection 8.2)** to analyze and elaborate on this table. We believe that this addition will significantly help readers understand the trade-offs among different methods and make informed decisions about which technique is most appropriate under specific constraints.
>
> > Add the coverage of distillation in LLM
>
> Thank you for your valuable suggestion. I found the suggested direction very interesting and necessary to include in the survey. In response, we have investigated recent papers that explore the application of KD during the pre-training stage of LLMs.
>
> These works have been incorporated into the LLM subsection of the paper. They include approaches where KD is combined with pruning and quantization for post-training of SLMs, as proposed in NVIDIA research. Additionally, we have discussed distillation strategies employed in LLMs such as Gemma3 and Qwen3.
>
> Below are some of the references we have added:
> - Qwen3 technical report
> - Gemma 3 technical report
> - Llm-qat: Data-free quantization aware training for large language models
> - Compact language models via pruning and knowledge distillation
> - Llm pruning and distillation in practice: The minitron approach
>
>
> > Broader societal or ethical impacts
>
> Thank you for suggestion. We fully agree that this is a critical and often under-addressed dimension of the field, especially with the increasing use of proprietary LLMs and sensitive data.
>
> In response, we have added a new subsection titled “Practical and Ethical Aspects” **(Section 8.4)** to the paper. This section highlights key concerns such as:
>
> - Intellectual Property (IP) Risks, particularly when using commercial teacher models;
> - Privacy vulnerabilities, including risks of memorization or leakage when using internal representations or synthetic data;
> - Bias and fairness issues that may arise through data-free KD or teacher-guided behavior;
> - Environmental and legal implications, including sustainability benefits and licensing challenges.
>
> We believe this addition significantly strengthens the completeness and real-world relevance of the paper and are grateful for your suggestion.

---

### Review · Reviewer_HjiW · 2025-06-11

**Summary Of Contributions:**

This paper looks at the Knowledge Distillation problem and aims at providing a comprehensive survey on KD. The paper explores KD from various perspectives, including distillation sources (logits, features, similarity), schemes (offline, online, self-distillation), algorithms (attention, adversarial, multi-teacher, cross-modal etc.), modalities (image, text, speech, video, and a strong focus on 3D/multi-view), and diverse applications. The authors highlight recent advancements, particularly in feature-based distillation, and dedicate significant attention to the applications of KD in diffusion models, 3D inputs, foundational models (like CLIP, SAM, ViTs), transformers, and Large Language Models (LLMs). Different from existing survey paper, as claimed in this work, this paper covers more recent work on KD and present in a new structure by covering more directions and emphasizing on several popular aspects. It also includes performance comparisons on standard benchmarks and concludes with a discussion of current challenges and future research directions in the field of KD.

**Audience:**

Yes

**Broader Impact Concerns:**

The paper primarily analyzes existing methods in KD and does not introduce new techniques itself. Perhaps the authors can add a section that suggests the need for responsible development and bias mitigation strategies in distilled models.

**Claims And Evidence:**

Yes

**Requested Changes:**

1. A brief discussion on best practices for benchmarking KD or the challenges in establishing fair comparisons.

2. Adding section for providing more explicit guidance on selecting appropriate KD strategies.

3. Proofread the paper.

**Strengths And Weaknesses:**

+ ***Strengths***

1. The paper covers a broad spectrum of KD techniques and it thoroughly reviews recent literature, including KD for LLMs, foundation models (CLIP, SAM, ViTs), diffusion models, and 3D data.

2. The paper emphasizes on KD for 3D inputs and multi-view data is very interesting and important.

3. In general, the paper is well-structured.

- ***Weaknesses***

1. As mentioned in the paper, it is inherently difficult to provide fair performance comparisons in KD. While Tables 12 and 13 are useful starting points, the impact of varying teacher-student architectures, training hyperparameters (e.g., loss weights), and specific datasets can be substantial. A brief discussion on best practices for benchmarking KD or the challenges in establishing fair comparisons could add further value. Additionally, CIFAR-100 and Cityscapes are relatively small compared to ImageNet etc. And the generalizability of KD effectiveness observed on these smaller datasets to very the larger and more complex data regimes can be limited.

2. While this survey paper is comprehensive in describing methods, it would be more beneficial if the paper can provide more explicit guidance on selecting appropriate KD strategies. For example, a decision tree or a section discussing how task requirements (e.g., real-time inference, data scarcity, model interpretability) might influence the choice of KD source, scheme, or algorithm could be a valuable addition.

3. There are some typo and writing errors: for example, "Buciluundefined et al. (2006)" on page 1 should be "Bucilua et al. (2006)"

4. Some related work is missing:

[A] Xu, Yangyang, Yibo Yang, and Lefei Zhang. "Multi-task learning with knowledge distillation for dense prediction." Proceedings of the IEEE/CVF International Conference on Computer Vision. 2023.

[B] Li, Wei-Hong, and Hakan Bilen. "Knowledge distillation for multi-task learning." European Conference on Computer Vision. Cham: Springer International Publishing, 2020.

[C] Li, Wei-Hong, Xialei Liu, and Hakan Bilen. "Universal representations: A unified look at multiple task and domain learning." International Journal of Computer Vision 132.5 (2024): 1521-1545.

---

> ### Author Response · Authors · 2025-08-05
> **Detailed Responses to the Reviewer’s Feedback**
>
> First, we would like to thank you for your valuable feedback on our paper. Please find our responses to your concerns below:
>
> > A brief discussion on best practices for benchmarking KD or the challenges in establishing fair comparisons.
>
> Thank you for your valuable feedback. As you rightly pointed out, a fair comparison of existing methods is challenging due to differences in training hyperparameters—particularly the weights of pre-trained teacher model. This issue is especially pronounced in semantic segmentation tasks, where different papers often report inconsistent results for the same methods, diverging from the numbers originally reported.
>
> Regarding your request to include a table comparing the impact of training hyperparameters, we respectfully believe that this is not feasible. Very few studies explicitly evaluate the robustness of their methods to varying hyperparameters. Moreover, as previously mentioned, reproducing the results of all methods is impractical due to the diverse teacher models used across papers. (Please refer to **Section 8.3**, which has been added to the paper to outline **best practices for benchmarking KD** and to discuss the challenges involved in establishing fair comparisons.)
>
> That said, we greatly appreciate your two other comments. In response, we have added two new tables and updated the existing ones. Following your suggestion, we now include comparative results on the **ImageNet dataset** (for classification) and the **PASCAL VOC** dataset (for segmentation). Additionally, where available, we report results for scenarios in which the teacher and student models have **different architectures**. These updates can be found in **Tables 12, 13, 14, 15,** in the revised paper.
>
> We believe that these additions significantly enhance the comprehensiveness of our quantitative results. The expanded tables help readers better understand the generalization capabilities of each method on **large-scale datasets** and their robustness to **architectural differences** between teacher and student models. Thank you again for your suggestions.
>
> > Adding section for providing more explicit guidance on selecting appropriate KD strategies.
>
> We appreciate this helpful suggestion. We agree that including practical guidance beyond the methodological classification is a valuable addition. In response, we have added a new **subsection (8.2)** to the survey’s “Discussion” section, where we have provided a **task-oriented perspective** on knowledge distillation design. Specifically, we discuss how common task constraints, such as real-time inference, data scarcity, model interpretability, along with other common design requirements, can influence the choice of KD strategy. To assist readers, we have included a **decision-oriented table (Table 17)** that maps typical task scenarios to suitable KD components.
>
>
> > Proofread the paper.
>
> Thank you for your feedback. Due to the large volume of content in the survey, all authors have thoroughly proofread the paper again. In addition, we involved third parties to review the manuscript and used AI-based methods to ensure that all typographical errors were corrected.
>
> > Missing citations
>
> Thank you for your detailed suggestion. We found the suggested references to be valuable contributions to the field, and all have been cited in the appropriate sections. Furthermore, we have reviewed newly published papers in the field, particularly those appearing in top venues, and incorporated them to ensure the survey remains up to date.

---

### Review · Reviewer_7t1b · 2025-06-14

**Summary Of Contributions:**

This paper presents a comprehensive survey on Knowledge Distillation (KD) methods, aiming to provide an updated and systematic review of the field. The main contributions include:

1. **Comprehensive categorization framework**: The authors organize KD methods across five dimensions: distillation sources (logit-based, feature-based, similarity-based), algorithms (attention-based, adversarial, multi-teacher, etc.), schemes (offline, online, self-distillation), modalities (3D, multi-view, text, speech, video), and applications (foundation models, LLMs, diffusion models, etc.).

2. **Focus on recent developments**: Special attention is given to emerging areas including adaptive distillation, contrastive distillation, 3D input distillation, and applications to foundation models and Large Language Models (LLMs).

3. **Expanded scope**: Coverage of previously under-explored areas such as 3D point cloud distillation and multi-view input distillation, addressing gaps in existing surveys.

**Audience:**

Yes

**Claims And Evidence:**

Yes

**Requested Changes:**

1. Provide quantitative analysis. Include performance comparisons and meta-analysis of different KD approaches rather than just categorical descriptions. This is essential for a meaningful survey contribution.

2. Clarify survey coverage and progress. Explicitly indicate which sections of Figure 2 are already covered in the current published survey.This transparency will help readers understand the survey's current scope and final coverage.

3. Establish clear classification principles for multi-category papers. Many papers in knowledge distillation belong to multiple categories simultaneously (e.g., a method might be both feature-based and contrastive, or apply to both transformers and foundation models). The authors should:

- Define explicit principles for handling papers that span multiple categories
- Specify whether papers should be classified by their primary contribution, listed in all applicable categories, or handled through a hierarchy of classification

4. Create a comprehensive paper classification matrix: Develop a master reference table that enables readers to instantly identify the classification of any paper across all dimensions. This should include:

- A multi-dimensional classification table with papers listed alphabetically (maybe) and columns for each category (Sources: Logit/Feature/Similarity; Algorithms: Attention/Adversarial/Multi-teacher/etc.; Schemes: Offline/Online/Self-distillation; Modalities: 3D/Image/Text/etc.; Applications: LLMs/Foundation Models/etc.)
- A searchable index organized by paper title, first author, and year for quick lookup
- Cross-reference citations throughout the main text that direct readers to this classification table
- Online supplementary material (if possible) with an interactive version that allows filtering by multiple criteria

**Strengths And Weaknesses:**

**Strengths**

1.The paper covers a broad range of KD methods and provides a well-structured taxonomy that helps readers navigate the extensive literature.

2. The inclusion of foundation models, LLMs, and diffusion models addresses current research trends that previous surveys have missed.

3. The authors clearly articulate the limitations of existing surveys and justify the need for this comprehensive update.

**Weaknesses**

1. Limited novelty in core content. While comprehensive, much of the content appears to be an organizational exercise rather than providing new insights into KD mechanisms or principles. From the provided portion, it's unclear how the authors will provide meaningful comparisons between different methods beyond simple categorization.

2. Missing critical analysis: The survey format lacks deep analysis of why certain approaches work better than others or fundamental principles governing successful knowledge transfer.

---

> ### Author Response · Authors · 2025-08-05
> **Detailed Responses to the Reviewer’s Feedback**
>
> we would like to thank you for your valuable feedback. Please find our responses below:
>
> > Provide quantitative analysis. Include performance comparisons and meta-analysis of different KD approaches rather than just categorical descriptions. This is essential for a meaningful survey contribution.
>
> Thank you for your suggestion. We agree that quantitative analysis of existing methods complements the categorical classification. Accordingly, we have added two new tables for quantitative comparison and updated the existing two tables. The paper now includes four tables **(Table 12, 13, 14, and 15)**: the first two present comparisons on the ImageNet and CIFAR-100 datasets (for classification tasks), while the latter two focus on the Cityscapes and PASCAL VOC datasets (for semantic segmentation).
>
> Furthermore, we have included results for scenarios where the teacher and student models use different architectures. Presenting results across datasets of varying scales and architectural differences allows readers to assess the robustness of each method with respect to **data size** and **architecture design**.
>
> In addition, we have added a new subsection in the discussion section **(Section 8.3)**, where we conduct a meta-analysis of the existing methods based on their reported performance.
>
>
> > Clarify survey coverage and progress. Explicitly indicate which sections of Figure 2 are already covered in the current published survey.This transparency will help readers understand the survey's current scope and final coverage.
>
> Thank you for your suggestion. While we had already included a comparison between our survey and previous works in **Table 1**, we had not specifically highlighted which topics are new and have not been covered in prior surveys. To improve clarity, we have made the **new topics bold** in the table and updated the caption to reflect this change. The caption now explicitly states that new topics are indicated in bold, which we believe will make the advancements in our survey clearer to the readers and fully address your concern.
>
>
>
> > Establish clear classification principles for multi-category papers. Many papers in knowledge distillation belong to multiple categories simultaneously (e.g., a method might be both feature-based and contrastive, or apply to both transformers and foundation models).
>
> We thank you for this insightful observation. Indeed, many knowledge distillation methods **span multiple categories**, and explicitly stating the principle of our classification enhances the clarity of the survey.
>
> In this survey, we categorized existing KD works according to five key dimensions: distillation sources, schemes, algorithms, modalities, and applications. This classification was done through a close examination of the literature, where we grouped each method based on its novel and primary contribution, specifically, how, when, and why KD was used within the proposed framework.
>
> We acknowledge that many KD methods span multiple categories, and to maintain clarity and coherence, we classified such works according to their **central innovation**, while **cross-referencing** them when appropriate in related sections. This approach strikes a balance between comprehensive coverage and structural readability, while also recognizing the inherently **multi-dimensional** nature of KD research. We agree that stating this classification principle in the survey strengthens the structure of the paper and helps readers better navigate the literature; therefore, we have now clarified our categorization principle in a newly added subsection to the survey’s “Discussion” section **(Subsection 8.1)**.
>
> > Create a comprehensive paper classification matrix: Develop a master reference table that enables readers to instantly identify the classification of any paper across all dimensions.
>
> While we appreciate the suggestion to include a large, detailed table summarizing all surveyed papers (e.g., schema, algorithm, application, modalities, etc.), we believe that such an extensive table may be more overwhelming than helpful for readers. KD is inherently a **multi-dimensional topic**, and many papers fall into multiple **overlapping categories**. In many cases, clear-cut classification is not feasible due to the interdisciplinary nature of the contributions.
>
> In our survey, we chose instead to highlight each paper based on its **central idea or the novelty** that contributes meaningfully to the field. However, we fully agree that a searchable and structured list of papers is highly beneficial. To address this, we have curated a categorized and chronologically organized list on our lab’s GitHub repository **(which will be made publicly available upon publication to preserve anonymity during the review process)**, which allows researchers to filter by section and access code when available. We believe this approach offers a more practical and scalable solution than a static and potentially cluttered table.

---

### Decision · Action_Editor_HMY9 · 2025-09-16

**Recommendation:** Accept with minor revision

**Audience:**

Yes

**Audience Explanation:**

As the topics influences a variety of recent AI topics, I believe researchers from general AI community might be interested in the paper.

**Claims And Evidence:**

Yes

**Claims Explanation:**

This is a long survey paper about knowledge distillation. The paper covers a large number of popular distillation scenarios from different perspectives, including distillation sources, algorithms, schemes, modalities and applications. Initially the paper lacks some aspects to make it a good survey paper and support its claims, e.g., lacks of a systematic comparison between different distillation methods and insights and analysis on which methods are better. The authors took a full consideration of the comments from the reviewers and add further comparisons, results and discussions, which I believe has sufficiently resolved the reviewers' concerns.